# CREIMBO: CROSS-REGIONAL ENSEMBLE INTERACTIONS IN MULTI-VIEW BRAIN OBSERVATIONS

**Noga Mudrik**
Biomedical Engineering, Kavli NDI, CIS
The Johns Hopkins University
Baltimore, MD, USA.
nmudrik1@jhu.edu

**Ryan Ly**
Scientific Data Division
Lawrence Berkeley National Laboratory
Berkeley, CA, USA.
rly@lbl.gov

**Oliver Rübel**
Scientific Data Division
Lawrence Berkeley National Laboratory
Berkeley, CA, USA.
oruebel@lbl.gov

**Adam S. Charles**
Biomedical Engineering, Kavli NDI, CIS
The Johns Hopkins University
Baltimore, MD, USA.
adamsc@jhu.edu

## ABSTRACT

Modern recordings of neural activity provide diverse observations of neurons across brain areas, behavioral conditions, and subjects; presenting an exciting opportunity to reveal the fundamentals of brain-wide dynamics. Current analysis methods, however, often fail to fully harness the richness of such data, as they provide either uninterpretable representations (e.g., via deep networks) or oversimplify models (e.g., by assuming stationary dynamics or analyzing each session independently). Here, instead of regarding asynchronous neural recordings that lack alignment in neural identity or brain areas as a limitation, we leverage these diverse views into the brain to learn a unified model of neural dynamics. Specifically, we assume that brain activity is driven by multiple hidden global sub-circuits. These sub-circuits represent global basis interactions between neural ensembles—functional groups of neurons—such that the time-varying decomposition of these sub-circuits defines how the ensembles' interactions evolve over time non-stationarily and non-linearly. We discover the neural ensembles underlying non-simultaneous observations, along with their non-stationary evolving interactions, with our new model, **CREIMBO** (Cross-Regional Ensemble Interactions in Multi-view Brain Observations). CRE-IMBO identifies the hidden composition of per-session neural ensembles through novel graph-driven dictionary learning and models the ensemble dynamics on a low-dimensional manifold spanned by a sparse time-varying composition of the global sub-circuits. Thus, CREIMBO disentangles overlapping temporal neural processes while preserving interpretability due to the use of a shared underlying sub-circuit basis. Moreover, CREIMBO distinguishes session-specific computations from global (session-invariant) ones by identifying session covariates and variations in sub-circuit activations. We demonstrate CREIMBO's ability to recover true components in synthetic data, and uncover meaningful brain dynamics in human high-density electrode recordings, including cross-subject neural mechanisms as well as inter- vs. intra-region dynamical motifs. Furthermore, using mouse whole-brain recordings, we show CREIMBO's ability to discover dynamical interactions that capture task and behavioral variables and meaningfully align with the biological importance of the brain areas they represent.

## 1 INTRODUCTION

Identifying the interactions between and within brain areas is fundamental to advancing our understanding of how the brain gives rise to behavior. Recent advances in neural recording technologies present an exciting opportunity to study brain-wide interactions by enabling simultaneous recording of neural activity across many brain areas through multiple high-density electrodes. Such experiments,

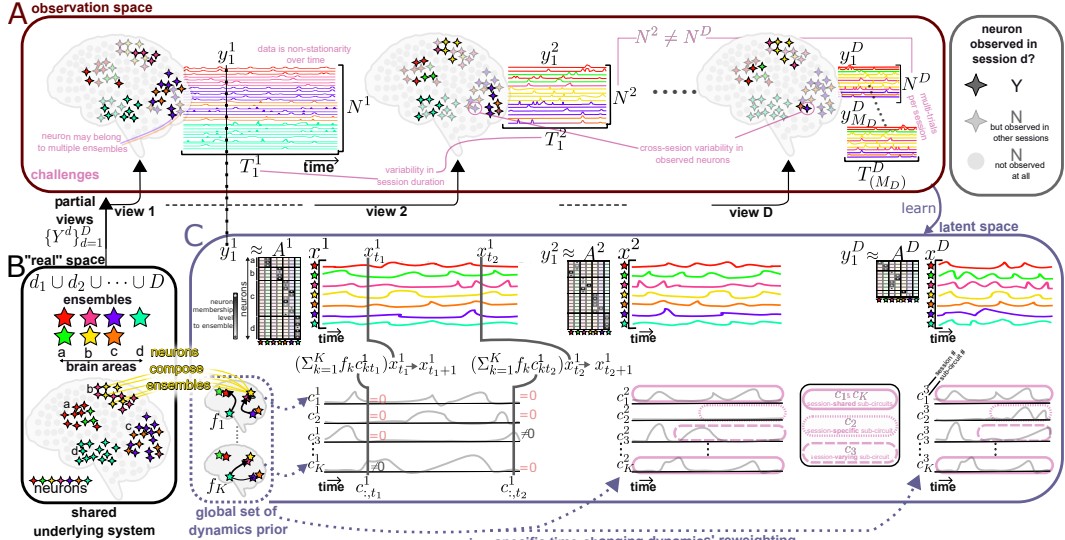

Figure 1: **CREIMBO's Illustration. A:** Real-world multi-regional neural datasets consist of multiple ($D$) non-simultaneous recording sessions ($\{\boldsymbol{Y}^d\}_{d=1}^D$) that cannot be matched in terms of individual neuron identities, quantity, or cross-region distribution, resulting in neurons appearing only in certain sessions (4-pointed stars) or not at all (gray circles). This variability hinder our ability to draw unified conclusions of whole-brain neural computation, while analyzing sessions individually could lead to session-specific biased results. This challenge is further complicated by variability in session (or trial) duration, the non-stationarity of brain dynamics (even within a session), and the presence of multiple trials within each session. **B →A:** Instead of considering these asynchronous sessions as a challenge, we frame them as an opportunity to obtain distinct, potentially complementary "views" onto the shared underlying brain system, thereby facilitating the learning of a unified brain dynamics model. We assume that brain computations are mediated by multiple neural ensembles—★ groups of same-region neurons with shared functionalities—whose interactions yield meaningful neural representations. However, the specific neurons and their membership degrees within each ensemble are unknown, and a neuron may belong to multiple ensembles with varying membership levels based on its diverse functionalities. **C:** CREIMBO leverages partial brain views to co-learn ensemble compositions per session ($\{\boldsymbol{A}^d\}_{d=1}^D$), their temporal activity ($\{\boldsymbol{x}^d\}_{d=1}^D$), and cross-regional interactions $\boldsymbol{x}_{t+1}^d = \boldsymbol{F}_t^d \boldsymbol{x}_t^d$. It posits that these non-stationary interactions stem from a session-shared dictionary prior of up to $K$ global interactions, here termed sub-circuits ($\{\boldsymbol{f}_k\}_{k=1}^K$), whose sparse time-changing decomposition $\boldsymbol{F}_t^d = \Sigma_{k=1}^K \boldsymbol{f}_k \boldsymbol{c}_{kt}^d$ shapes the overall ensemble dynamics at each time point $t$ and session $d$. The sub-circuit temporal coefficients enable distinguishing session-specific interactions from session-shared interactions in different time periods (pink curves in subplot C right). Under certain assumptions (Sec. D), the universal sub-circuit dictionary ($\{\boldsymbol{f}_k\}_{k=1}^K$), shared across sessions, allows CREIMBO to auto-sort ensembles across sessions by functionality, aligning the $j$-th ensemble of session $d$ with the $j$-th ensemble of session $d'$, despite differences in observed neurons.

repeated over many sessions with different settings, offer multiple asynchronous measurements of the brain system, with each session encompassing hundreds of distinct neurons across regions.

Individual recording sessions, however, provide a singular perspective of the brain dynamics, as the reinsertion of recording devices results in the capture of different subsets of neurons in each session. Thus, fitting population-level models often reverts to per-session analysis (i.e., fitting each session independently), which is inherently constrained by the inability to incorporate the full set of recorded activity across all recorded brain areas across all sessions. Moreover, analyzing the data session-by-session may hinder our ability to distinguish between computations that are session- or subject-specific and those that are task-related or globally session-invariant.Compounding the difficulty of merging data from different sessions is the non-linear and non-stationary nature of neural activity, along with noise and variability across trials—both of which require additional model complexity to account for. Moreover, to extract scientifically meaningful insights, models must be interpretable, with model parameters directly related to task variables or connections between recorded units. This gap between the opportunities offered by modern neural data and the limited

capabilities of current methods necessitates new approaches to leverage the richness in modern brain data and discover the fundamental neural sub-circuits governing brain activity.

One current approach used to combine multi-session data into unified model is deep learning. By training to predict all sessions' data, deep networks implicitly merge the datasets into the fit network weights. While these powerful models can combine information across sessions, their uninterpretable nature and complexity typically hinders their ability to reveal the fundamental building blocks of neural computation. Other approaches, including tensor factorization and its variations (e.g., Chen et al. (2015); Harshman (1970)) may be limited in their ability to merge sessions of varying durations or model the interactions between the identified neural components.

Here, we re-frame cross-session recording variability not as a drawback, but as a valuable advantage, viewing this data as providing multiple, complementary "views" into a single, shared brain system (Fig. 1B → A). This approach allows a comprehensive discovery of the underlying system by extracting joint information from the entire dataset, despite individual sessions differing in the identity and quantity of the neurons they cover. We hypothesize that the population dynamics at each session is driven by a universal set of global time-invariant sub-circuits shared across sessions. To effectively capture the non-linear and non-stationary dynamics of the brain, we enable these sub-circuits to present varying activation levels over time and across sessions. This approach allows us to identify evolving neural patterns by analyzing the joint activity of the sub-circuits at each time point, captured through their time-varying decomposition. We assume that these sub-circuits govern the interactions between neural ensembles, which in turn drive the neural dynamics within a latent space shared across sessions, despite variations in recorded neuron identities across sessions. To account for this cross-session variability in neuron identities, we propose that the joint low-dimensional space in which the latent dynamics evolve transforms to each session's observation space via a per-session mapping, while ensuring cross-session alignment of the neural groups' functionalities.

In this work, we lay out this new model, which we term CREIMBO (**C**ross-**R**egional **E**nsemble **I**nteractions in **M**ulti-view **B**rain **O**bservations), and demonstrate its ability to capture latent dynamics in multi-view neural data. Specifically, our contributions include:

- We discover multi-regional brain dynamics through leveraging the richness of modern neural datasets while ensuring interpretability.
- We distinguish between intra- and inter-region interactions by identifying sparse, localized neural ensembles and disentangling co-active global circuits that govern their interactions.
- We accurately recover ground truth components in synthetic data and discover meaningful cross-regional brain interactions underlying human and mouse brain data.

## 2 RELATED WORK

Over the past decade, remarkable advances in neural data acquisition technologies have enabled the measurement of hundreds of neurons across brain regions, subjects, and experimental settings. This has spurred follow-up work to leverage this extensive data for novel neural discoveries, including merging non-simultaneous sessions and examining cross-regional brain interactions, all while accounting for the complex, non-linear, and non-stationary nature of brain activity.

**Merging Non-simultaneous Neural Recording Sessions.** Existing work for combining non-simultaneous neuronal population recordings, here termed "sessions", include Turaga et al. (2013) who introduced the term "stitching" for this problem and modeled neural activity with a shared latent dynamical systems model and Soudry et al. (2013) who proposed to stitch asynchronous recordings for inferring functional connectivity by assuming shared cross-session latent input. However, these methods are not designed to capture more than two simultaneously interacting neural sub-circuits across multiple populations. They also do not distinguish between within- and between-area interactions, nor do they specifically address temporal non-stationarity. Bishop (2015) and Bishop & Yu (2014) proposed integrating the neural activity structure into inference and expanding stitching to study the communication between two neuronal populations from potentially different brain areas. Nonnenmacher et al. (2017) suggested extracting low-dimensional dynamics from multiple sessions by learning temporal co-variations across neurons and time. Both of these methods, though, assume at least a small overlap in the identity of cross-session observed neurons. LFADS (Keshtkaran et al., 2021; Pandarinath et al., 2018) enables stitching while coping with limited neural overlap by finding a shared non-linear latent dynamical model. However, its latent dynamics may be difficult to interpret in relation to the specific neural sub-circuits that compose it, and because it is based on a variational autoencoder, tuning it can be challenging.

**Maintaining Interpretability in Non-Stationary Neural Dynamics Models.** Real-world neural activity follows non-stationary nature due to changing environmental settings. Classical dynamical models often simplify neural dynamics to linear (e.g., jPCA Churchland et al. (2012)) or non-linear but stationary (e.g., Gallego et al. (2020)) processes. Models that capture non-stationarity often rely on "black box" deep learning models (e.g., Schneider et al. (2023); Pandarinath et al. (2018); Zhu et al. (2022)), which are powerful but often present limited interpretability with respect to neural interactions. Other models are built on switching piece-wise linear models Linderman et al. (2016; 2017); Murphy (1998), which do not enable the identification of multiple co-active dynamic processes. dLDS (decomposed Linear Dynamical Systems) (Mudrik et al., 2024a; Chen et al., 2024b)) and its multi-step extension (Mudrik et al., 2024c)), addresses this gap by generalizing the switching assumption to sparse time-changing decomposition of dynamical systems, thus capturing co-active sub-circuits. However, dLDS is not inherently designed to distinguish multi-session within- and between-regional interactions with session-to-session variability in neuron identity.

**Identification of Functional Neural Ensembles.** A potential way to find interpretable loading matrix that links low to high dimensional space is through the identification of functional neural ensembles—here referred to as groups of neurons with similar functionalities—that determine the axes in the latent space. For instance, the mDLAG model (Gokcen et al., 2024) used Automatic Relevance Determination (ARD) to promote population-wise sparsity patterns of each latent variable. Recently, Mudrik et al. (2024b) presented an approach to identify sparse hidden building blocks underlying multi-way data while accounting for trial and condition variability. Their method includes a graph-driven regularization technique that groups co-active units into the same building blocks while pushing apart units with different functionalities. It also enables capturing structural adjustments in block configurations across task conditions and per-trial variability in the temporal activity of the building blocks. However, their approach does not model the dynamic interactions between units nor provide a closed-form prior for the evolution of the functional groups' temporal traces. While other methods, e.g., clustering approaches like (Grossberger et al., 2018) or Sparse Principal Component Analysis (SPCA) (Zou et al., 2006) enable the recognition of underlying sparse groups, they either do not allow a neuron to belong to multiple groups with varying degrees of membership or do not enable the ensembles to present structural adjustments to specific conditions.

**Multi-regional Brain Interactions.** With the ability to record from multiple brain areas and the recognition of the importance of multi-area communication (Pesaran et al., 2021), recent works have explored communication within and between brain areas. In (Gokcen et al., 2022; 2024), the authors proposed an approach to identify time delays in cross-regional brain interactions and to distinguish indirect, concurrent, and bidirectional interactions in two or more populations. However, their model focuses on analyzing sessions individually and is not intended to uncover the full underlying set of transition matrices, but rather to recognize meaningful time delays that can imply interactions. Other models addressed multi-regional interactions through a communication subspace Semedo et al. (2019) using dimensionality reduction, or by Generalized Linear Models (GLMs), with either Poisson (Yates et al., 2017) or Gaussian (Yates et al., 2017) statistics to identify functional coupling between areas. These approaches, though, are not tailored to capture non-simultaneous cross-sessions variability in neuron identity and subject processing. Other recent methods (Karniol-Tambour et al. (2024); Glaser et al. (2020); Li et al. (2024)) were tailored to model multi-regional interactions. Particularly, they have proposed studying brain-wide dynamics using switching dynamical systems approaches, where the model switches between different states over time to capture distinct periods. However, this approach cannot disentangle interactions among multiple simultaneously active sub-circuits, nor can it resolve interactions at different temporal resolutions occurring concurrently.

## 3   PROBLEM INTRODUCTION AND APPROACH

Let $\{\boldsymbol{Y}^d\}_{d=1}^D$ be a set of estimated neural firing rates (e.g. from Neuropixels data) over $D$ asynchronous sessions, indexed by $d = 1 \ldots D$. The neural recordings of each $d$-th session, $\boldsymbol{Y}^d \in \mathbb{R}^{N^d \times T^d}$, capture $T^d$ time observations of the activity of $N^d$ neurons from up to $J$ distinct brain areas, thus offering a *partial* view of the brain system. Note that sessions may refer to the same or different subjects, can vary in duration or number of neurons (i.e., $N^d$ and $T^d$ need not be equal to $N^{d'}$ and $T^{d'}$ for different sessions $d, d'$), and can differ in the subset of brain areas (out of $J$) they capture. Hence, these cross-session data matrices cannot be aligned into a single data array (e.g., via a tensor), which hinders the direct application of existing analyses such as tensor factorization.

We assume that these brain observations reflect the hidden activity $\boldsymbol{X}^d \in \mathbb{R}^{p \times T^d}$ of $p << N^d$ functional neural groups that evolve and interact over time. Each of these groups encompasses a sparse set of same-area neurons, where each neuron can belong to more than one group with varying degrees of cross-group membership. The low-dimensional latent space in which the groups interact is shared across sessions and projected to each session's observed neurons via an unknown per-session projection $g^d : \mathbb{R}^p \to \mathbb{R}^{N^d}$ with $\boldsymbol{Y}^d = g^d(\boldsymbol{X}^d)$.

Since brain dynamics have been shown to be temporally changing (Petersen & Sporns, 2015), we assume that the interactions between these functional groups follow a per-session non-linear and non-stationary dynamics $\boldsymbol{F}_t^d : \mathbb{R}^p \to \mathbb{R}^p$ where $\boldsymbol{X}_t^d = \boldsymbol{F}_t^d(\boldsymbol{x}_{t-1}^d)$. Finding such $\boldsymbol{F}_t^d$, however, is nontrivial (as solving $\arg\max_{\boldsymbol{F}_t^d} p(\boldsymbol{F}_t^d \mid \boldsymbol{x}_t^d, \boldsymbol{x}_{t-1}^d)$ at each time point is intractable due to having more unknowns than equations). Therefore, we follow dLDS (Mudrik et al., 2024a) and assume that these dynamics can be described by a set $\{\boldsymbol{f}_k\}_{k=1}^K$ of $K$ global time- and session-invariant "sub-circuits". Each sub-circuit $\boldsymbol{f}_k \in \mathbb{R}^{p \times p}$ represents a core interaction between ensembles and is reused by the system at different time-points throughout its trajectory. Each of these latent sub-circuits may capture either global session-invariant brain interactions and/or session- and subject-specific interactions. Moreover, these sub-circuits may be active simultaneously or intermittently, yielding a model that can flexibly fit neural trajectories that differ between sessions while drawing upon a single underlying mechanism. CREIMBO thus addresses the problem of leveraging the joint information from asynchronously collected observations $\{\boldsymbol{Y}^d\}_{d=1}^D$ to 1) identify the $K$ latent multi-regional sub-circuits $\{\boldsymbol{f}_k\}_{k=1}^K$ and their non-stationary activation levels, and 2) recognize how they jointly and individually drive the interactions between neural ensembles, and ultimately define their trajectories in a shared-session latent space.

## 4  CREIMBO

Our approach simultaneously fits the global sub-circuits that underlie the inter-ensemble interactions across multiple sessions. It is predicated on three key assumptions: 1) ensembles of neurons, rather than individual neurons, form the basic units that interact in brain dynamics, 2) it is feasible to identify functionally analogous neuronal ensembles across sessions, and 3) the full repertoire of interactions between the neural ensembles is spanned by a global set of linear dynamical systems, with their linear combinations evolving over time to capture the non-stationary dynamics.

Let $\boldsymbol{A}^d$ be a sparse matrix where each column, $\boldsymbol{A}_{:,i}^d$, encodes the neuronal composition of the $i$-th ensemble in the $d$-th session, such that non-zero values represent the membership magnitudes of neurons to this $i$-th ensemble. Since we aim to capture multi-regional interactions, we design the ensemble matrix $\boldsymbol{a} := \boldsymbol{A}^d$ as a block diagonal matrix where the $j$-th block, $\boldsymbol{a}^j \in \mathbb{R}^{n_j \times p_j}$, contains neural ensembles from the $j$-th area only. We denote $p_j$ and $n_j = N_j^d$ as the number of ensembles and the number of observed neurons in that area for session $d$, respectively ($p = \Sigma_{j=1}^J p_j, N^d = \Sigma_{j=1}^J n_j$).

Let $\tau := \mathrm{T}_m^d$ indicate the number of time-points recorded in each $m$-th trial of session $d$, such that the firing rate observations in that trial are captured by $\boldsymbol{y} := \boldsymbol{Y}_m^d \in \mathbb{R}^{N^d \times \mathrm{T}_m^d}$ and arise from the joint activities of these hidden neural ensembles, up to a normally-distributed error $\epsilon \sim \mathcal{N}(0, \sigma)$. Namely, $\boldsymbol{y} = \boldsymbol{a}\boldsymbol{x} + \epsilon$, where $\boldsymbol{x} := \boldsymbol{X}_m^d \in \mathbb{R}^{p \times \tau}$ are the temporal trajectories of the ensembles. Please refer to Section G for Poisson statistics inference in low spiking rate regimes.

The evolution of the ensemble temporal trajectories ($\{\boldsymbol{X}_m^d\}_{m,d}$) is determined via the latent interactions between them, which we model with a non-stationary linear dynamical system $\boldsymbol{x}_{t+1} = \boldsymbol{F}_t\boldsymbol{x}_t$ for $t = 1 \ldots \tau$, where $\boldsymbol{F}_t \in \mathbb{R}^{p \times p}$. A key desired property of our model is to capture the overlapping activity of multiple sub-circuits of interacting ensembles. We thus model the interactions ($\boldsymbol{F}_t$) through a time-varying sparse decomposition $\boldsymbol{F}_t = \sum_{k=1}^K c_{kt}\boldsymbol{f}_k^d$, where $c_{kt} = \boldsymbol{C}_{kt}^d$ are the sub-circuits' time-varying coefficients in session $d$ that capture the modulation of each circuit interactions.

Fitting this model requires identifying the sub-circuits, their per-session coefficients, and the ensemble compositions. Hence, we employ an alternating approach that iterates until convergence, between updating the circuits $\{\boldsymbol{f}_k\}_{k=1}^K$ and inferring, for each session $d$, 1) the ensemble compositions ($\boldsymbol{a} = \boldsymbol{A}^d$), 2) their trajectories ($\boldsymbol{x} = \boldsymbol{X}^d$), and 3) the sub-circuit activations ($c_{kt} = \boldsymbol{C}_{kt}^d$).

**Ensemble Update:** We update $\boldsymbol{a}$ per-row (neuron) $n$ ($\boldsymbol{a}_n$). Following the work of Mudrik et al. (2024b), we infer the sparse ensemble structures with graph-driven re-weighted $\ell_1$ regularization that groups together neurons with shared activity patterns, while pushing apart neurons that do not

behave similarly. The neuron-similarity graph for each area $j$, $\boldsymbol{h}^j \in \mathbb{R}^{n_j \times n_j}$, is calculated based on a data-driven Gaussian kernel that measures the temporal similarity between neurons, such that $\boldsymbol{h}^j_{n1,n2} = exp(\frac{\|\boldsymbol{y}^j_{n1} - \boldsymbol{y}^j_{n2}\|^2_2}{\sigma_h})$, where $\sigma_h$ is a hyperparameter. Specifically, we infer $\boldsymbol{a}$ row-wise, where the $n$-th row ($\boldsymbol{a}_{n,:} \in \mathbb{R}^p$) indicates which ensembles neuron $n$ belongs to and to what degree. Due to the block-diagonal structure imposed on $\boldsymbol{a}$, in practice, we only need to update the membership magnitudes for ensembles within the region to which neuron $n$ belongs (region $j$), while nullifying the membership to the ensembles of other areas. This is done via:

$$\widehat{\boldsymbol{a}}_{n,:} = \arg\min_{\boldsymbol{a}_{n,:}} \|\boldsymbol{y}_{n,:} - \boldsymbol{a}_{n,:}\boldsymbol{x}_{j,:}\|^2_2 + \sum_{j=1}^p \boldsymbol{\lambda}^d_{n,j}|\boldsymbol{a}_{n,j}| \text{ where } \boldsymbol{\lambda}_{n,j} = \frac{\beta_1}{\beta_2 + |\widehat{\boldsymbol{a}}_{n,j}| + \beta_3|\boldsymbol{h}^j_{n,:}\widehat{\boldsymbol{a}}_{:,j}|}, \quad (1)$$

where $\boldsymbol{x}_{j,:}$ is the activity of the ensembles of the $j$-th region and $\beta_1$, $\beta_2$, and $\beta_3$ are scalars that control the effect of the graph on the regularization. The multiplication $\boldsymbol{h}_{n,:}\widehat{\boldsymbol{a}}_{:,j}$ tests the correspondence between 1) the temporal similarity of all neurons to neuron $n$ ($\boldsymbol{h}_{n,:}$), and 2) the membership magnitudes of these neurons to the current estimate of the $j$-th ensemble ($\boldsymbol{a}_{:,j}$). This way, if $\boldsymbol{h}_{n,:}\widehat{\boldsymbol{a}}_{:,j}$ is high (i.e., the neighbors of neuron $n$ on the graph are part of the current estimate of the $j$-th ensemble), $\boldsymbol{\lambda}_{n,j}$ is small, resulting in lower sparsity regularization for including neuron $n$ in that ensemble, which ultimately promotes its inclusion. Alternatively, if $\boldsymbol{h}_{n,:}\widehat{\boldsymbol{a}}_{:,j}$ is small, the regularization weight $\boldsymbol{\lambda}_{n,j}$ is large, and neuron $n$ will be less likely to be included in ensemble $j$.

**Latent State and Dynamic Coefficients Update:** The ensemble trajectories ($\boldsymbol{x}_t$) and the sub-circuits coefficients ($\{\boldsymbol{c}_{k,t}\}_{k=1}^K$) are updated iteratively for each time-point $t$ via the LASSO optimization

$$\widehat{\boldsymbol{x}}_t, \widehat{\boldsymbol{c}}_t = \arg\min_{\boldsymbol{x}_t, \boldsymbol{c}_t} \|\boldsymbol{y}_t - \boldsymbol{a}\boldsymbol{x}_t\|^2_2 + \left\|\boldsymbol{x}_{t+1} - \left(\sum_{k=1}^K \boldsymbol{c}_{k,t}\boldsymbol{f}_k\right)\boldsymbol{x}_t\right\|^2_2 + \lambda_c\|\boldsymbol{c}_t\|_1, \quad (2)$$

where $\lambda_c \in \mathbb{R}$ is the sparsity-regularization weight on the sub-circuit activations.

**Dynamical System Update:** The sub-circuits ($\{\boldsymbol{f}_k\}_{k=1}^K$), which capture global core ensemble interactions, are assumed to be sparse (i.e., not fully interconnected), and are identified directly by:

$$\boldsymbol{F}^{\text{all}} = \arg\min_{\boldsymbol{F}^{\text{all}}} \|\boldsymbol{x}^+ - \boldsymbol{F}^{\text{all}}\boldsymbol{\Psi}\|^2_2 + \lambda_F\|\text{vec}(\boldsymbol{F}^{\text{all}})\|_1 + \lambda_\rho \sum_{k_1, k_2, (k_1 \neq k_2)} \rho(\boldsymbol{f}_{k_1}, \boldsymbol{f}_{k_2}), \quad (3)$$

where $\boldsymbol{F}^{\text{all}} \in \mathbb{R}^{p \times pK}$ is an horizontal concatenation of all $\{\boldsymbol{f}\}$s, and $\boldsymbol{x}^+ \in \mathbb{R}^{p \times (\sum_{d=1}^{\widetilde{D}} T^d)}$ is the horizontal concatenation of all $\{\{\boldsymbol{x}^d_{t+1}\}_{t=1}^{T^d}\}_{d=1}^{\widetilde{D}}$, where in practice this operation is taken on a random subset of sessions $\widetilde{D} \subset D$ for computational complexity considerations. Above, $\boldsymbol{\Psi} \in \mathbb{R}^{Kp \times \sum_{d=1}^{\widetilde{D}} T^d}$ is the horizontal concatenation of all $\{\{\boldsymbol{\Psi}^d_t\}_{t=1}^{T^d}\}_{d=1}^{\widetilde{D}}$, with $\boldsymbol{\Psi}^d_t \in \mathbb{R}^{Kp \times 1}$ defined as $\boldsymbol{\Psi}^d_t = [(\boldsymbol{c}^d_t \otimes [1]_{1 \times p}) \circ ([1]_{1 \times K} \otimes \boldsymbol{x}^d_t)^T]^T$, where $\otimes$ is the Kronecker product, and $\circ$ is element-wise multiplication. The operator $\text{vec}(\cdot)$ flattens a matrix to a vector and $\lambda_F$ promotes sparsity within the sub-circuits. $\rho(\boldsymbol{f}_{k_1}, \boldsymbol{f}_{k_2})$ is a de-correlation term with weight $\lambda_\rho$ used to ensure that distinct $\boldsymbol{f}$s are not too similar. See Algorithm 1 for a method summary and Section E for complexity analysis.

## 5 EXPERIMENTS

**CREIMBO Recovers Ground Truth Components in Synthetic Data**: We first assess CREIMBO's ability to recover ground-truth components from synthetic data. We generated $K = 3$ synthetic sub-circuits represented by a set of 3 rotational matrices (Fig. 5B). These sub-circuits capture interactions among $p = 3$ ensembles from $J = 3$ regions ($p_j = 1$ for all $j = 1, \ldots, J$). Each region consists of a random number of neurons drawn from a uniform distribution between 4 to 9. We generated the observation data $\{\boldsymbol{Y}^d\}_{d=1}^5$ by simulating 5 sessions with different neuron identities and varying numbers of neurons per region (Fig. 5A and D). CREIMBO was able to recover the ground truth components with high accuracy, as measured by correlation with the ground truth, across 312 random noisy initialization repeats with different parameters (Tab. 1). The accurate recovery included all parameters—1) the reconstructed observations $\boldsymbol{y}$ (Fig. 2A), 2) ensemble compositions $\boldsymbol{A}$ (Fig. 2B) and activity $\boldsymbol{x}$ (Fig. 2C), 3) the sub-circuits $\boldsymbol{F}$ (Fig. 2D), and 4) circuit coefficients $\boldsymbol{c}$ (Fig. 2E). When comparing CREIMBO's results to components identified under diverse ablations (Sec. D), we found that approaches that either separate the ensemble identification from the dynamic identification, analyze each session separately, or do not consider the localized structure of the ensembles—fail to

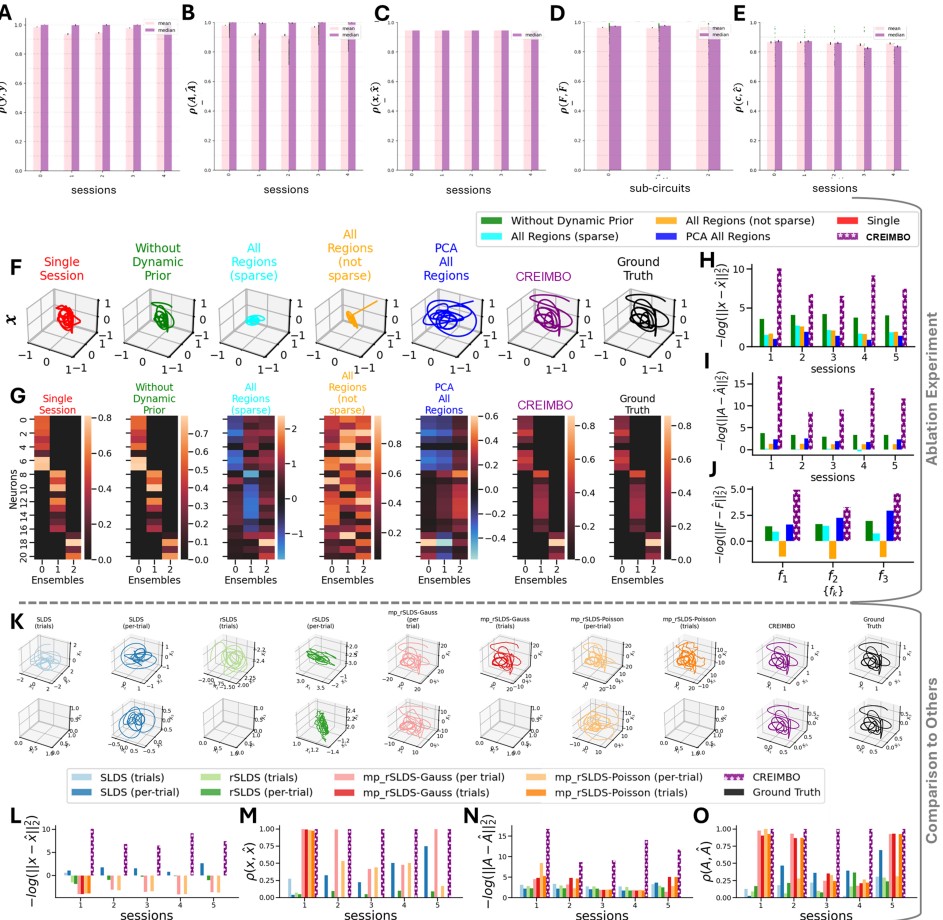

Figure 2: **A-E:** Testing CREIMBO over 312 repeats with varying random initializations and random seeds (Sec. B) reveals high correlations between ground truth and fit components for all unknowns: $Y$, $A$, $x$, $F$, $c$ (subplots **A-E** respectively). **F-G:Ablation Experiment:** Comparison of the ensemble trajectory ($x^d$, subplot F) and the ensemble compositions ($A^d$, subplot G) for the 1-st session as identified by CREIMBO vs. various ablations (details in Sec. D, see Figure 7 for more sessions). **H-J:** Comparing CREIMBO to ablation variants in terms of $(-\log(\text{MSE}))$ between ground truth and identified dynamics, ensembles, and sub-circuits. "Single Session" refers to training CREIMBO on sessions individually. Details in Sec. D. **K: Baseline Comparisons:** The latent dynamics identified by the baselines (SDLS, rSLDS (Linderman et al., 2016), mp-rSLDS (Glaser et al., 2020)) for the first two sessions (details in Sec. D, merged-session baselines provide a single trajectory). **L-O:** Comparison of the baselines' latent dynamics and ensemble matrices with the ground truth.

recover the ground-truth components (i.e., their resulting fit model parameters are less correlated with the ground-truth observations, Fig. 2F-J, 7). We further evaluated CREIMBO's ability to recover hidden components compared to other methods, including SLDS, rSLDS, and mp-rSLDS, and found that CREIMBO more accurately recovers the ground truth hidden dynamics and ensembles (Fig. 2 K-O, Fig. 8, baseline details in Sec. D). Importantly, these existing models are not designed to identify multiple co-occurring processes or sparse ensembles, in contrast to CREIMBO, and they are not suitable for multi-sessions with varying neural identities, which limits the comparison from the outset.

Next, we tested CREIMBO on a richer synthetic example with $D = 15$ sessions. Each session had a varying number of neurons (a random integer between 14 and 19 per region from a uniform distribution) and distinct sub-circuit coefficients (see Fig. 11 for the generated ground truth components and Fig. 9 for regional distributions and masking). The sub-circuits were set as rotational matrices, similar to the previous experiment. Across 204 random initializations with different parameters (see Sec. B), we again found that CREIMBO consistently identifies the ground truth latent dynamics, ensemble compositions, and sub-circuits. Moreover, CREIMBO accurately reconstructed the observations with high correlation to the ground truth across all initializations (Fig. 11E) with low relative error

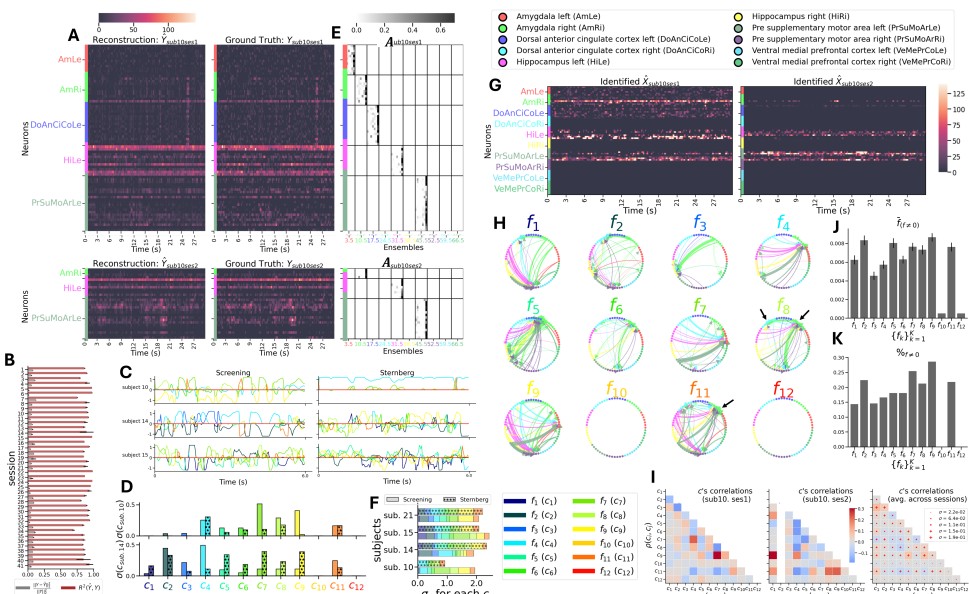

Figure 3: CREIMBO identifies cross-regional neural sub-circuits underlying multi-session human brain recordings. **A:** Two exemplary observations ($Y$, right) compared to their reconstruction by CREIMBO ($\widehat{Y}$, left). **B:** Reconstruction performance in terms of $R^2$ and relative error across all sessions. **C:** The sub-circuit coefficients for 3 exemplary subjects. **D:** The standard deviation (std) of the coefficients over time across the two exemplary sessions. **E:** Exemplary identified sparse ensemble matrices ($\{A^d\}_{d=1}^D$) for the observations from **A**. **F:** Relative std of the coefficients across four subjects reveals usage of similar sub-circuits. **G:** The identified ensembles' activity ($X^d$) for the two exemplary sessions. **H:** The identified sub-circuits. Edge width indicates effect magnitude. Nodes represent ensembles, with colors indicating the ensemble area. Black arrows near $f_8$ and $f_{11}$ highlight within-region interactions. **I:** Pairwise correlations between within-session sub-circuit coefficients (left, middle) and average within-session correlations across all sessions (size of '+' markers indicates std). **J:** Mean and std of $f$s values. **K:** Percentage of non-zero connections per $f_k$.

(Fig. 11F). We further tested CREIMBO on more advanced data that included $D = 40$ sessions with a maximum of $J = 4$ regions and $p_j = 3$ ensembles per region (Fig. 12). We again found that CREIMBO recovers the components with high correlation with the ground-truth (Fig. 13).

**CREIMBO Discovers Multi-Regional Dynamics in Electrophysiology Data:** We then tested CREIMBO on human neural recordings from a high-density electrode array provided by Kyzar et al. (2024b). The data consists of neural activity from overall $J = 10$ brain areas with limited cross-session regional overlap (Fig. 14, 15), recorded while subjects performed a screening task (details in Kyzar et al. (2024b)). The data encompasses 21 subjects across $D = 41$ non-simultaneous sessions. For each subject (except Subject 19), the data offers 1) a "Screening" session, and 2) a "Sternberg test" session (details in Kyzar et al. (2024b)). We first converted the spike-sorted units to firing rate matrices by convolving the spike trains with a 30ms Gaussian kernel (Fig. 10), and then tested CREIMBO on all sessions together with a maximum of $K = 12$ sub-circuits and at most $p_j = 7$ ensembles per region (full parameter list in Tab. 2). First, we note that CREIMBO can accurately reconstruct the data with high accuracy across all sessions (Fig. 3A, B). Interestingly, the ensemble compositions identified by CREIMBO (Fig. 3E) present sparse patterns with one dense ensemble for most regions that we hypothesize capture the "background" mean field activity of that region (right-most column of diagonal blocks in Fig. 3E). Meanwhile, the other ensembles (the sparse ones) likely capture more specialized, nuanced functionality. The ensemble trajectories ($X^d$, Fig. 3G) reveal some constantly active ensembles, implying their importance for neural processing.

The universal sub-circuits ($\{f_k\}_{k=1}^K$) exhibit distinct localized motifs (Fig. 3H), i.e., most sub-circuits present clear trends of either sourcing or targeting the same area. This is in contrast to other approaches (Sec. D), that yield overlapping uninterpretable sub-circuits (Fig. 22) that emphasize only limited circuitry (Fig. 24). Interestingly, sub-circuits $f_{10}$, $f_{12}$ ended up almost empty (mainly zero-ish values, Fig. 3H, 19). This highlights CREIMBO's ability to automatically nullify redundant sub-circuits—i.e., a form of model selection—via the sparsity regularization. The number of active connections and

average connection strength (Fig. 10J, K) show modest variability between sub-circuits, highlighting the model's ability to identify distinct sub-circuits with varying interaction styles, including those with varying numbers of connections, rather than being limited to sub-circuits with fixed sparsity. From a neuroscience perspective, this suggests that underlying neural sub-circuits can vary in density and the number of participating areas and ensembles. Interestingly, most identified interactions occur between distinct areas (i.e., inter-regional interactions) with either multiple ensembles from the same area affecting together ensembles of another area (e.g., $\boldsymbol{f}_3$), or multiple ensembles from diverse areas converging to the same target area (e.g., $\boldsymbol{f}_5$). These source or target ensemble groups vary across sub-circuits, meaning a single sub-circuit cannot capture the full repertoire of activity. This highlights the importance of CREIMBO's ability to disentangle the combined activity of multiple circuits. We also observe, but to a lesser degree, within-area interactions (e.g., black arrows in $\boldsymbol{f}_8$, $\boldsymbol{f}_{11}$), which emphasize that within-area amplification or regulation exist in addition to cross-region computations.

When exploring the activation patterns of the sub-circuits (Fig. 3D, 20), CREIMBO reveals cross-subject and cross-session variability. Notably, some sub-circuits are consistently used with high variability over time (e.g., $\boldsymbol{f}_8$, in light-green), while others, such as $\boldsymbol{f}_{12}$, are not utilized at all in any exemplary session (Fig. 3C, F). Moreover, when further exploring the sub-circuits that exhibit high activity variability across sessions (Fig. 3D,F), we observe that some are used within individual subjects in both tasks (e.g., $\boldsymbol{f}_2$ in subject 14, Fig. 3D), while others are used both across subjects and tasks (e.g., $\boldsymbol{f}_{11}$, used by subjects 10 and 14 in both tasks). Within-session pairwise correlations between sub-circuit coefficients (Fig. 10I) reveals low correlations ($<0.1$) for most circuit pairs, which indicates that distinct sub-circuits differ in activity and reflect different cognitive processes.

Since ground-truth components are unavailable in real data, direct validation of such components is impossible. Hence, we sought a proxy to evaluate CREIMBO's performance on real-data by testing its consistency under increasing observation noise levels. Particularly, we re-ran CREIMBO on the human recordings after adding increasing variance *i.i.d* Gaussian noise (with $K = 8, p_j = 6$, Fig. 23). CREIMBO remains robust to increasing noise, and experiences a phase transition at a specific noise level ($\frac{\sigma_{\text{noise}}}{\sigma_{\text{data}}} \sim 0.2$, Fig. 23A, B, E, F, H, I), which align with the dictionary-learning literature (Studer & Baraniuk, 2012). We found that the structure of correlations between within-area ensemble compositions (Fig. 23C, D) remains consistent under increasing noise levels, until around $\sigma_{\text{noise}} \approx 10$, after which these correlations weaken as noise increases. This may imply that high noise in the data can obscure meaningful relationships between ensembles. Additionally, CREIMBO identifies that the sub-circuit coefficients remain robust until $\sigma_{\text{noise}} \approx 1.8$, while exhibiting an increase in internal frequency as noise rises, potentially to account for noise that causes the observations to deviate from the system's dynamical rules (Fig. 23G, 25). This suggests that rapid transitions in CREIMBO's circuit coefficients may indicate noise that disrupts the system's dynamics.

**CREIMBO Discovers Regional Interactions Predictive of Task Variables**: We tested CREIMBO's ability to infer task-related variables from mice whole-brain Neuropixels multi-session data during a memory-guided movement task (data from Chen et al. (2024a), Fig. 26, 27). CREIMBO identifies intra- and inter-area brain interactions via the sub-circuits (Sec. F, Fig. 29C,D), including cross-regional flows into or from key areas associated with skills needed for the task, such as memory (the hippocampus, serves as a source in $\boldsymbol{f}_5$ and as a target in $\boldsymbol{f}_8$ ), planning (flows from frontal cortex in $\boldsymbol{f}_3$), and movement (flows from primary motor cortex in $\boldsymbol{f}_4$). Within-area sub-circuits (Fig. 29D) further show within-area ensemble interactions (e.g., the basal ganglia in $\boldsymbol{f}_1$ and the secondary motor cortex in $\boldsymbol{f}_2, \boldsymbol{f}_7$), along with self-activation/inhibition of ensembles in other sub-circuits (e.g., $\boldsymbol{f}_4$). Moreover, the sub-circuit coefficients ($\{\boldsymbol{c}_{kt}\}_{k=1}^K$) capture task-related patterns across trials and sessions (Fig. 4A,B). When training a regularized logistic regression model with the sub-circuit activations ($\{\boldsymbol{c}_{kt}\}_{k=1}^K$) as the only inputs, we were able to predict various task variables, including outcome, early lick, and lick side, well above chance levels (Fig. 4C-J, p-values $< 1 \times 10^{-10}$). Furthermore, sub-circuit activations across different task periods highlight how specific multi-regional interactions capture different aspects of the task. For instance, towards the trial end ($t_3$ window in Fig. 4), which includes the lick movement, $c_7$ shows an increased importance (Fig. 4E). Notably, $c_7$ corresponds to the activity of the $\boldsymbol{f}_7$ sub-circuit that captures flows to the secondary motor cortex (Fig. 29C). Another example is $c_8$ (activity of $\boldsymbol{f}_8$ that includes flows to hippocampus, Fig. 29C), which shows an increased feature importance in the first time window $t_0$. This early-in-task importance of this hippocampal-circuit aligns with the expectation that memory is required at the beginning of this memory-guided task. CREIMBO thus demonstrates the ability to capture interpretable neural interactions and to predict task variables that reveal complex regional interactions (Sec. F).

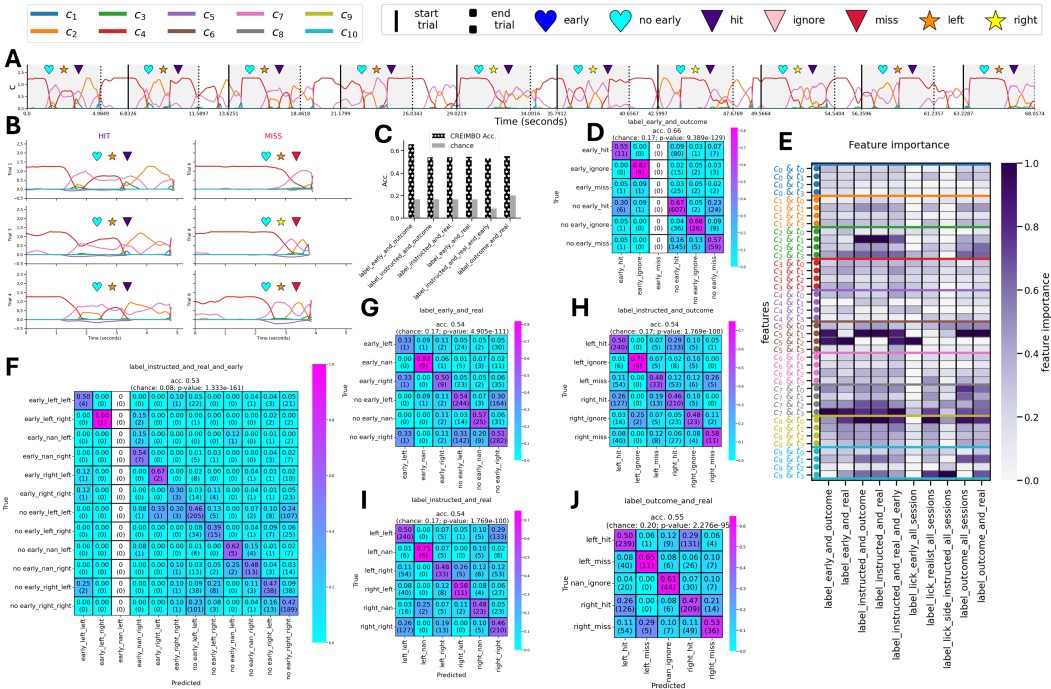

Figure 4: Task-variable prediction using CREIMBO's dynamic coefficients as input, based on coefficients from 19 sessions (40-60 trials each, Sec. F). **A:** Sub-circuit coefficients from an example session, shade marks within trials, bordered by starting and ending points. **B:** Sub-circuit coefficients for hit vs. miss trials. **C:** Accuracy score of predicting task variables based on coefficients vs. chance. **D, F,G,H,I,J:** Confusion matrices of predicting varying task-variables based on sub-circuit coefficients. **E:** importance of the different coefficients and time points for prediction.

## 6 DISCUSSION, LIMITATIONS, AND FUTURE WORK

Here, we introduced CREIMBO—a novel approach for uncovering multi-regional dynamics in neural data collected across multiple sessions. CREIMBO addresses the challenge of integrating non-simultaneous neural recordings by joint dynamical inference and sparsity regularization to capture the underlying neural sub-circuits governing brain activity. We further demonstrated the efficacy of CREIMBO through multiple synthetic and neural data, and found that CREIMBO recovered ground truth components and is robust to noise in identifying cross-regional motifs that span cross-session interactions. CREIMBO offers several advantages over existing methods. Chiefly, it identifies a cross-session shared latent space where non-stationary ensemble interactions are governed by a time-varying decomposition of universal basis dynamics. By structuring these dynamics in terms of global sub-circuits, CREIMBO allows the discovery of sub-circuits meaningful for various cognitive processes, enabling the identification of variability in neural activity across subjects and sessions.

An important feature of CREIMBO is its ability to unify sessions with different neuron subsets through the universal dictionary of dynamical interactions prior that aligns ensembles in terms of functionality. This can enable the robust inference of ensemble activities even if different neurons are observed across sessions. This ability, however, depends on the extent to which the same sub-circuits are used across sessions, the distinctiveness of different $\{f_k\}$s, and the premise that sub-circuits containing interactions from or to ensembles that are entirely absent in certain sessions do also include ensembles that are observed in those same sessions (Sec. D). Another limitation is CREIMBO's reliance on linear projections from the latent space to per-session observation space, restricting flexibility. Extending to non-linear projections offers potential for development, though it introduces computational and interpretability challenges. Finally, CREIMBO uses dictionary learning, which is computationally demanding, and future iterations will include parallel processing. Extending CREIMBO to additional applications, including non-neural data (e.g., immune-cell counts), is an exciting future step. Moreover, integrating CREIMBO with mDLAG (Gokcen et al., 2024), with the latter identifying optimal communication delays and the former leveraging these to further identify the underlying set of co-active LDSs, presents an exciting future work.

## Acknowledgments

N.M. was supported by the Kavli NeuroData Discovery Award of the Kavli Foundation, and as a Kavli Fellow by the Kavli Neuroscience Discovery Institute at Johns Hopkins University. R.L, O.R, and N.M. were supported by the National Institute of Neurological Disorders and Stroke under Award Number 5U24NS120057 (PI: O. Ruebel). A.S.C was supported by NSF CAREER Award 2340338 and a Johns Hopkins Bridge Grant.

## Code and Data Sharing

The human neural data used in this study Kyzar et al. (2024b;a) is publicly available via DANDI Archive at `https://dandiarchive.org/dandiset/000469/0.240123.1806`. The mice neural data (Chen et al., 2024a; 2023) is publicly available via DANDI Archive at `https://dandiarchive.org/dandiset/000363?search=mesoscale&pos=2`. The code is available on GitHub at this link.

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

# Appendix

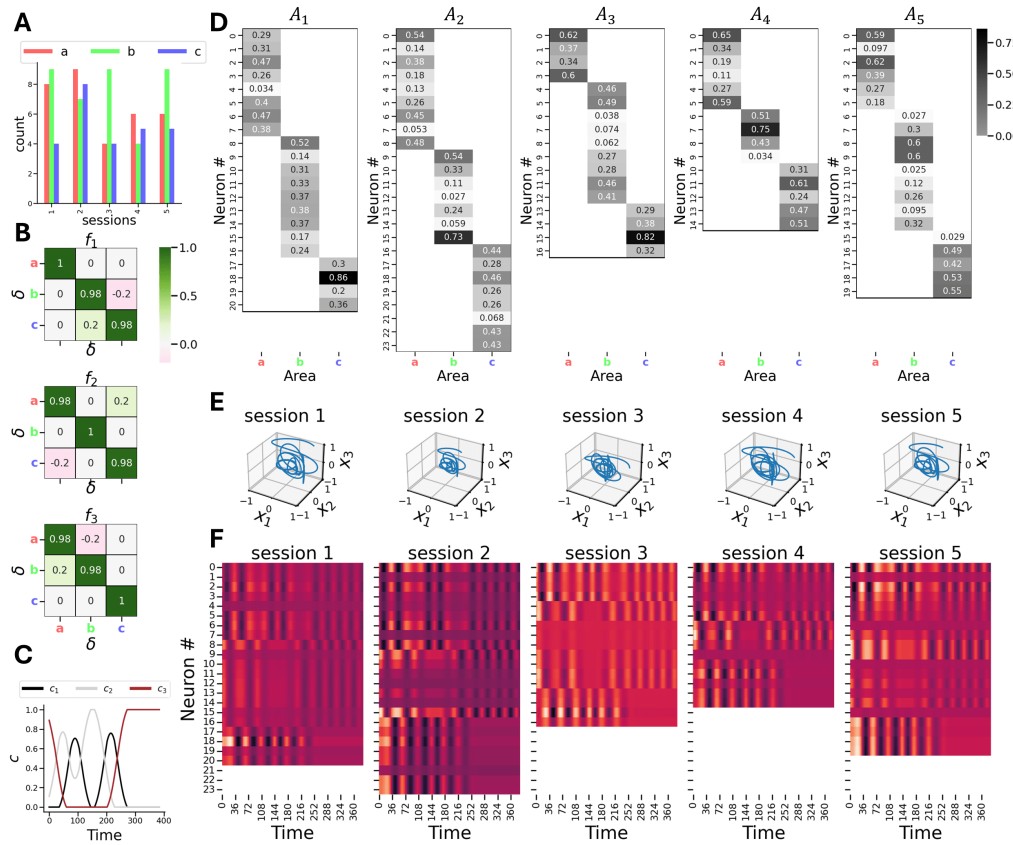

Figure 5: **A:** Distribution of neurons across areas (areas $a$, $b$, and $c$) over the $5$ different synthetic sessions. **B:** Ground truth sub-circuits are $K = 3$ rotational matrices, each captures a different rotational direction. **C:** Time-coefficients of synthetic sub-circuits. **D:** Synthetic ensembles $(\{\boldsymbol{A}^d\}_{d=1}^5$ **E:** Latent (ensemble) dynamics. Different trajectories in the latent space emerge as distinct cross-session time-changing decomposition of the ensemble sub-circuits. **F:** Synthetic Ground Truth observations $(\{\boldsymbol{Y}_d\})$

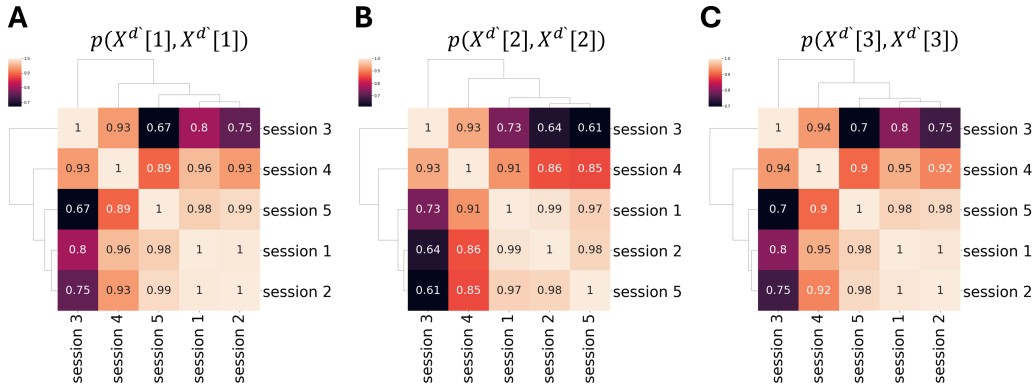

Figure 6: Clustermaps of correlations of cross-session latent dynamics. Each subplot captures one dimension of the latent dynamics.

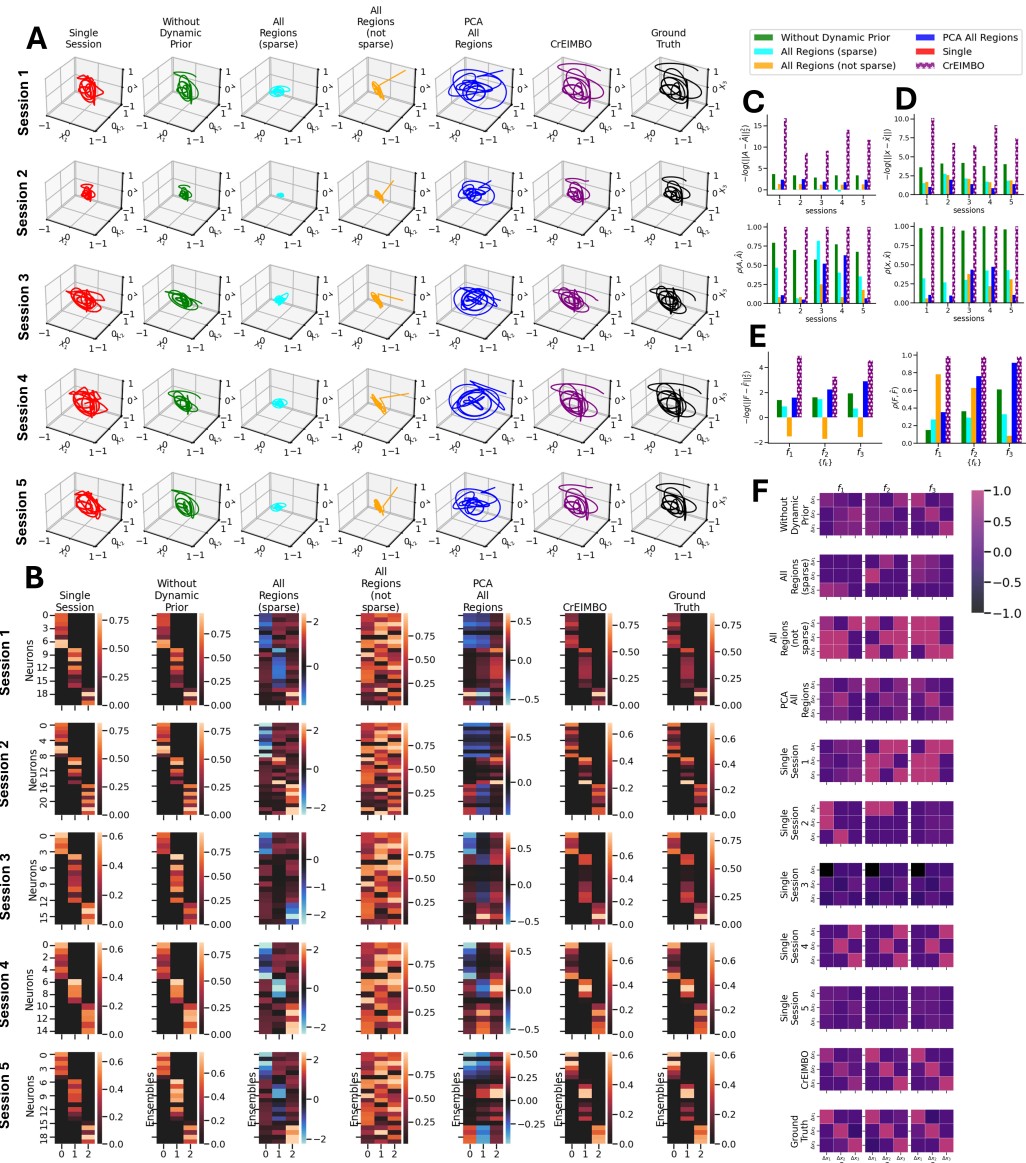

Figure 7: Comparison between CREIMBO to the baselines reveal that CREIMBO outperforms other approaches (see baselines details in Sec. D). **A:** Learned latent dynamics ($\boldsymbol{X}$) for each of the 5 sessions (rows) for the different methods (columns). **B:** Learnt ensemble compositions ($\boldsymbol{A}$) for each of the 5 sessions (rows) for the different methods (columns). **C-E:** $-log(MSE)$ and correlation between the ground truth and the the identified ensemble compositions (in **C**), latent dynamics (in **D**), and sub-circuits (in **E**). **F:** Heatmaps of the identified sub-circuits for the different methods.

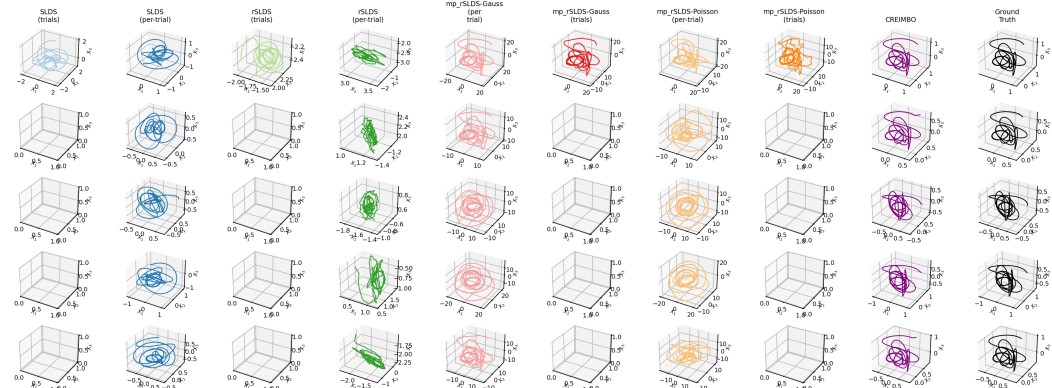

Figure 8: Further comparisons of the learned latent dynamics to these learned by SLDS, rSLDS, and multi-region rSLDS, with their variations (see Sec. D for details). Different columns represent different methods. Different rows capture different sessions.

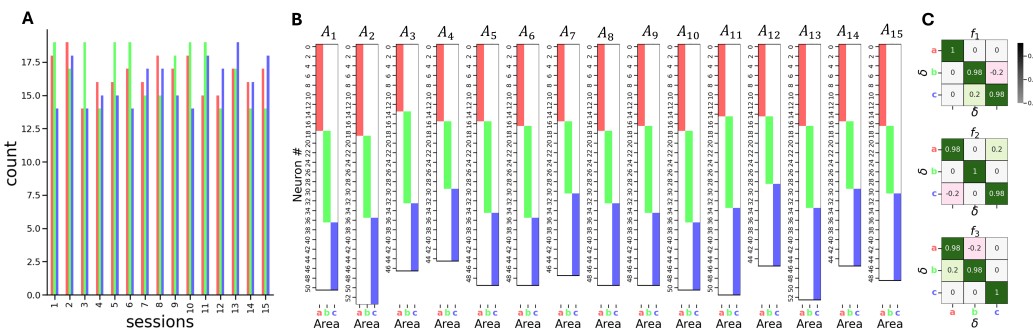

Figure 9: Distribution of areas in second synthetic dataset. **A:** Histogram of neural counts per area and session. **B:** Ensemble masks ($A_{\mathrm{mask}}$) across sessions. **C:** Ground truth sub-circuits.

---

**Algorithm 1** CREIMBO training

---

1: **Input:** $p, K, \sigma_h, \beta_1, \beta_2, \beta_3, \lambda_x, \lambda_c, \lambda_f, \lambda_\rho, \lambda_{\mathrm{obs}},$ Batch Size
2: **Initialize:** $\{f_k\}_{k=1}^K, \{c_d\}_{d=1}^D, \{x_d\}_{d=1}^D$
3: **Pre-calculate:** $\{h_j\}_{j=1}^J$
4: **repeat**
5:     **for** each session $d$ in a random batch of $D$ **do**
6:         Update ensembles
7:         Update hidden dynamics and coefficients
8:     Select a random batch of sessions from $D$
9:     Update networks $\{f_k\}$
10:    **if** stuck in local minimum **then**
11:        Perturb $\{f_k, c_d\}$
12: **until** convergence

---

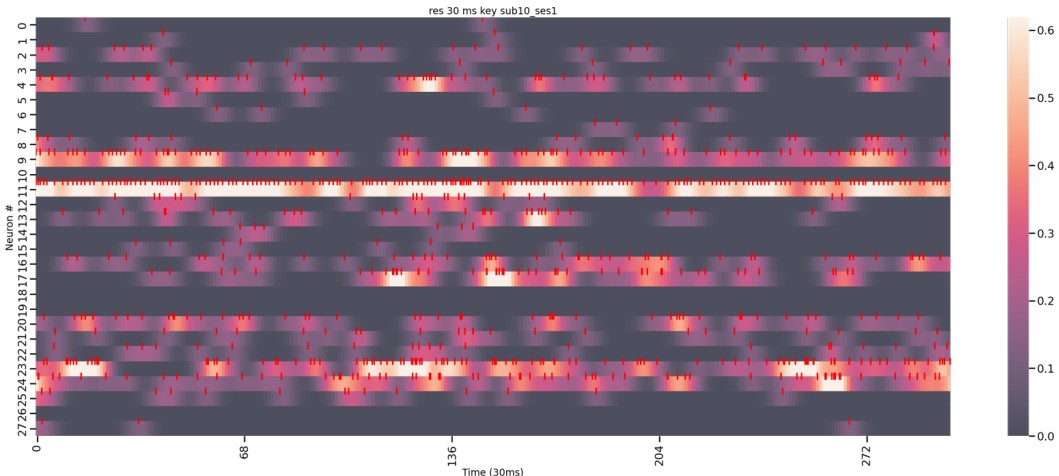

Figure 10: Rate estimation example for a specific session using a 30 ms Gaussian kernel.

## A  NOTATIONS

| Symbol | Description |
|--------|-------------|
| $D$ | Number of sessions (encompassing both same and different subjects) |
| $M_d$ | Number of trials for each session $d$ |
| $\boldsymbol{Y}_m^d$ | Neural recordings firing rate estimation for trial $m$ and session $d$ |
| $N_m^d$ | Number of neurons observed in trial $m$ of session $d$ |
| $\mathrm{T}_m^d$ | Number of time points observed in trial $m$ of session $d$ |
| $\{\boldsymbol{Y}_m^d\}_{m,d}$ | Overall set of observations |
| $\boldsymbol{A}_m^d$ | The neuronal composition of each sparse ensemble in trial $m$ and session $d$ |
| $\boldsymbol{A}_{\text{block}}$ | Block of $\boldsymbol{A}$ representing neural dynamics in a specific brain area |
| $n_j$ | Total neurons in area $j$ |
| $p_j$ | Maximum number of ensembles discernible within area $j$ |
| $p$ | Total number of ensembles across all areas |
| $J$ | Total number of distinct brain areas |
| $n$ | Total neurons involved across all trials or sessions |
| $\boldsymbol{x}_t$ | State vector at time $t$ |
| $\boldsymbol{F}_t$ | Transition matrix at time $t$, representing ensemble interactions |
| $K$ | Number of global interacting units |
| $\boldsymbol{c}_{kt}$ | Influence of the $k$-th global interaction at time $t$ |
| $\boldsymbol{f}_k$ | Basic linear system representing the $k$-th global interaction |
| $\tau = \mathrm{T}_m^d$ | Duration of the trial or session |

## B  SYNTHETIC DATA EXPERIMENT DETAILS

We simulated the synthetic data to represent the non-stationary dynamics of brain activity including multi-region interactions. Current assumptions about brain activity suggest Zeki (2015); Nelson & Bower (1990) it functions through distributed, parallel processing across multiple regions, potentially involving co-active circuits that can process a range of variables (e.g., task demands, feedback signals, and sensory information), simultaneously. This biological reality thus motivated our data generation approach, to reflects the multi-process nature of neural activity, where different processes can interact and contribute to brain dynamics. We simulated $D = 5$ sessions, each sharing a common set of basis rotational dynamics, denoted as $\{\boldsymbol{f}_k\}_{k=1}^{K=3}$, with $p_j = 3$ regions of interest. Different regions were associated with different ensembles, and the dynamics for each session were modeled using sparse decomposition of the basis dynamics. For each session, the dynamics' coefficients, $\boldsymbol{c}_t^k$, were generated over $T = 500$ time points. These coefficients were generated by initially giving them binary values (active vs. inactive states), with random switches occurring every $x$ time points, where

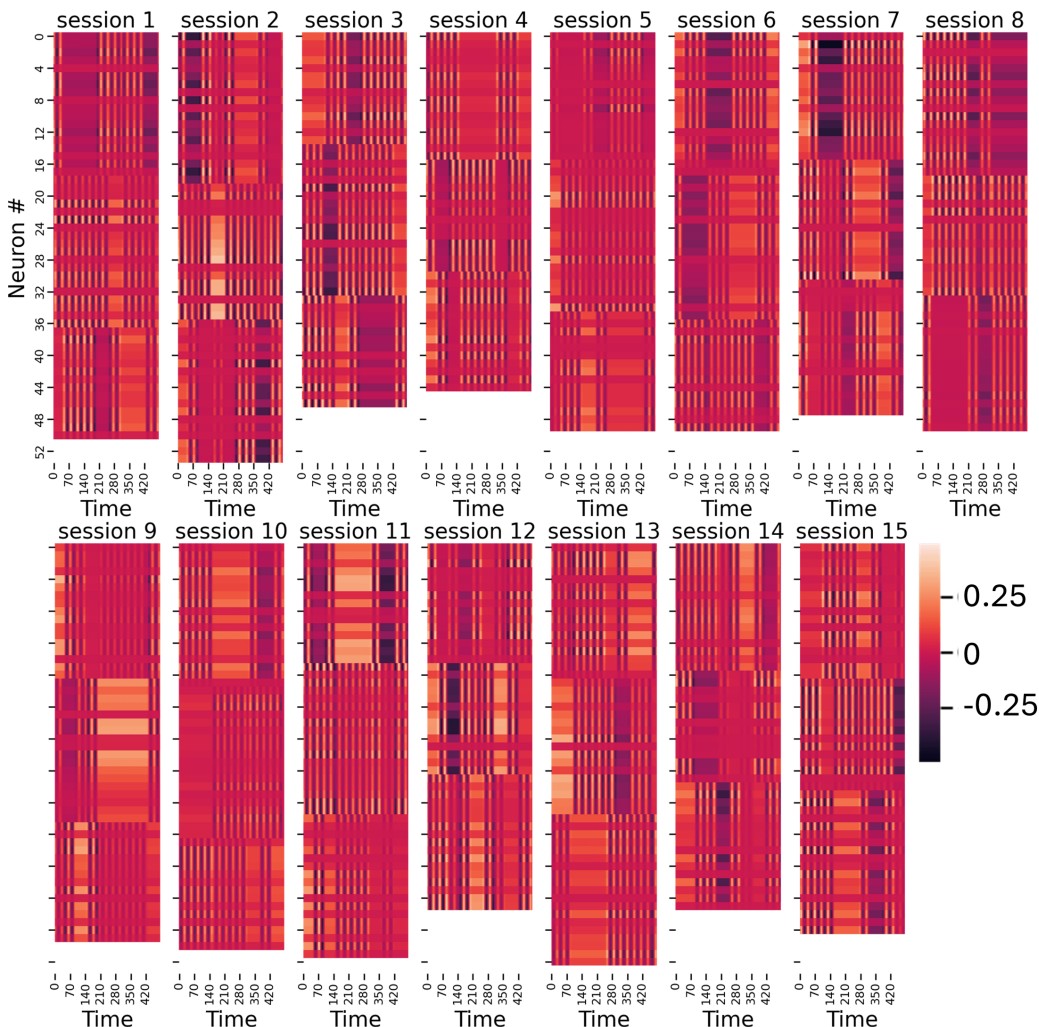

Figure 11: Data and results for the second synthetic experiment (overall $D = 15$ sessions, $J = 3$ regions and $p_j = 1$ ensembles for each region $j = 1 \dots J$). **A:** Ground truth synthetic observations. **C:** Ground truth ensemble

$x$ was uniformly drawn between 10 and 30. The sparsity constraint was maintained at each time point, ensuring that only one dynamic was active at any time ($\|\boldsymbol{c}_t^k\|_0 = 1 \forall t$).

Then, to introduce smooth transitions between switching states, a Gaussian convolution with a time window of 6 points (standard deviation = 0.7) was applied to the coefficients, to promote a smoother / more gradual changes between active states. Different sessions maintained the same sequence of active dynamics but allowed for varying durations of each state, introducing temporal variability across sessions.

Each session $d$ was augmented with a unique projection matrix $\boldsymbol{A}^d$ that captures the mapping between the latent space and the observations (neural) space in the respective regions. These projection matrices $\boldsymbol{A}^k$ consisted of $P = 3$ columns, each corresponding to one of the three regions. The number of neurons in each region was randomly chosen between 4 and 9, with the count varying independently across regions and sessions. This variability ensured that different sessions were not identical in their neural configurations. Latent dynamics for each session were generated using the recursive equation $\boldsymbol{x}_t = \sum_{k=1}^{K} \boldsymbol{c}_{kt} \boldsymbol{f}_k \boldsymbol{x}_{t-1}$, starting from an initial state $\boldsymbol{x}_0 = [1, -1, 1]^\top$. CREIMBO was trained based on the observations only, which were defined for each session as $\boldsymbol{y}^d = \boldsymbol{A}^d \boldsymbol{x}^d + \epsilon$ where $\epsilon$ is an addition of *i.i.d* Gaussian noise with standard-deviation of $\sigma_{\text{noise}} = 0.1$

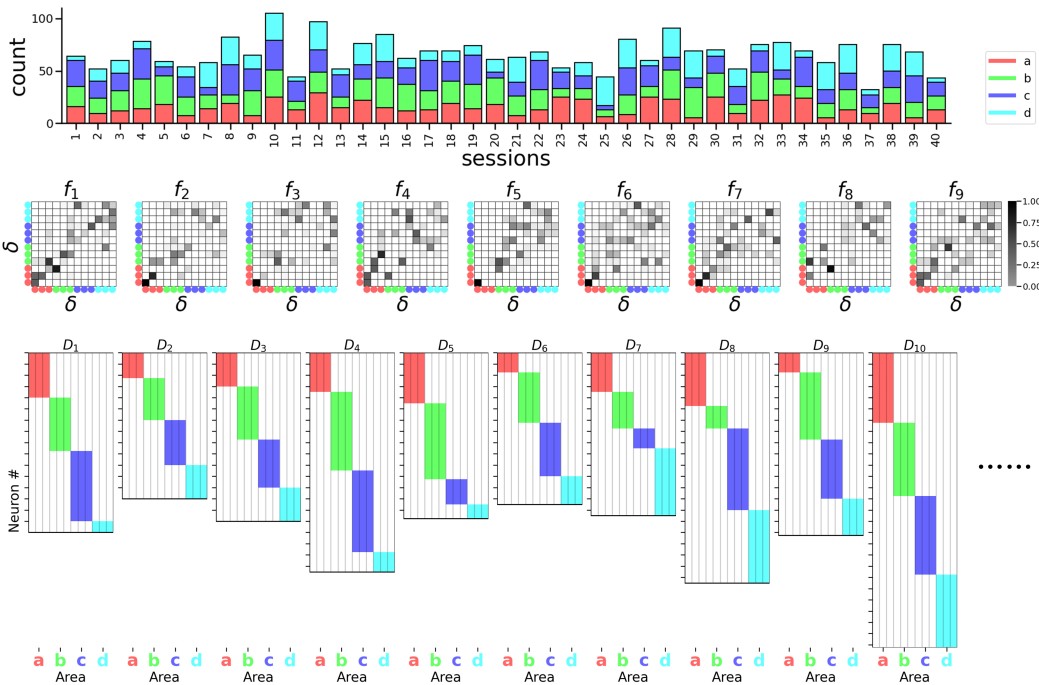

Figure 12: Multi-ensemble synthetic data. Top: Distribution of areas across sessions. Middle: Ground truth sub-circuits. Bottom: Masks for ensembles identification.

The full set of parameters for the synthetic experiment are available in Table 1.

## C  PARAMETERS FOR HUMAN MULTI-REGIONAL EXPERIMENT

The full set of parameters for the human data experiment are available in Table 2.

## D  INFORMATION ABOUT THE BASELINES:

We distinguish between two types of baselines: (1) ablation experiments and (2) comparison to the performance of entirely different methods. It is important to note that the latter cannot fully capture the unique capabilities of CREIMBO, particularly in identifying basis ensemble interactions shared across sessions through sparse decomposition. Therefore, we only compare certain aspects that can be extracted from other methods (e.g., multi-regional SLDS, which requires the same number of neurons per session). However, these comparisons do not imply that other models outperform or underperform CREIMBO; they simply offer different insights, with varying strengths and limitations depending on the application.

Importantly, since the sub-circuits $\{f_k\}$ is invariant to its ordering, we used SciPy's implementation of the 'linear_sum_assignment' problem Crouse (2016) to re-order the identified sub-circuits before comparing them with the ground truth, matching the $f$s values by minimizing the $\ell_2$ error. **Ablation Experiments:**

1. "**Without Dynamic Prior**": Running CREIMBO while removing the prior over the dynamics from the inference. Instead, inferring the temporal traces of the ensembles using regularized least-squares with the addition of smoothness, de-correlation, and Frobenius norm regularization terms on the dynamics.

2. "**All Regions (sparse)**": Running CREIMBO without applying the multi-regional diagonal mask over the ensemble compositions ($A$), thus supporting the finding of non-localized ensembles.

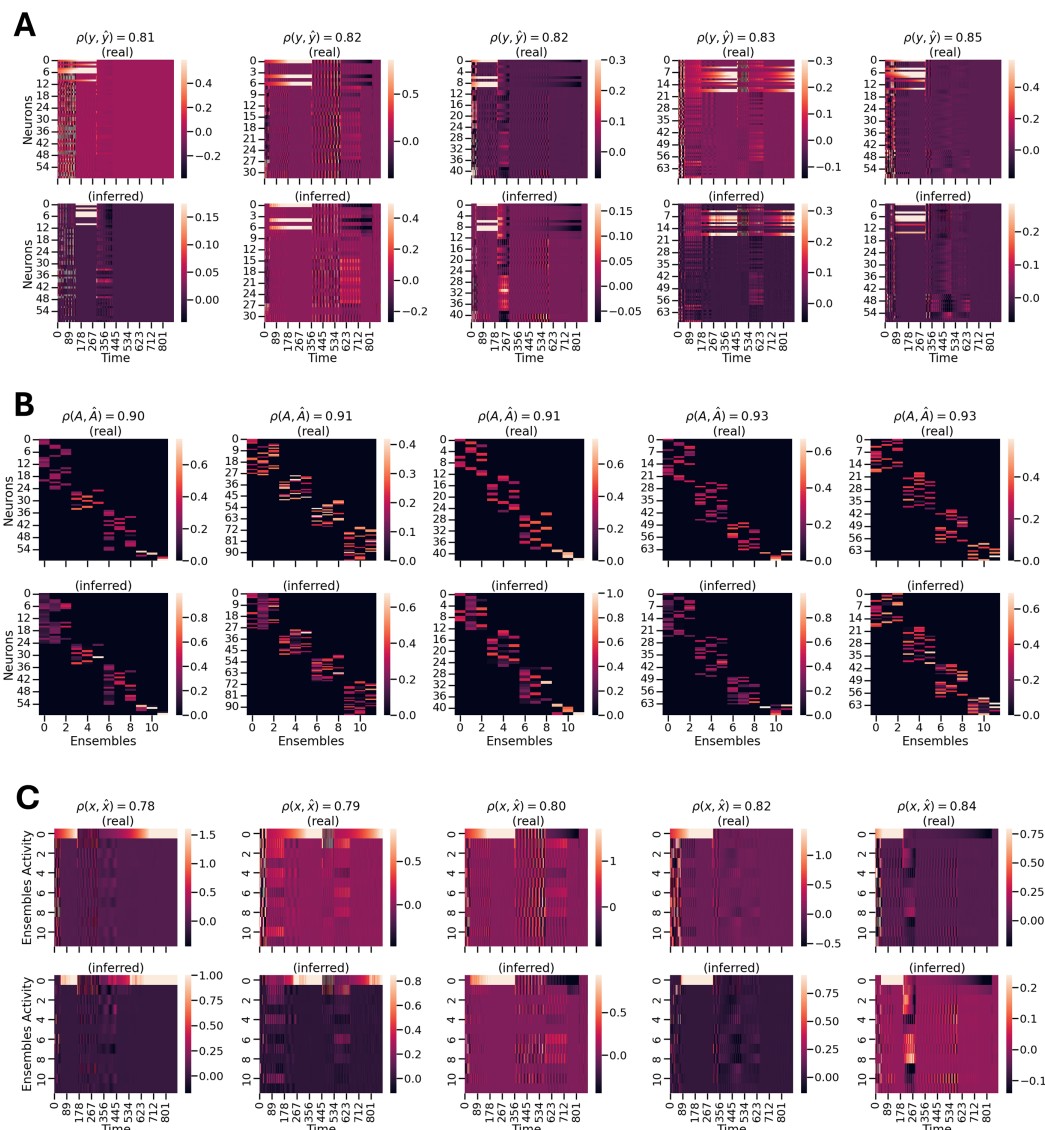

Figure 13: Multi-ensemble synthetic data results. Top: Recovering the observations $Y$ (five exemplary sessions). Middle: Identified Ensembles (five exemplary sessions). Bottom: Identified latent dynamics (five exemplary sessions).

3. "**All Regions (non-sparse)**": Similar to the previous one but without applying sparsity on the ensemble matrix.

4. "**PCA All Regions**": Running CREIMBO while replacing the dimensionality reduction step with PCA.

5. "**Single Session #**": Running CREIMBO on a single session (view) of the data rather than leveraging cross-session information.

**Other methods:**

1. "**SLDS**": We used the SSM Python package described by Linderman et al. at Linderman et al. (2020). For the (non-recurrent) SLDS option, we used the "gaussian_orthog" emission parameter, the "bbvi" as the fitting method, "variational_posterior" was set to "mf", and the number of iterations ("num_iters") was set to 500. The rest of the fitting parameters were left as default. We were inspired by

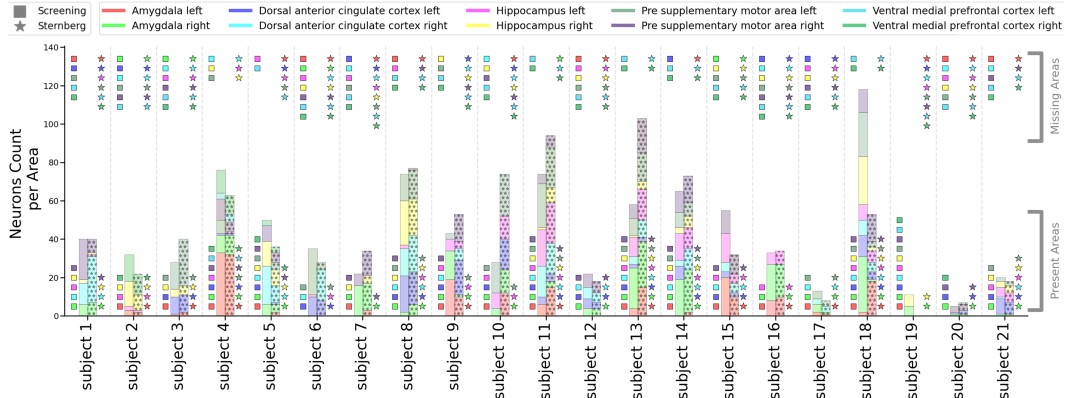

Figure 14: Areas distribution in real world human neural data.

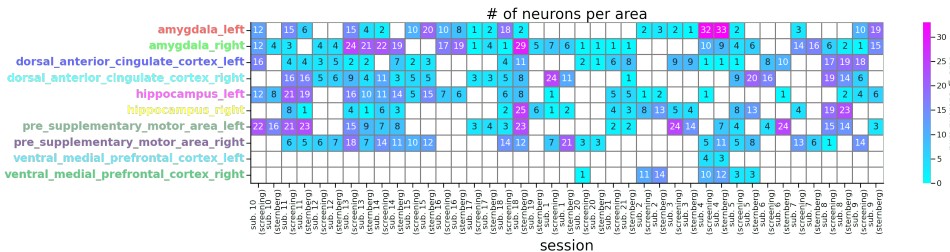

Figure 15: Number of neurons per-area and per-session in the real-world human data taken from Kyzar et al. (2024b).

this notebook `https://github.com/lindermanlab/ssm/blob/master/notebooks/3-Switching-Linear-Dynamical-System.ipynb` by Linderman et al. (2020) for our comparison. This includes:

1) "**SLDS (trials)**" which captures the training of SLDS with "bbvi" fitting method from across all trials (by stacking trials information vertically),

2) "**SLDS (per-trial)**" which refers to training SLDS individually per-trial.

2. "**rSLDS**": Similarly to SLDS, for the recurrent version (rSLDS, (Linderman et al., 2016)), we used the same SSM package provided by Linderman et al. (2020). Here, we used the recurrent option for the transitions (i.e., the "transitions" parameter was set to "recurrent_only"), using "diagonal_gaussian" dynamics and "gaussian_orthog" emissions. This includes:

3) "**rSLDS (trials)**" which captures training rSLDS with the "bbvi" fitting method from across all trials (by stacking trials information vertically),

4) "**rSLDS (per-trial)**" which refers to training rSLDS individually per-trial.

3. "**mp_rSLDS**": Multi-Regional rSLDS, as described in Glaser et al. (2020). To run our comparison, we were inspired by the colab-notebook for this "mp_rslds" method, provided by the SSM Python Package Linderman et al. (2020) (at `https://colab.research.google.com/github/lindermanlab/ssm/blob/master/notebooks/Multi-Population-rSLDS.ipynb`). This includes:

5) "**mp_rSLDS-Gauss (trials)**" which captures training "mp_rSLDS" from across all trials (by stacking trials information vertically) under Gaussian statistics.

6) "**mp_rSLDS-Gauss (per-trial)**" which refers to training "mp_rSLDS" individually per-trial under Gaussian statistics.

7) "**mp_rSLDS-Poisson (trials)**" which captures training "mp_rSLDS" with the "bbvi" fitting method from across all trials (by stacking trials information vertically) under Poisson statistics.

8) "**mp_rSLDS-Poisson (per trial)**" which refers to training "mp_rSLDS" individually per-trial under Poisson statistics.

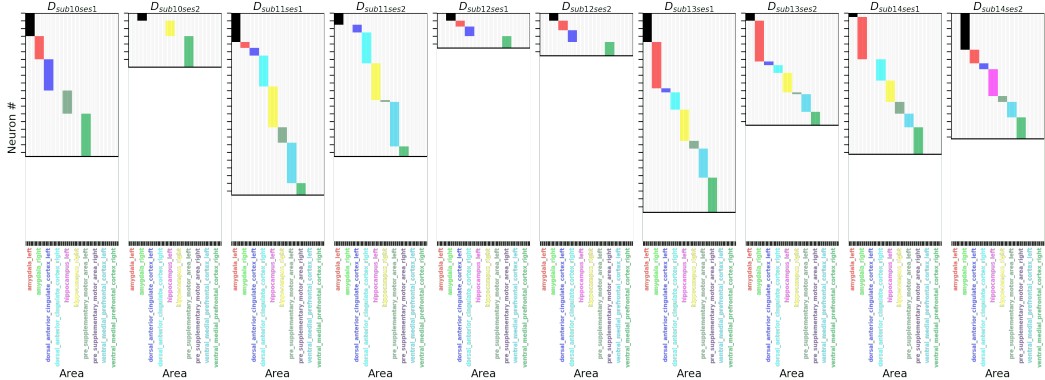

Figure 16: Multi-regional Masks for real-world human data.

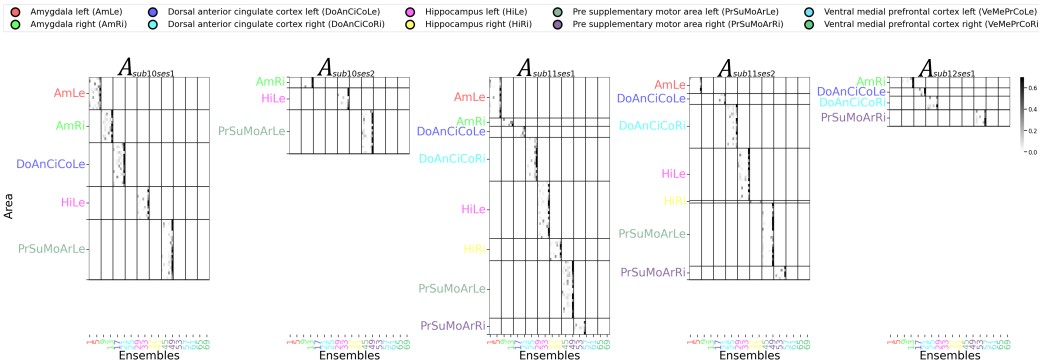

Figure 17: Five Exemplary Ensemble Matrices of the Real World Data, as Identified by CREIMBO

## ASSUMPTIONS

The CREIMBO model is based on a core set of assumptions over the nature of the data. These assumptions draw on both known properties of neural processes and general well-known statistical models used widely in data science. We categorize these assumptions into two types:

1. Priors over the underlying neural processes.
2. Priors over Observational Constraints in order for CREIMBO to properly infer the underlying system by leveraging multi-session information.

### D.1 PRIORS OVER THE UNDERLYING NEURAL PROCESSES

1. The neural dynamics in each session $d$ are assumed to lie on a low-dimensional manifold, embedded in a $P << N^d$ low-dimensional space that is defined by $P$ functional groups of neurons (referred to as "ensembles").

2. We assume that these functional ensembles consist of neurons with co-activation patterns, where neurons can belong to more than one, but only *a few* (a sparse number) of ensembles, with varying degrees of membership.

3. We further propose that the interactions between these ensembles drive the evolution of the latent manifold over time and are key to encoding changes in conditions and behavior.

4. For CREIMBO to be effective, we assume these interactions arise from the joint synchronous activity of multiple co-occurring processes, captured by a limited-size set of "basis-ensemble-interactions". The time-varying, sparse decomposition of these basis-ensemble-interactions, weighted by their time-local contributions in every time point, can adequately describe the manifold's evolution over time.

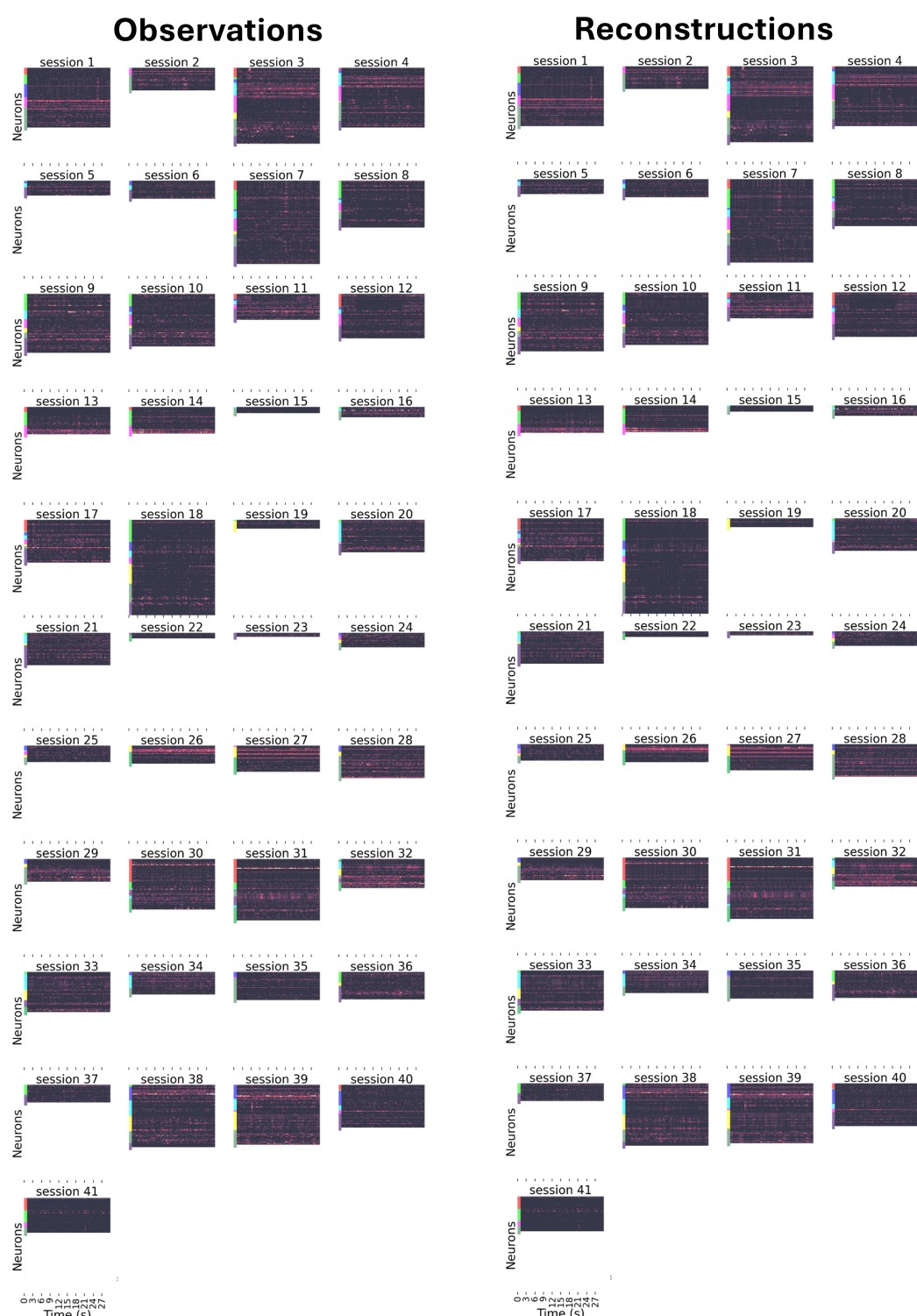

Figure 18: Observations vs. CREIMBO's reconstructions for the real-world human data experiment.

5. We assume that different dynamics basis elements capture distinct processes or behaviors. Some processes are required to be globally related to the cognitive task, while others capture session- or subject-specific processes (see Sec. D.2).

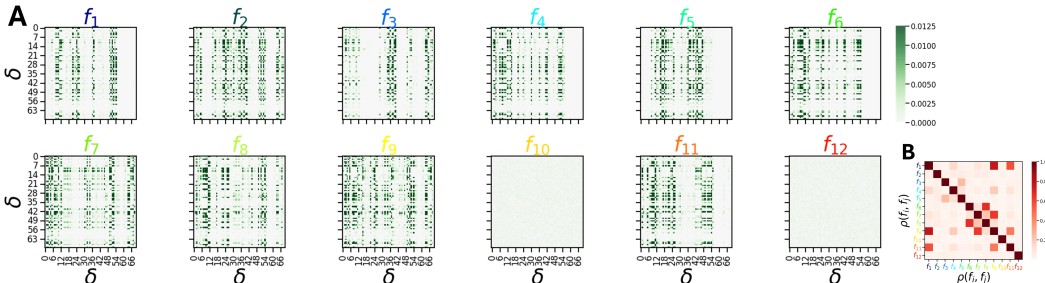

Figure 19: The sub-circuits underlying human data as identified by CREIMBO. **A**: The identified 12 sub-circuits. **B**: Pairwise correlations between the identified sub-circuits.

Table 1: Parameter values for synthetic experiments

| Parameter | Value (range) | Explanation |
| --- | --- | --- |
| same_c | False | Indicates if to use shared sub-circuits coefficients across sessions. |
| step_f | 0.1 - 0.5 | Gradient descent step size for updating $f$ |
| GD_decay | 0.99 - 0.999 | Decay over iterations of gradient-descent step size. |
| max_error | 1e-09 | The error threshold to stop training the model. |
| max_iter | 500 | Threshold on the maximum number of iterations. |
| include_D | True | Indicates the inclusion of a projection to a latent space. |
| step_D | $10^{-5} - 10^{-2}$ | Range of Gradient Descent Step size for $A$. |
| seed | 0 - 4000 | Random seed to choose from |
| normalize_eig | True , False | Indicates if to normalize the sub-circuits by |
| start_sparse_c | False , True | Whether to initialize $c$ to be sparse. |
| sparse_f | True | Whether to apply sparsity on the sub-circuits |
| num_gradient_steps | 1 -3 | Number of gradient steps in an iteration. |
| add_avg | True , False | Whether to add a moving average to an iteration. |
| sparsity_on_f_max | 40-60 | Percentile of sparsity applied on each $f$ |
| take_multiple_gd | False, True | Indicates if to take multiple steps of gradient descent. |
| D_graph_driven | True | Indicates that D is inferred with graph-driven way. |
| infer_x_c_together | False , True | Indicates if the inference of $x$ and $c$ occurred simultaneously. |
| include_mask | True | Indicates the inclusion of a mask on $A$ |
| norm_D_cols | True | Indicates the normalization of columns of $A$ |
| lambda_D | 0.1 - 0.3 | Regularization weight on updating $A$ |
| step_D_decay | 0.99 - 0.9999 | Decay rate for updating $A$ parameters |
| num_regions | 3 | Number of brain regions. |
| lambda_x | 0 - 0.1 | Sparsity term on the latent dynamics. |
| latent_dim | 3 | Latent dimension. |
| noise_level | 0.2-0.6 | Standard deviation of noise added to the data. |
| num_subdyns | 3 | Number of sub-circuits. |
| lasso_solver | 'spgl1' | Pylops solver used for $\ell_1$ regularization. |

6. Each of these ensemble interactions may capture between-area interactions, within-area interactions, or both.

## D.2 PRIORS OVER OBSERVATIONAL CONSTRAINTS

To effectively learn a unified representation by leveraging information across sessions, we make the following assumptions on the statistics of the data.

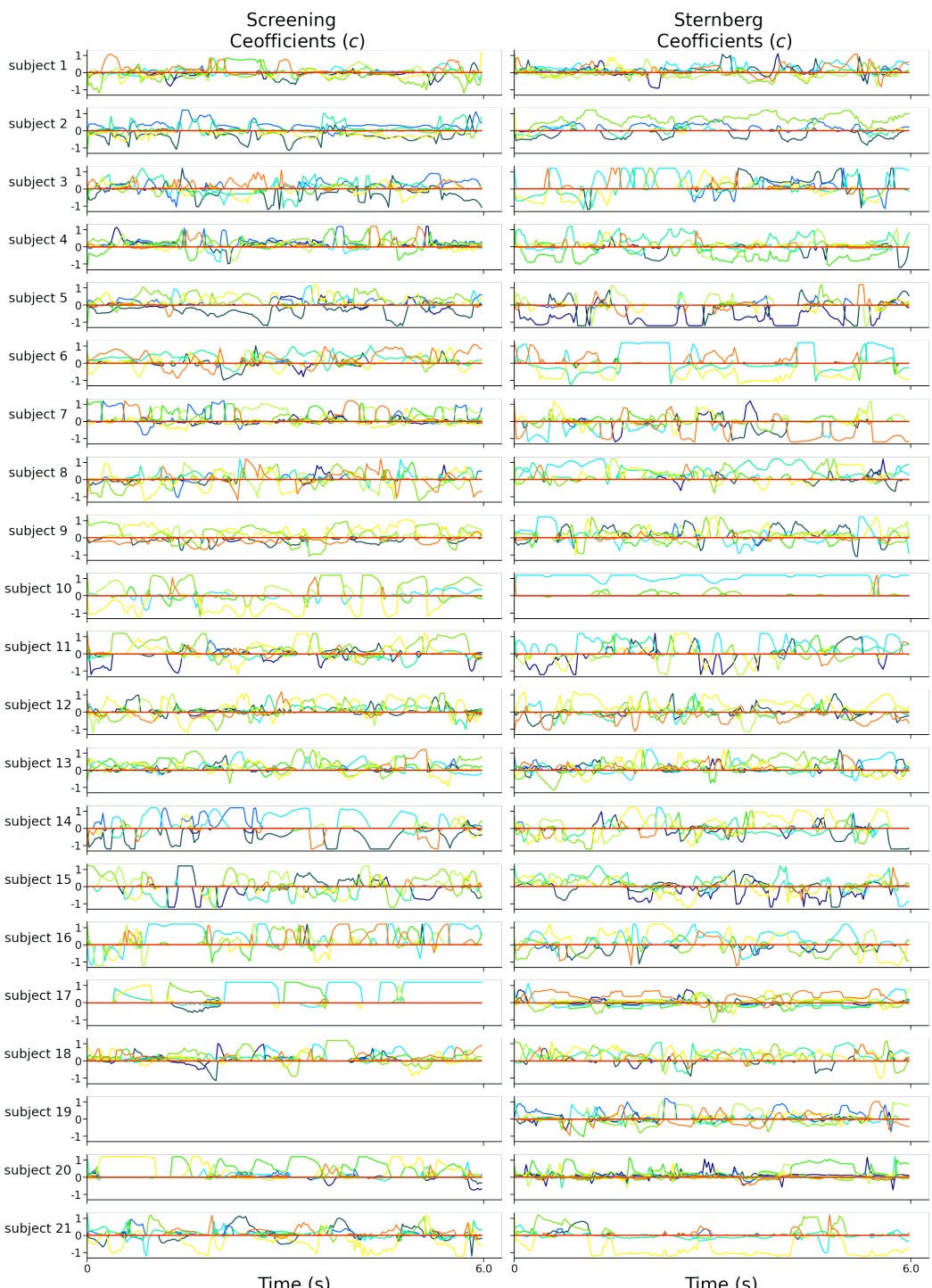

Figure 20: Sub-circuits Coefficients separated by task type ("Screening" vs "Sternberg") and subject.

1. Our ability to infer the underlying latent state (see Fig. 1B) is contingent upon the overall observability of the dynamical system. In traditional linear systems, the observability matrix can guarantee that any state is visible in a finite time horizon output. Non-stationary systems, as in CREIMBO, do not have succinct guarantees, the closest of which come in the form of observability conditions on switched linear systems (Tanwani et al., 2012).

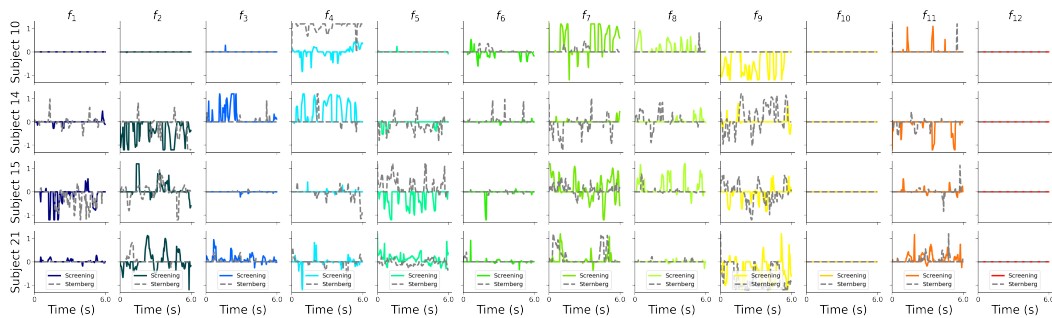

Figure 21: Comparing the use of the same coefficient under 'Sternberg' vs 'Screening' within four exemplary subject, reveals usage of similar sub-circuits within subjects.

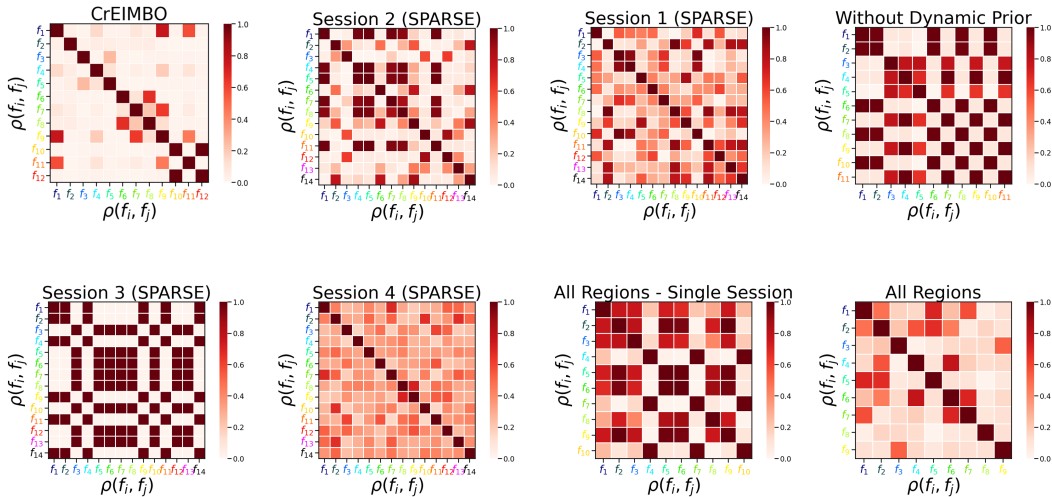

Figure 22: Correlations between the brain sub-circuits identified by CREIMBO compared to the other methods. These results highlight the CREIMBO discovers more distinct sub-circuits than these idetified by the other methods.

As the heart of the observability condition is that the dynamics "well mixes" the state such that any state element in the null space of the readout matrix will eventually be rotated into the its span, and thus visible. We thus assume that the spectral radius of each dynamical system (a measure of mixing in linear systems (Simchowitz et al., 2018)) in the basis is close to one. This assumption is loose, due to the lack of theoretical guarantees on general non-stationary systems, and further analysis should identify tighter, and less stringent, assumptions.

2. Another prior we want to consider is which $f$s capture ensembles unobserved in certain sessions, allowing us to assert that CREIMBO can recover the activity of these unobserved ensembles during those sessions. To achieve this, we must ensure that in the sessions where these ensembles are unobserved, at least some $f$s representing them also capture the activity of observed ensembles. For example, if certain missing ensembles are represented solely by $f$s that do not capture any observed ensembles, we will not be able to infer the activity of these unobserved ensembles in the sessions where they are absent, because there are no observed ensembles linked through the dynamic prior.

3. To ensure uniqueness of the sparse decomposition of the dynamical systems model at each session and time point, we assume that the effective spark of the dynamics basis is large ($S^* \geq 2S$). Essentially, there is no $f_k$ that can be linearly composed of $2S^*$ other dynamical systems. This ensures that there cannot exist two (or more) equivalent

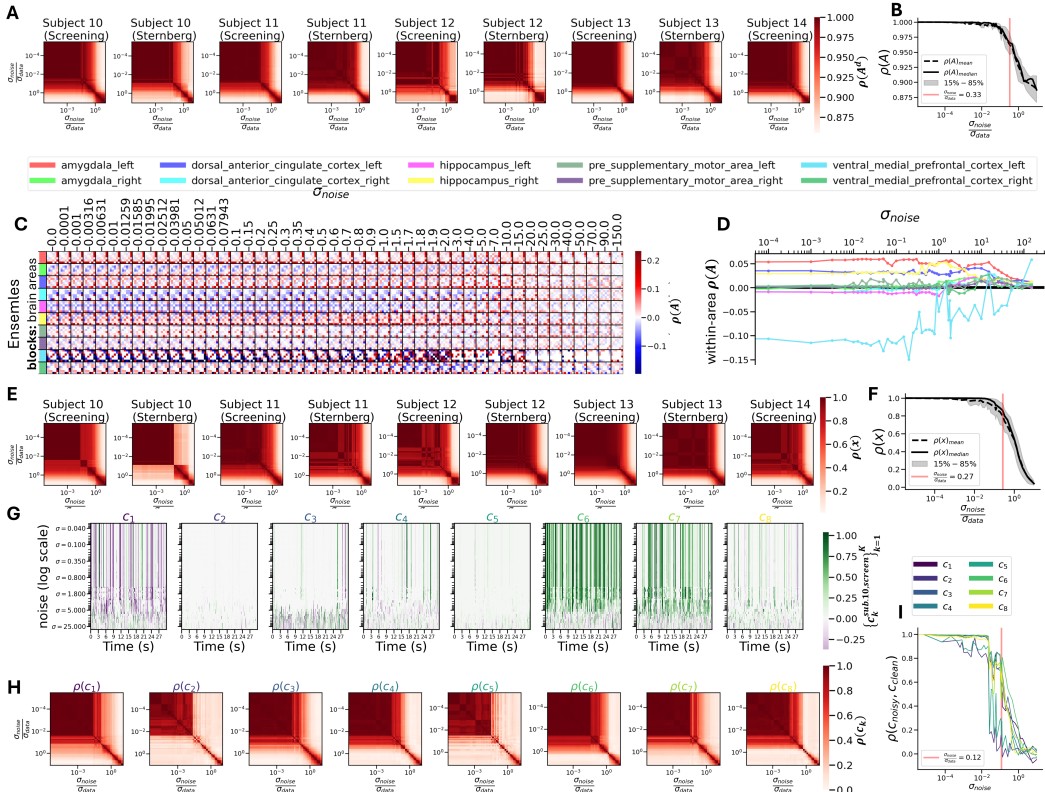

Figure 23: **CREIMBO's Robustness to Increasing Random Normal Noise Levels. A & B:** Correlations of identified ensemble compositions ($A^d$) under increasing noise, for individual sessions (**A**) and all sessions combined (**B**). Robustness decreases only when $\frac{\sigma_{\text{noise}}}{\sigma_{\text{data}}} = \frac{1}{3}$. **C & D:** Correlations of ensemble compositions (from data concatenated across all conditions) displayed in a heatmap, with areas represented by rows of blocks (vertical) and noise levels by columns (horizontal). Block $(i,j)$ captures the correlation of ensembles from area $i$ at the $j$-th noise level. A sharp drop in within-area correlations (**D**) occurs at $\sigma_{\text{noise}} = 1$ ($\frac{\sigma_{\text{noise}}}{\sigma_{\text{data}}} \approx \frac{2}{3}$). **E & F:** Correlations of ensembles' trajectories ($\{x^d\}_{d=1}^D$) for each condition (**E**) or combined (mean/median) (**F**) show robustness until $\frac{\sigma_{\text{noise}}}{\sigma_{\text{data}}} = 0.27$. **G:** Time-varying interaction coefficients for the 1st condition "(sub. 10, Screening)" under increasing noise reveal similar but more frequent changes, with a sharp frequency change at $\sigma_{\text{noise}} = 5$ ($\frac{\sigma_{\text{noise}}}{\sigma_{\text{data}}} = 0.33$). **H & I:** Correlations between corresponding $\{c_k\}_{k=1}^K$ "(sub. 10, Screening)" over increasing noise (**H**), and compared to the coefficients identified from the original data (**H**).

decompositions $\sum_{k=1}^K f_k c_{kt}$ and $\sum_{k=1}^K f_k c_{kt}^*$ for distinct coefficients that both compose the same dynamical matrix.

4. Lastly, to ensure accurate cross-session alignment, we propose that each dynamical system projects uniquely onto the obseerved subset of ensembles in each session. This assumption prevents two dynamical systems (e.g., "ensemble 1 → ensemble 2 → ensemble 3" and "ensemble 1 → ensemble 4 → ensemble 3") from being indistinguishable from the data available in that session (e.g., a session with only ensembles 1 and 3 recorded).

# E COMPUTATIONAL COMPLEXITY

CREIMBO's learning process involves learning both the ensemble compositions per session ($\{A^d\}_{d=1}^D$), the ensembles' temporal activity ($\{x_t^d\}_{d=1}^D$), and its underlying temporal evolution, which requires identifying the global (session-invariant) dynamic operators ($\{f_j\}_{j=1}^J$) and their per-session temporal coefficients ($\{c_t\}_{t=1}^T$).

Table 2: Parameter values for real-world human experiment

| Parameter | Value | Explanation |
|---|---|---|
| same_c | False | Indicates if sub-circuit coefficients are shared across sessions. |
| step_f | 0.2 | Step size for gradient descent in updating $\boldsymbol{f}$. |
| GD_decay | 0.992 | Gradient descent step-size decay rate over iterations. |
| max_error | $1 \times 10^{-9}$ | Maximum allowable error. |
| max_iter | 100 | Maximum number of iterations. |
| seed | 0 | Random seed. |
| normalize_eig | True | Normalize eigenvalues. |
| start_sparse_c | False | Start with sparse $\boldsymbol{c}$. |
| include_D | True | Whether to include $\boldsymbol{A}$. |
| step_D | 0.0001 | Step size for updating $\boldsymbol{A}$. |
| num_gradient_steps | 4 | Number of gradient steps. |
| add_avg | True | Add average. |
| sparsity_on_f_max | 45 | Maximum sparsity on $\boldsymbol{f}$. |
| take_multiple_gd | False | Take multiple gradient descents. |
| D_graph_driven | True | Graph-driven $\boldsymbol{A}$. |
| infer_x_c_together | False | Infer $\boldsymbol{x}$ and $\boldsymbol{c}$ together. |
| include_mask | True | Include mask. |
| norm_D_cols | False | Normalize columns of $\boldsymbol{A}$. |
| lambda_D | 0.3 | Regularization parameter for $\boldsymbol{A}$. |
| step_D_decay | 0.9999 | Decay rate for step size of $\boldsymbol{A}$. |
| l1_D | 2.74 | $\ell_1$ regularization for $\boldsymbol{A}$. |

### E.1 COMPLEXITY OF ENSEMBLE-MATRIX UPDATE

The loading matrix update relies on 4 main computational steps:

**1) Channel Graph Construction:** This operation, performed once for all $N$ channels of every state $d = 1 \dots D$, generates a channel graph $\boldsymbol{H}^d \in \mathbb{R}^{N \times N}$ for each state $d \in [1, D]$ by concatenating within-state trials $1 \dots M_d$ horizontally, resulting in a $N \times \sum_{m=1}^{M_d} T_m^d$ matrix. For simplicity, let $\widetilde{T} = \sum_{m=1}^{M_d} T_m^d$. The computational complexity of calculating the pairwise similarities of this concatenated matrix for all $D$ states is thus $\mathcal{O}\left(D\widetilde{T}^2 N(N-1)\right)$.

**2) The k-threshold step:** involves keeping only the $k$ largest values in each row while setting the other values to zero—the complexity will be $\mathcal{O}\left(\widetilde{T}\log k\right)$ per row for a total computational complexity of $\mathcal{O}\left(DN\widetilde{T}\log k\right)$ for $N$ rows and $D$ states.

**3) State Graph Construction:** This is a one-time operation that involves calculating the pairwise similarities between each pair of states. For simplicity, if we assume the case of user-defined scalar labels, and as in this case there are $D$ states (and accordingly $D$ labels), the computation includes $D(D-1))/2$ pairwise distances for $\mathcal{O}\left(D^2\right)$.

**4) Ensemble Inference (Eq. equation 1):** This iterative step involves per-channel re-weighted $\ell_1$ optimization. If the computational complexity of a weighted $\ell_1$ is denoted as $\mathcal{C}$, then the computational complexity of the re-Weighted $\ell_1$ Graph Filtering is $NL\mathcal{C} + LNk$, where $N$ is the number of channels, $L$ is the number of iterations for the RWLF procedure, and $k$ is the number of nearest neighbors in the graph. For the last term in Eq. equation 1, there are $p^2$ multiplicative operations involving the vector $\nu$ and the difference in ensembles, arising from the $\ell_2^2$ norm. Additionally, there is an additional multiplication step involving $\boldsymbol{P}_{dd'}$. For each state $d$, this calculation repeats itself $D-1$ times (for all $d' \neq d$). This process is carried out for every $d = 1 \dots D$. In total, these multiplicative operations sum up to $\left(p^2 + 1\right) D(D-1)$, resulting in a computational complexity of $\mathcal{O}\left(D^2 p^2\right)$.

### E.2 COMPLEXITY OF INFERRING THE ENSEMBLES ACTIVITY AND INTERACTIONS COEFFICIENTS (EQ. EQUATION 2)

The inference of $c_t^d$ for each $t = 1 \ldots T$ amd each condition $d = 1 \ldots D$, involves: $\widehat{c}_t^d = \arg\min_{c_t, x_t} \|y_t^d - ax_t\|_F^2 + \|\widehat{x_{t+1}}^d - FX_K^d c_t^d\|_F^2 + \lambda_c \|c_t\|_1$ where $FX \in \mathbb{R}^{p \times K}$ is the horizontal concatenation of $\{f_k x_t\}_{k=1}^K$.

This simplifies to a Lasso problem of the form

$$\min_\theta \frac{1}{2}\|\xi - \mathcal{M}\theta\|^2 \quad \text{s.t.} \quad \|\theta\|_1 \leq \tau,$$

with dimensions $\xi : (N + p) \times 1$, $\mathcal{M} : (N + p) \times (p + K)$, and $\theta : (p + K) \times 1$, where

$$\xi = \begin{bmatrix} y_t \\ 0_{p \times 1} \end{bmatrix} \in \mathbb{R}^{(N+p) \times 1}, \quad \theta = \begin{bmatrix} x_t \\ c_t \end{bmatrix} \in \mathbb{R}^{(p+K) \times 1},$$

and

$$\mathcal{M} = \begin{bmatrix} A & 0_{N \times K} \\ I_p & FX \end{bmatrix} \in \mathbb{R}^{(N+p) \times (p+K)}.$$

This can be solved with SPGL1 with up to num_iters iterations, resulting in a computational complexity of

$$O(\text{num\_iters} \cdot (N + p)(p + K)).$$

per session and time point and overall (across all sessions and time points):

$$O\left(\text{num\_iters} \cdot (N + p)(p + K) \sum_{d=1}^D T^d\right).$$

### E.3 COMPLEXITY OF INFERRING THE DYNAMICS' DICTIONARY

The inference of the basis-interactions dictionary $\{f_k\}_{k=1}^K$ (via Eq. equation 3) involves solving

$$\widehat{F}_{all} = \arg\min_{F_{all}} \|x^+ - F_{all}(CX)\|_F^2,$$

where $x^+ \in \mathbb{R}^{p \times \sum_{d \in \text{batch}} T^d}$ and $cx \in \mathbb{R}^{Kp \times \sum_{d \in \text{batch}} T^d}$. For simplicity, assuming that $\lambda_\rho = 0$, The problem simplifies to a LASSO formulation, which is solved using SPGL1 for $F^{all}$, with an overall computational complexity per iteration of $O(\text{num\_iter} \cdot Kp^2\tau)$, where num_iter represents the number of iterations required for convergence.

## F APPLICATION TO MULTI-REGIONAL CROSS-SESSION MICE NEURAL ACTIVITY

We further tested CREIMBO's ability to infer meaningful task variables underlying whole-brain multi-session data from mice Chen et al. (2024a) while performing a memory-guided movement task. The measurements were obtained using multi-electrode extracellular electrophysiology, with the overall dataset stored in DANDI Archive under the NWB format Chen et al. (2023).

Particularly, the data include electrophysiology recordings of neural activity from multiple brain areas per session (Figs. 26, 27), across overall 175 sessions, with numerous trials in each. The task the mice performed during the recordings involved selecting one of two "lick ports" based on auditory cues, with the left or right port leading to a reward depending on whether they heard a high or low

Table 3: Parameter table for the CREIMBO on Mice Mesoscale.

| Parameter | Value |
|---|---|
| max_error | 1e-09 |
| seed | 0 |
| normalize Fs | True |
| sparse_f | True |
| sparsity_on_f_max | 40 |
| increase_in_sparsity_f | 1.4318632048860938 |
| norm_D_cols | True |
| D_graph_params | {'with_kNN': True, 'with_norm': True, 'k': 15} |
| lambda_D | 0.4392235894758376 |
| update_type_D | spgl1 |
| latent_dyns_initialization | random |
| D_with_lasso | True |
| l1_D | 2.7440675196366238 |
| params_D | {'update_c_type': 'spgl1'} |
| num_regions | 10 |
| reg_type_on_c | spgl1 |
| lambda_x | 1.0291322176748077 |
| latent_dim | 30 |
| num ensembles per region | 30 |
| number of sub-dynamics | 10 |
| seed_f | 0 |
| sigma_mix_f | 0.1 |
| sparse_f_params | {'axis': '1', 'percent0': 20} |

tone. Neural activity was recorded across several brain regions in each session, with some regions overlapping between sessions, though no two sessions had identical coverage (refer to Fig. 1., **A-B**, in Chen et al. (2024a)).

We first loaded data from all sessions and selected areas for CREIMBO that featured the highest number of sessions including them. We identified the parent area of each sub-area labeled in the dataset and removed neurons with unlocated or NaN areas. Inspired by our exploratory analysis of parent-area distributions across sessions and neurons (Fig.27B,C), we selected the 10 parent areas with the highest number of neurons across sessions[1]. We then selected 20 random sessions. One of these 20 sessions contained only 'NaN'-located neurons, leaving 19 sessions for this analysis (Fig. 27D)[2].

We pre-processed the data using a 30 ms Gaussian kernel for firing rate estimation across all the sessions used (see F). We then selected only the neurons from the 10 parent areas and limited the analysis to the first 60 trials. Some sessions exhibited no activity or disconnection in certain trials, meaning some sessions included fewer than 60 trials (but no fewer than 40).

Next, we ran CREIMBO with 10 sub-circuits ($K = 10$) and $p_j = 3$ ensembles per area. The full list of parameters is shown in 3, and an example exploration of a random session is shown in (Fig.28,**A-C** show the firing rate, trial start-end times, and distribution of trial durations, respectively).

We identified ensembles per region (exemplary in 29A) with sub-circuits that capture both between-area (Fig.29C) and within-area interactions (Fig.29D), showing diverse motifs with varying degrees

---

[1]Used Parent Areas: Thalamus, Orbital Cortex, Secondary Motor Cortex, Basal Ganglia, Olfactory Cortex, Brain Stem, Somatosensory Cortex, Frontal Cortex, Hippocampus, Primary Motor Cortex

[2]Sessions DANDI names: '78a58614-09e4-4e29-a492-8bbeaf7a8cf6', 'ab103954-d99f-46b1-98ac-e6f91a1e9313', '32701cfa-8932-4d9b-9f4d-cc05cb22ae89', 'b080c738-17c9-4e0d-aee4-6d5371518f69', '886c4302-846a-4ef5-996a-6f02d6a81a5f', 'ba5b1ff3-753e-425c-baf0-e7cdd0c08093', 'ea856208-4240-404a-9034-d729ad6f4cda', 'ed759efa-dceb-4472-a2a3-4f7357d67665', '8a2ce9b2-2e98-4c37-8f2a-3b9c8b542086', '22791d80-26dc-4495-b4c0-651fe10e3298', 'a007fac1-96e7-4028-ad84-31a1b80db089', 'b39eb52a-0774-411b-b374-6eb08a73562e', '8dbc25ee-cc17-4504-a1b9-7d43643b9466', '981feb84-f209-4994-8838-02a0048feb87', '585dc9d4-be9c-4244-8fa7-e64309019afc', 'aa74e4d7-a79b-4179-adf3-497fb1237edc', 'ebfab58b-7c80-4f31-9b4e-2b33883bb14a', '8c724097-3876-48fc-94e0-832779e4f1be', '2e9cb8a1-457d-49ea-97fd-f7023434d231'

of cross-circuit correlations (Fig. 29B). Some cross-area interactions reveal effects originating from specific regions involved in the task. For example, $\boldsymbol{f}_1$ demonstrates hippocampal effects, which align with the task's memory demands, while, e.g., $\boldsymbol{f}_3$ shows interactions involving the frontal cortex, including follow-up activations, corresponding to the task's need for planning and execution. Other sub-circuits reveal inputs converging on specific areas. For instance, $\boldsymbol{f}_8$ shows motor cortex inputs, relevant for moving the tongue during the lick response, and $\boldsymbol{f}_2$ reflects inputs into the secondary motor cortex, which is involved in coordinating motor actions.

When extracting only within-area interactions, we observe unique motifs as well. For example, $\boldsymbol{f}_1$ shows further internal processing between basal ganglia ensembles, which may reflect the internal coordination and integration of motor planning and execution signals. $\boldsymbol{f}_7$ and $\boldsymbol{f}_2$ show strong effects of one ensemble on another within the secondary motor cortex, which could represent the modulation and refinement of motor commands for more precise movements. $\boldsymbol{f}_5$ demonstrates brainstem effects, with two ensembles influencing two other ensembles, potentially supporting basic motor functions such as muscle activation and coordination. $\boldsymbol{f}_4$ shows multiple self-activations of ensembles, which may capture self-regulatory processes like feedback inhibition or facilitation that modulate motor output and prevent over-activation.

We further examined CREIMBO's dynamics coefficients ($\boldsymbol{c}_t$) and identified task-related patterns with similar coefficient profiles across trials for the same task variable (Fig 4A for an example session, and B for hit vs. miss outcomes). Similar differences were observed across other task variables. To quantify the predictive power of the dynamic coefficients, we used them as input for training a simple one-vs-rest logistic regression classifier. Each trial was split into four equal-duration time windows, and the 10 dynamic coefficients were averaged within each window, resulting in 40 features (4 windows × 10 coefficients). These features were then used to predict task variables, including outcome (hit/miss/ignore), early lick (whether a lick was early), lick side instructed, and lick side performed in practice. Additionally, we tested the model to predict 2-3 variables simultaneously, which was more challenging due to the need to distinguish between multiple options. The resulting accuracy levels (Fig 4C, black stars) were significantly higher than chance levels (gray bars). We further analyzed the confusion matrix for each variable's prediction and applied the $\chi^2$ test to assess whether the distribution of predictions significantly deviated from the expected distribution under chance. We found p-values well below $1 \times 10^{-10}$ (see subtitles of Fig 4D, F, G, H, I, J), strongly supporting CREIMBO's predictive power.

Furthermore, we assessed the importance of subcircuits' activations across the four time windows (Fig.29E), with each block of rows representing one subcircuit and the blocks corresponding to different time windows. CREIMBO revealed patterns where certain time windows and dynamic coefficients were pivotal for different task variables. For instance, $c_7$ at $t_3$ was critical, while other coefficients, like $c_9$ at $t_3$, were specific to encoding the lick side. This specificity aligns with the fact that this time point likely reflects the final stages of processing during the task, including aspects of learning. When we examined $c_9$ (Fig.29C), we observed multiple inputs into the secondary motor cortex, related to the execution of the lick towards the trial's end, highlighting CREIMBO's ability to pinpoint region-specific interactions. Additionally, $c_8$ showed increased importance in the initial window $t_0$, contributing across multiple variables and capturing inputs into the hippocampus—an area integral to memory processing—underscoring its role in early trial stages. These findings further validate CREIMBO's ability to capture and predict task-related neural dynamics, offering insights into circuit-level brain activity across time.

## G  INFERENCE ASSUMING POISSON STATISTICS

CREIMBO, as outlined in the main text, assumes Gaussian-distributed *i.i.d* noise. This assumption is reflected in the inference procedure, which minimizes the $\ell_2$ and Frobenius norms (e.g., equation 1, equation 2). Particularly, these norms arise from minimizing the negative log-likelihood of Gaussian-distributed observations noise ($p(\boldsymbol{Y}_{n,t}^d) = \frac{1}{\sqrt{2\pi\sigma^2}} \exp\left(-\frac{\boldsymbol{Y}_{n,t}^d - [A^d X^d]_{n,t}^2}{2\sigma^2}\right)$, with some noise standard-deviation $\sigma$.

Electrophysiology, however, records trains of action potentials (spikes) that are often modeled as a Poisson processes. Binned spike counts are thus typically approximated as a Poisson random variable. However, when firing rates (FRs) are high enough, the Gaussian assumption is an adequate

approximation, and thus the Gaussian assumption remains common in data-driven neuroscience models (e.g., Linderman et al. (2016); Aoi & Pillow (2018)). However, some species (e.g., bats (Allen et al., 2021)) exhibit extremely low FRs, which may require Poisson likelihood functions. Thus we propose an extension to CREIMBO that with a Poisson likelihood function for ensemble and traces inference, that can replace the Gaussian model in low spiking-rates scenarios.

Our Poisson extension first involves transforming the spike-timing data to binned spike counts for all sessions $d = 1 \ldots D$ ($\{\boldsymbol{Y}^d\}_{d=1}^{D}$). Next, we consider the same goal of finding the ensembles $\boldsymbol{A}^d$ and their traces $(\boldsymbol{X}^T)^d$, under the assumption that the data follows a Poisson distribution.

To simplify, we focus on a single session $d$, with $\boldsymbol{y} = (\boldsymbol{Y}^d)^T$ ($T$ for transpose), and note that the extension below applies to all sessions. In the Poisson model, for each neuron $n = 1 \ldots N$, the likelihood of a certain spike count at time $t$ given a latent rate $\lambda_{t,n}$ is:

$$p(\boldsymbol{y_{t,n}}|\lambda_{t,n}) = \frac{\lambda_{t,n}^{y_{t,n}} e^{-\lambda_{t,n}}}{y_{t,n}!}$$

where $y_{t,n}!$ denotes the factorial of $y_{t,n}$.

Here we model the unknown rate $\boldsymbol{\lambda} \in \mathbb{R}^{T \times N}$ as the low-dimensional representation captured via the ensembles $\boldsymbol{A}^d \in \mathbb{R}^{N \times p}$ and their traces $\widetilde{\boldsymbol{x}} := (\boldsymbol{X}^d)^T \in \mathbb{R}^{T \times p}$ ($T$ for transpose), i.e., $\boldsymbol{\lambda} = \boldsymbol{A}^d \boldsymbol{X}^d$, such that the likelihood becomes

$$p(\boldsymbol{y}|\widetilde{\boldsymbol{x}}, \boldsymbol{A}) = \prod_{n=1}^{N} \prod_{t=1}^{T} p(y_{t,n}|\lambda_{t,n})$$

$$= \prod_{n=1}^{N} \prod_{t=1}^{T} p(y_{t,n}|[\widetilde{\boldsymbol{x}}\boldsymbol{A}^T]_{t,n})$$

$$= \prod_{n=1}^{N} \prod_{t=1}^{T} \frac{([\widetilde{\boldsymbol{x}}\boldsymbol{A}^T]_{t,n})^{y_{t,n}} e^{-([\widetilde{\boldsymbol{x}}\boldsymbol{A}^T]_{t,n})}}{y_{t,n}!}.$$

This can be trained by iteratively updating $\widetilde{\boldsymbol{x}}$ and $\boldsymbol{A}$ to minimize the negative log of the above likelihood, which can be represented mathematically as:

$$\{\widehat{\widetilde{\boldsymbol{x}}}, \widehat{\boldsymbol{A}}\} = \arg \min_{\widetilde{\boldsymbol{x}}, \boldsymbol{A}} \left[ -log \left( \prod_{n=1}^{N} \prod_{t=1}^{T} \frac{([\widetilde{\boldsymbol{x}}\boldsymbol{A}^T]_{t,n})^{y_{t,n}} e^{-([\widetilde{\boldsymbol{x}}\boldsymbol{A}^T]_{t,n})}}{y_{t,n}!} \right) \right]$$

$$= \arg \min_{\widetilde{\boldsymbol{x}}, \boldsymbol{A}} \sum_{n=1}^{N} \sum_{t=1}^{T} \left[ [\widetilde{\boldsymbol{x}}\boldsymbol{A}^T]_{t,n} - y_{t,n} \log([\widetilde{\boldsymbol{x}}\boldsymbol{A}^T]_{t,n}) \right] \quad (4)$$

where the logarithm is taken using the natural exponential base. Notably, in the above $\arg\min$, we chose to omit the constant term that emerge from taking the logarithm (i.e., $log(y_{t,n}!)$) as it does not affect the argument minimization.

In this Poisson case, in contrast to the Gaussian, we can no longer solve for $\boldsymbol{x}$ using e.g., least squares or LASSO; instead, we will update $\widetilde{\boldsymbol{x}}$ via Gradient Descent. The first step is to compute the gradient of the cost function in equation 4 with respect to $\widetilde{\boldsymbol{x}}$.

To simplify the calculation, we first notate the two components obtained in equation 4 by the auxiliary terms

$$g_1(\widetilde{\boldsymbol{x}}, \boldsymbol{A}) = \sum_{t,n} [\widetilde{\boldsymbol{x}}\boldsymbol{A}^T]_{t,n}$$

and

$$g_2(\boldsymbol{y}, \widetilde{\boldsymbol{x}}, \boldsymbol{A}) = \sum_{t,n} -y_{t,n} \cdot \log([\widetilde{\boldsymbol{x}}\boldsymbol{A}^T]_{t,n}).$$

The cost function from equation 4 thus can be written in terms of these functions as:

$$\{\widehat{\widetilde{\boldsymbol{x}}}, \widehat{\boldsymbol{A}}\} = \arg\min_{\widetilde{\boldsymbol{x}}, \boldsymbol{A}} \left[ g_1(\widetilde{\boldsymbol{x}}, \boldsymbol{A}) + g_2(\boldsymbol{y}, \widetilde{\boldsymbol{x}}, \boldsymbol{A}) \right]. \tag{5}$$

Hence, for updating $\widetilde{\boldsymbol{x}}$ via Gradient Descent, we need to find the gradients of $g_1$ and $g_2$.

We first establish two notations to make subsequent steps clearer:

- let $[1]_{(m,k)}$ denote a matrix of ones of size $m \times k$.
- let $\delta_{(M,P)_{m,p}}$ denote a matrix of shape $M \times P$ whose entries are all zeros except for the entry at index $(m, p)$, which is set to 1.

**Calculate** $\frac{\partial g_1(\widetilde{\boldsymbol{x}}, \boldsymbol{A})}{\partial \widetilde{\boldsymbol{x}}}$:

To begin, we can rewrite $g_1(\widetilde{\boldsymbol{x}}, \boldsymbol{A})$ as

$$g_1(\widetilde{\boldsymbol{x}}, \boldsymbol{A}) = \sum_{t,n} [\widetilde{\boldsymbol{x}}\boldsymbol{A}^T]_{t,n} = [1]_{(1,T)} \widetilde{\boldsymbol{x}}\boldsymbol{A}^T [1]_{(N,1)}$$

and follow the identity (taken from Petersen et al. (2008) Eq. 20):

$$\frac{\partial [\boldsymbol{a}^T \boldsymbol{M} \boldsymbol{b}]}{\partial \boldsymbol{M}} = \boldsymbol{a}\boldsymbol{b}^T \quad \text{where } \boldsymbol{a}, \boldsymbol{b} \text{ vectors, and } \boldsymbol{M} \text{ matrix.} \tag{6}$$

Building on this, let $[1]_{(1,T)}$ correspond to the $\boldsymbol{a}^T$ vector from equation 6, $\boldsymbol{A}^T [1]_{(N,1)} \in \mathbb{R}^{p \times 1}$ correspond to the $\boldsymbol{b}$ vector of equation 6, and $\widetilde{\boldsymbol{x}}$ correspond to the matrix $\boldsymbol{M}$. Then, the gradient of $g_1(\widetilde{\boldsymbol{x}}, \boldsymbol{A})$ with respect to $\widetilde{\boldsymbol{x}}$ is:

$$\frac{\partial g_1(\widetilde{\boldsymbol{x}}, \boldsymbol{A})}{\partial \widetilde{\boldsymbol{x}}} = \frac{\partial \left( [1]_{(1,T)} \widetilde{\boldsymbol{x}}\boldsymbol{A}^T [1]_{(N,1)} \right)}{\partial \widetilde{\boldsymbol{x}}} = [1]_{(T,1)} [1]_{(1,N)} \boldsymbol{A} = [1]_{(T,N)} \boldsymbol{A}$$

**Calculate** $\frac{\partial g_2(\boldsymbol{y}, \widetilde{\boldsymbol{x}}, \boldsymbol{A})}{\partial \widetilde{\boldsymbol{x}}}$:

First, we will rewrite the term $[\widetilde{\boldsymbol{x}}\boldsymbol{A}^T]_{t,n}$ as:

$$[\widetilde{\boldsymbol{x}}\boldsymbol{A}^T]_{t,n} = \sum_{j=1}^{p} \widetilde{\boldsymbol{x}}_{t,j} (\boldsymbol{A}^T)_{j,n} = \delta_{(1,T)_{1,t}} \widetilde{\boldsymbol{x}}\boldsymbol{A}^T \delta_{(N,1)_{n,1}} \tag{7}$$

$\delta_{(1,T)_{1,t}}$ is a row vector of length $T$ with a value of 1 at index $t$ and 0 elsewhere, and $\delta_{(N,1)_{n,1}}$ is a column vector of length $N$ with a value of 1 at index $n$ and 0 elsewhere.

We will use again the identity from Petersen et al. (2008) presented in equation 6, and now mark $\delta_{(1,T)_{1,n}}$ with the equation 6's vector $\boldsymbol{a}^T$, $\widetilde{\boldsymbol{x}}$ is again $\boldsymbol{M}$, and $b := \boldsymbol{A}^T \delta_{(N,1)_{j,1}}$. When applying the identity in equation 6 to equation 7, we thus obtain

$$\frac{\partial [\widetilde{\boldsymbol{x}}\boldsymbol{A}^T]_{t,n}}{\partial \widetilde{\boldsymbol{x}}} = \delta_{(T,1)_{t,1}} \delta_{(1,N)_{1,n}} \boldsymbol{A} = \delta_{(T,N)_{t,n}} \boldsymbol{A}$$

Given the above, we receive

$$\begin{aligned}
\frac{\partial g_2(\boldsymbol{y}, \widetilde{\boldsymbol{x}}, \boldsymbol{A})}{\partial \widetilde{\boldsymbol{x}}} &= \frac{\partial \left( \sum_{n=1}^{N} \sum_{t=1}^{T} -y_{t,n} \log \left( [\widetilde{\boldsymbol{x}}\boldsymbol{A}^T]_{t,n} \right) \right)}{\partial \widetilde{\boldsymbol{x}}} \\
&= -\sum_{n=1}^{N} \sum_{t=1}^{T} \frac{y_{tn}}{[\widetilde{\boldsymbol{x}}\boldsymbol{A}^T]_{t,n}} \delta_{(T,N)_{t,n}} \boldsymbol{A} \\
&= -\sum_{t,n} \psi_{t,n} \delta_{(T,N)_{t,n}} \boldsymbol{A} \\
&= -\left( \sum_{t,n} \psi_{t,n} \delta_{(T,N)_{t,n}} \right) \boldsymbol{A} \\
&= -\boldsymbol{\Psi} \boldsymbol{A}
\end{aligned} \tag{8}$$

where $\psi_{t,n} = \frac{y_{t,n}}{[\widetilde{\boldsymbol{x}}\boldsymbol{A}^T]_{t,n}}$, and $\boldsymbol{\Psi} \in \mathbb{R}^{T \times N}$ is a matrix whose entry at index $(t,n)$ is $\psi_{t,n}$.

Going back to the expression in equation 5, the full gradient of the Poisson loss with respect to $\widetilde{\boldsymbol{x}}$ is:

$$\frac{\partial[g_1(\widetilde{\boldsymbol{x}}, \boldsymbol{A}) + g_2(\boldsymbol{y}, \widetilde{\boldsymbol{x}}, \boldsymbol{A})]}{\partial \widetilde{\boldsymbol{x}}} = [1]_{(T,N)}\boldsymbol{A} - \boldsymbol{\Psi}\boldsymbol{A} = ([1]_{(T,N)} - \boldsymbol{\Psi})\boldsymbol{A},$$

and the projected gradient step in $\widetilde{\boldsymbol{x}}$ at iteration $m$ thus follows:

$$\widetilde{\boldsymbol{x}}^{m+1} \leftarrow \Pi\left(\widetilde{\boldsymbol{x}}^m - \eta^m\left([1]_{(T,N)} - \boldsymbol{\Psi}\right)\boldsymbol{A}\right)$$

where $\Pi$ can project the columns of $\widetilde{\boldsymbol{x}}$ onto some constraint set, with the identity transform used if no projection is desired, and $\eta^m$ is the gradient descent step size at iteration $m$.

For inferring $\boldsymbol{A}$, at each iteration $m$, we can then use the Spiral-Tap package Harmany et al. (2012) to infer the sparse ensembles.

The above ensemble model (including $\boldsymbol{A}$ and $\boldsymbol{x}$) is the only part that directly connects to the Poisson-distributed spike-counts ($\boldsymbol{Y}^d$), while the latent evolution of $\boldsymbol{x}$ via $x_t = \boldsymbol{F}_t x_{t-1}$ is independent of the Poisson assumption on the spike-counts $\boldsymbol{Y}^d$. Particularly, as the transition from $\boldsymbol{x_t}$ to $\boldsymbol{x_{t+1}}$ is dictated by the ensembles' dynamic interactions ($\boldsymbol{F_t} = \sum_{k=1}^{K} c_{kt}\boldsymbol{f}_k$), which we continue to assume follows a Gaussian distribution in the latent space, the dynamic inference part of CREIMBO remains unchanged from the main text.

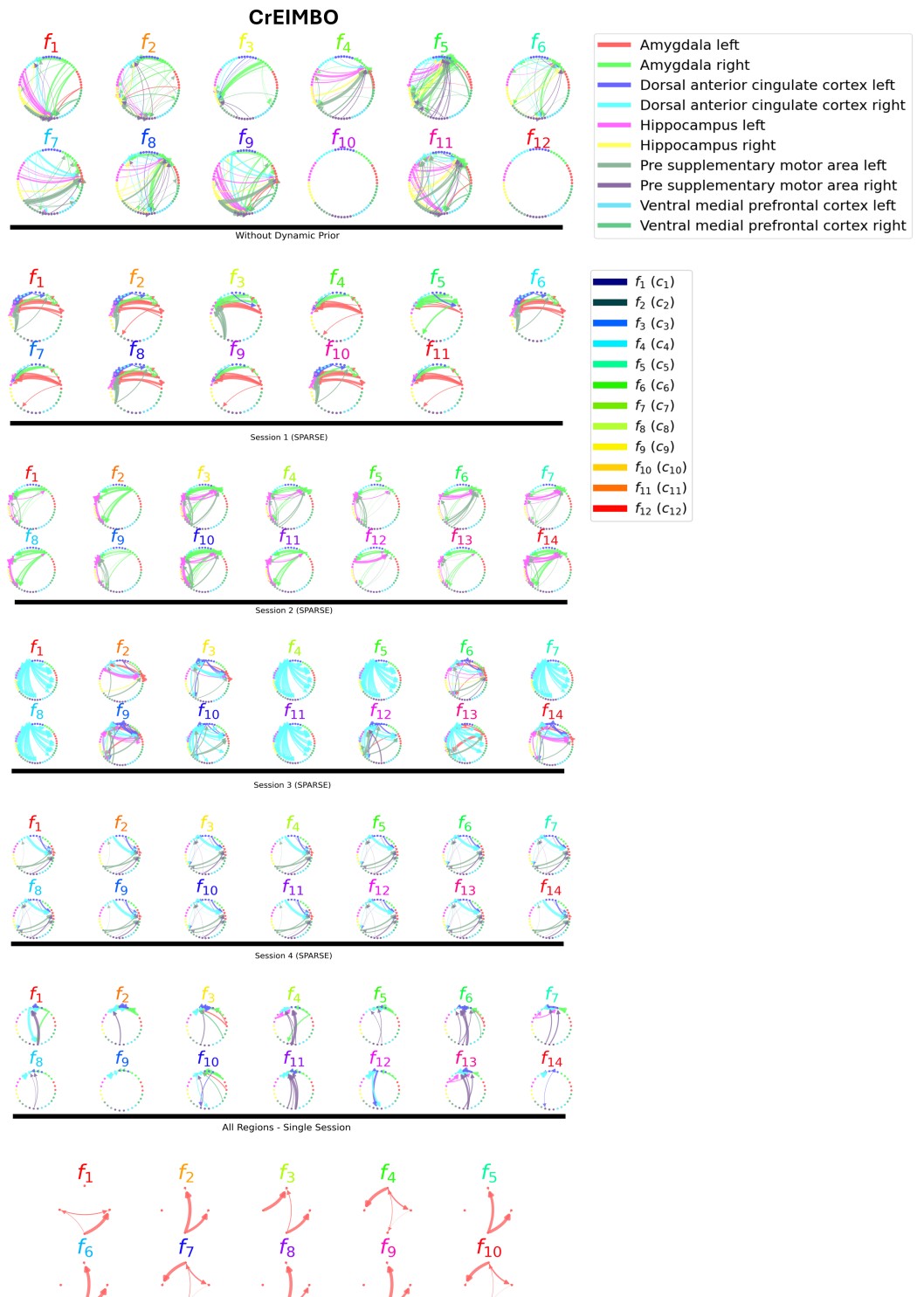

Figure 24: Comparison of sub-circuits identified by CREIMBO with those identified by other approaches using the real-world human neural recordings. CREIMBO reveals distinct motifs capturing multi-regional flows to specific areas, including both cross-regional and multi-regional interactions. In contrast, other approaches identify sub-circuits with overlapping flows that fail to capture the full spectrum of multi-regional interactions.

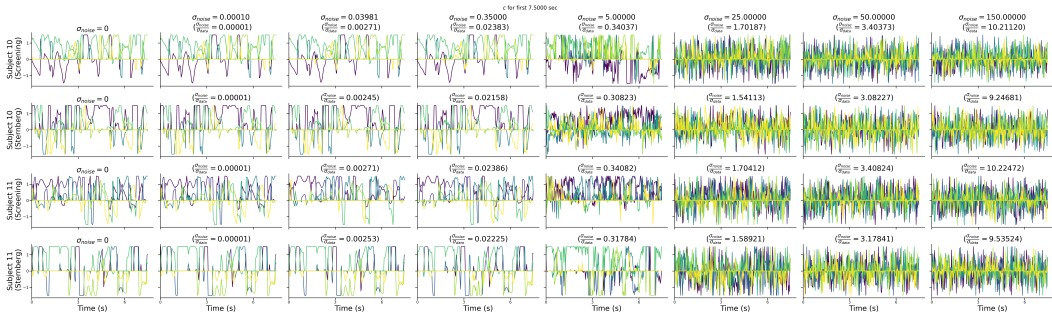

Figure 25: Identified dynamics' coefficients ($c_{k=1}^K$) for the human data experiment under increasing noise levels demonstrate robustness, with a rapid increase in internal pattern frequency at higher noise levels.

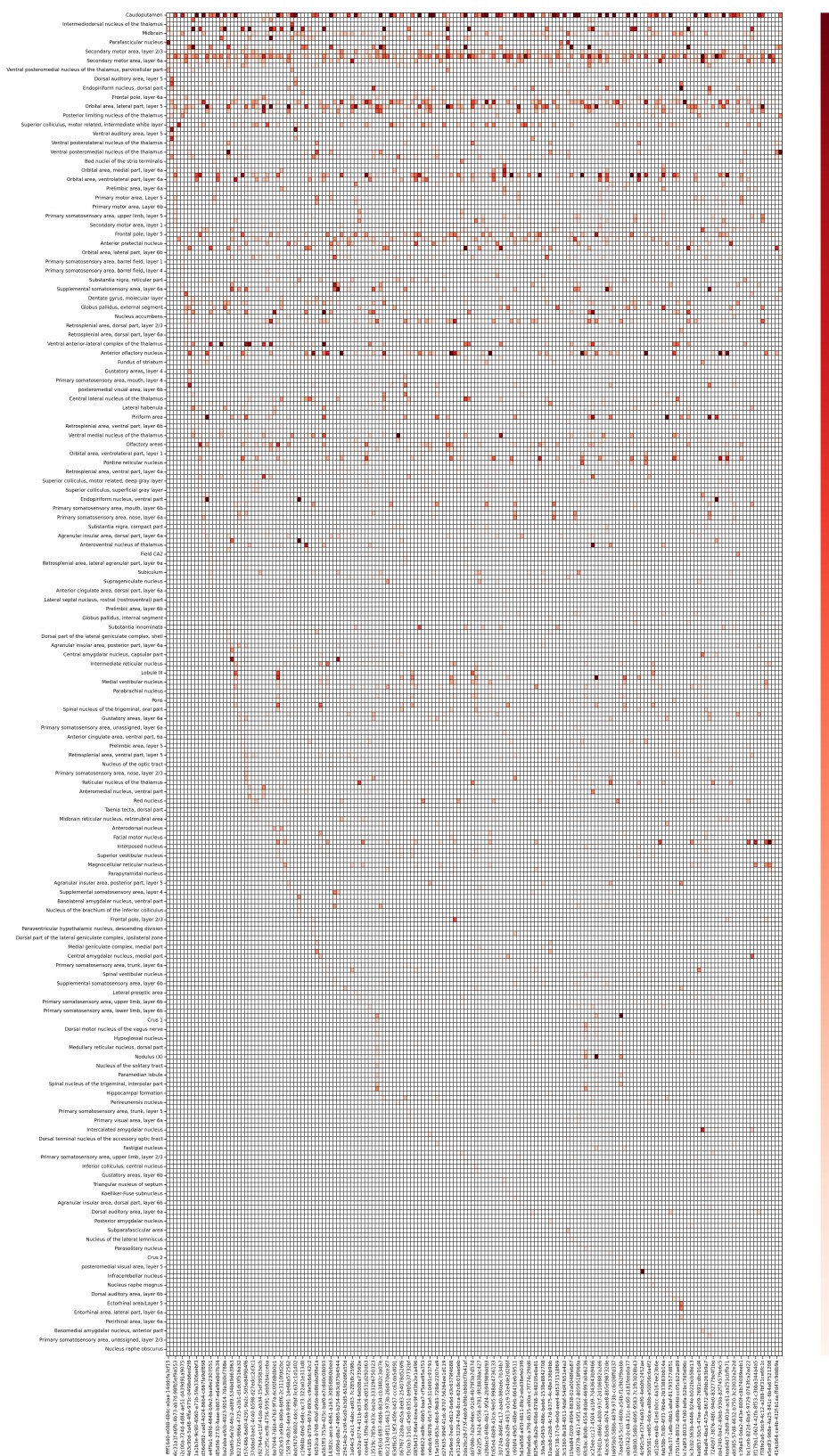

Figure 26: Number of neurons per area across all files

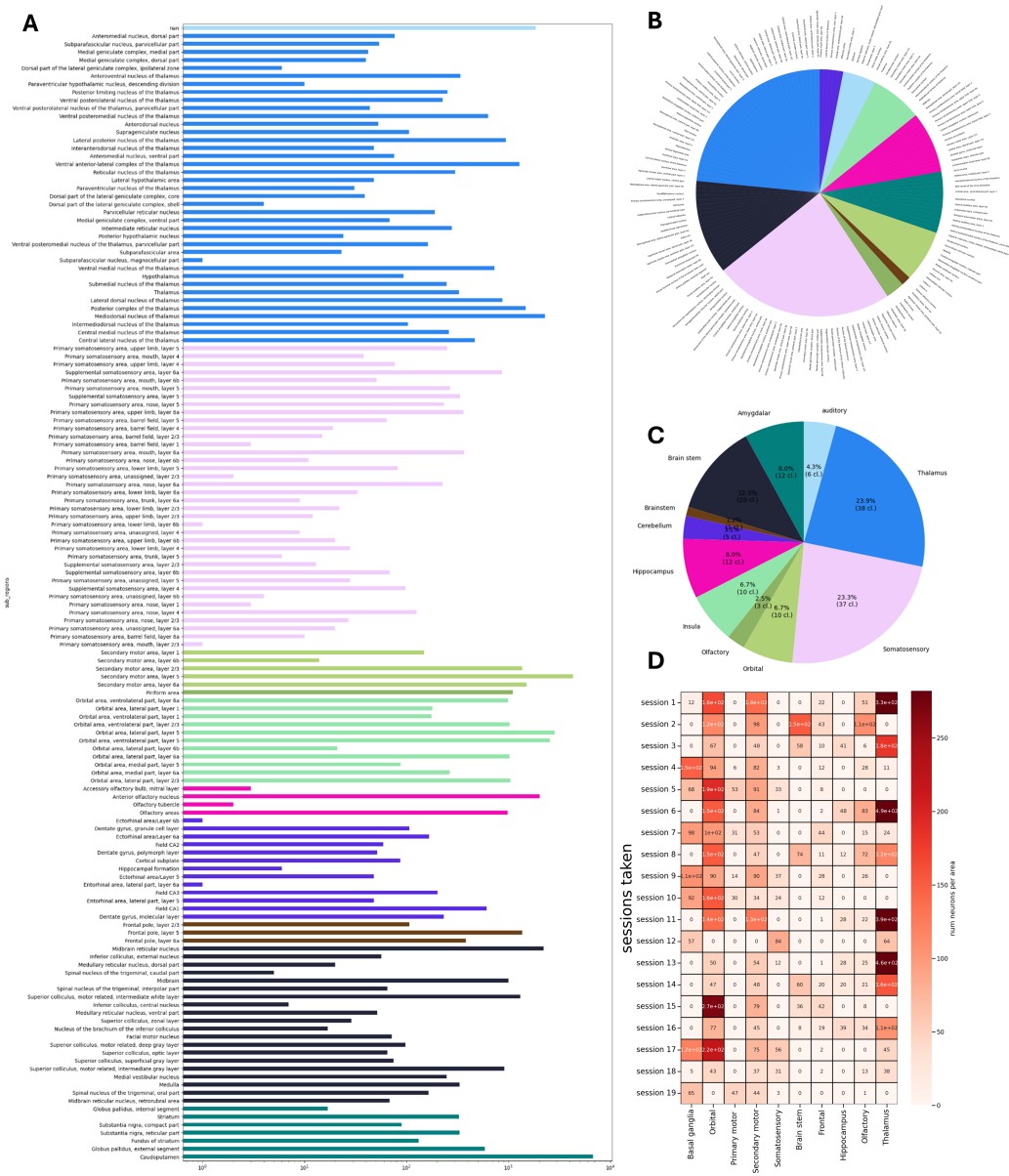

Figure 27: number of neurons per area as a motivation for area-focusing. **A:** Overall number of neurons per sub-area (colored by parent area) across all sessions. **B:** Distribution of areas with sub-area name across all sessions. **C:** Distribution of areas with parent-area name across all sessions. **D:** Number of neurons per parent-area (across the 10 selected areas) in the 19 used sessions.

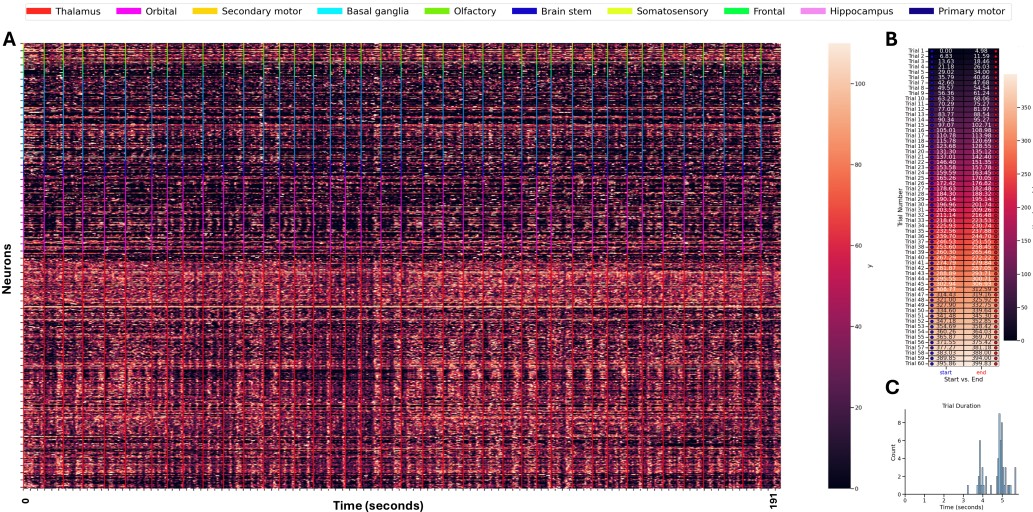

Figure 28: Example Session Chen et al. (2024a). **A:** Data firing rate over time separated by trials in vertical lines. the color of the vertical line captures the relevant brain area (example random session). **B:** Trials start and end times (in seconds) for examples session. **C:** Trial duration distribution for example random session.

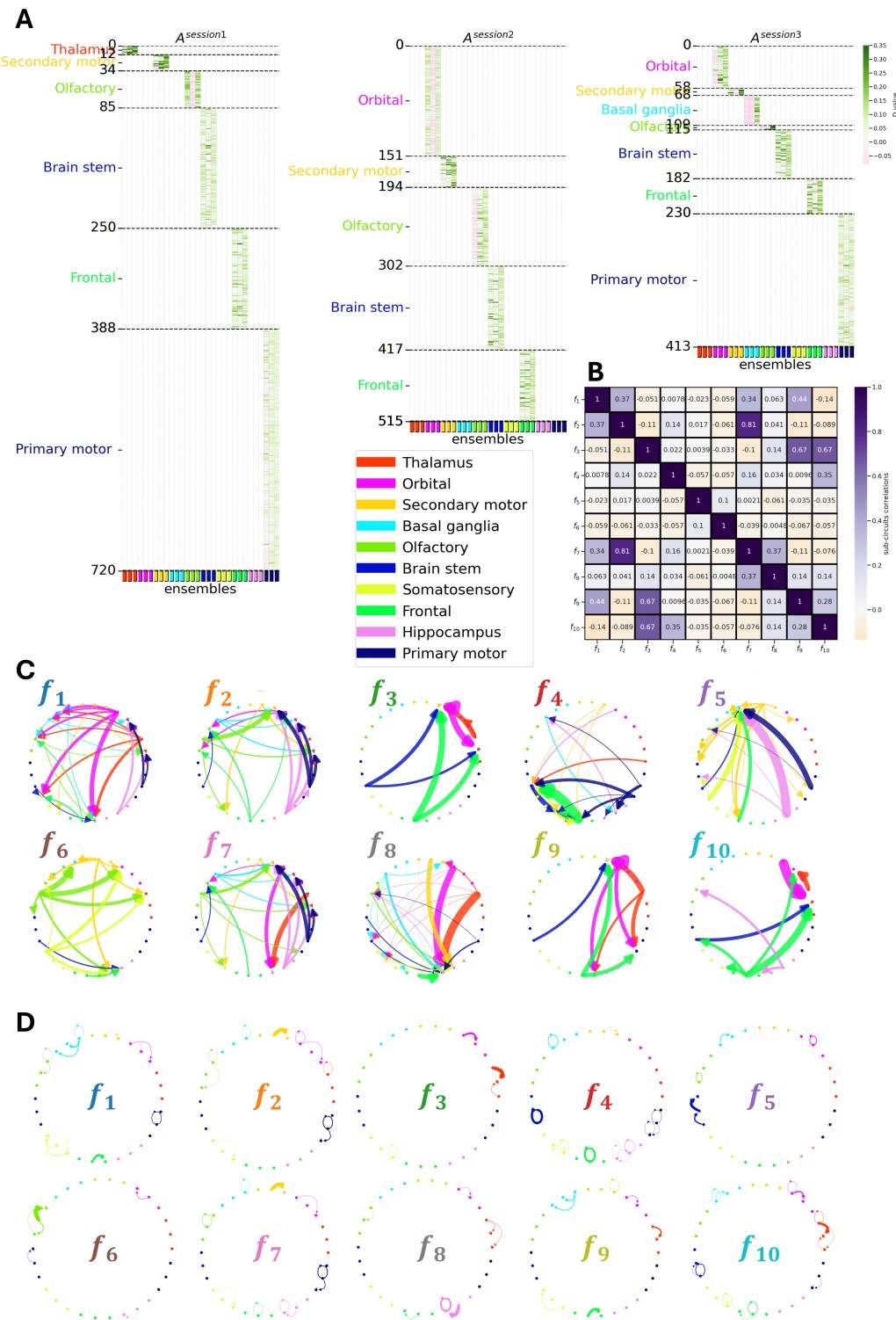

Figure 29: Identified components of the Mesoscale data (Chen et al., 2024a) using 19 random sessions with 40-60 trials within each. **A:** Identified ensemble matrices for three random sessions. **B:** Correlations between circuits. **C:** Between-areas interactions (extracted from $\{f_k\}_{k=1}^{K}$ via a block off-diagonal mask) **D:** Within-areas interactions (extracted from $\{f_k\}_{k=1}^{K}$ via a block diagonal mask)

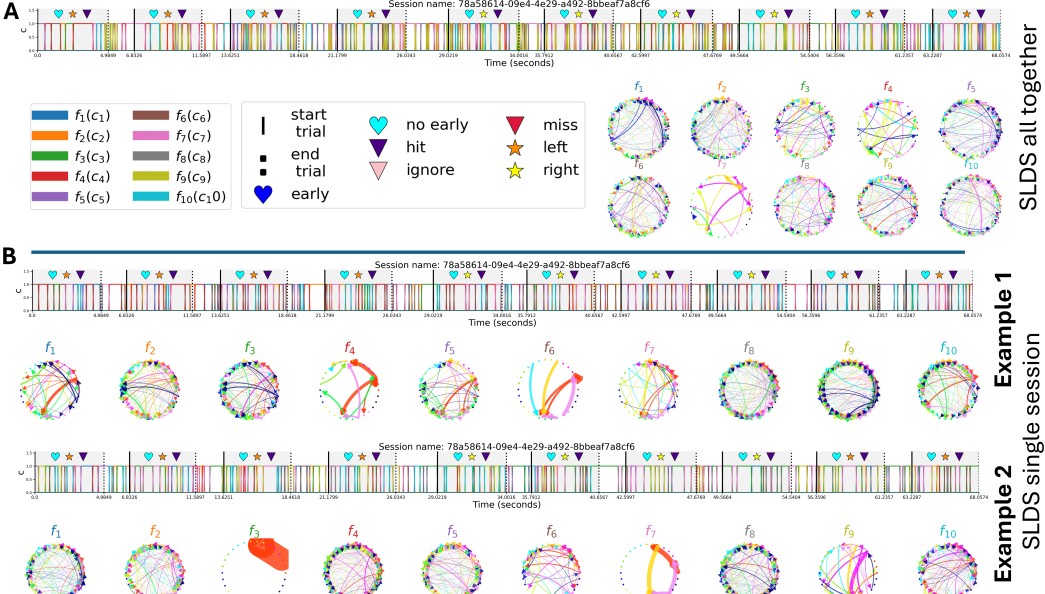

Figure 30: The components identified by the SLDS (Fig. 30) and rSDLS (Fig. 31) baselines (Linderman et al., 2016; 2020), when we applied them to the mouse mesoscale data from all sessions together (**A**) or individual sessions (**B**). For each model, we showed both the sub-circuits and their active coefficients for the first few trials. Compared to CREIMBO, the identified sub-circuits are more distributed and dense, and the dynamic coefficients capture only sharp transitions, i.e., unable to represent multiple co-occurring processes, in contrast to CREIMBO. Checking individual sessions with SLDS (**B**) or rSLDS (**C**), resulted in per-session set of sub-circuits (**F**s) that cannot enable cross-session sub-circuits comparison. All models ran with $K = 10$ sub-circuits and $p_j = 3$ ensembles per-region, to keep maximum consistency with our CREIMBO results from Figure 29.

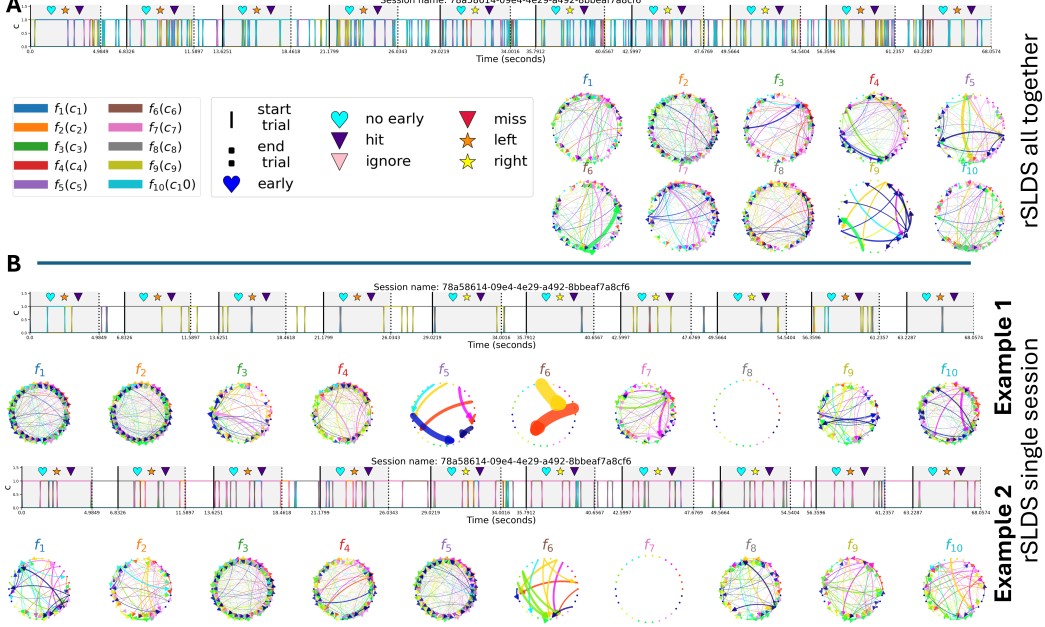

Figure 31: Same as Figure 30, but for the rSLDS baseline Linderman et al. (2016).

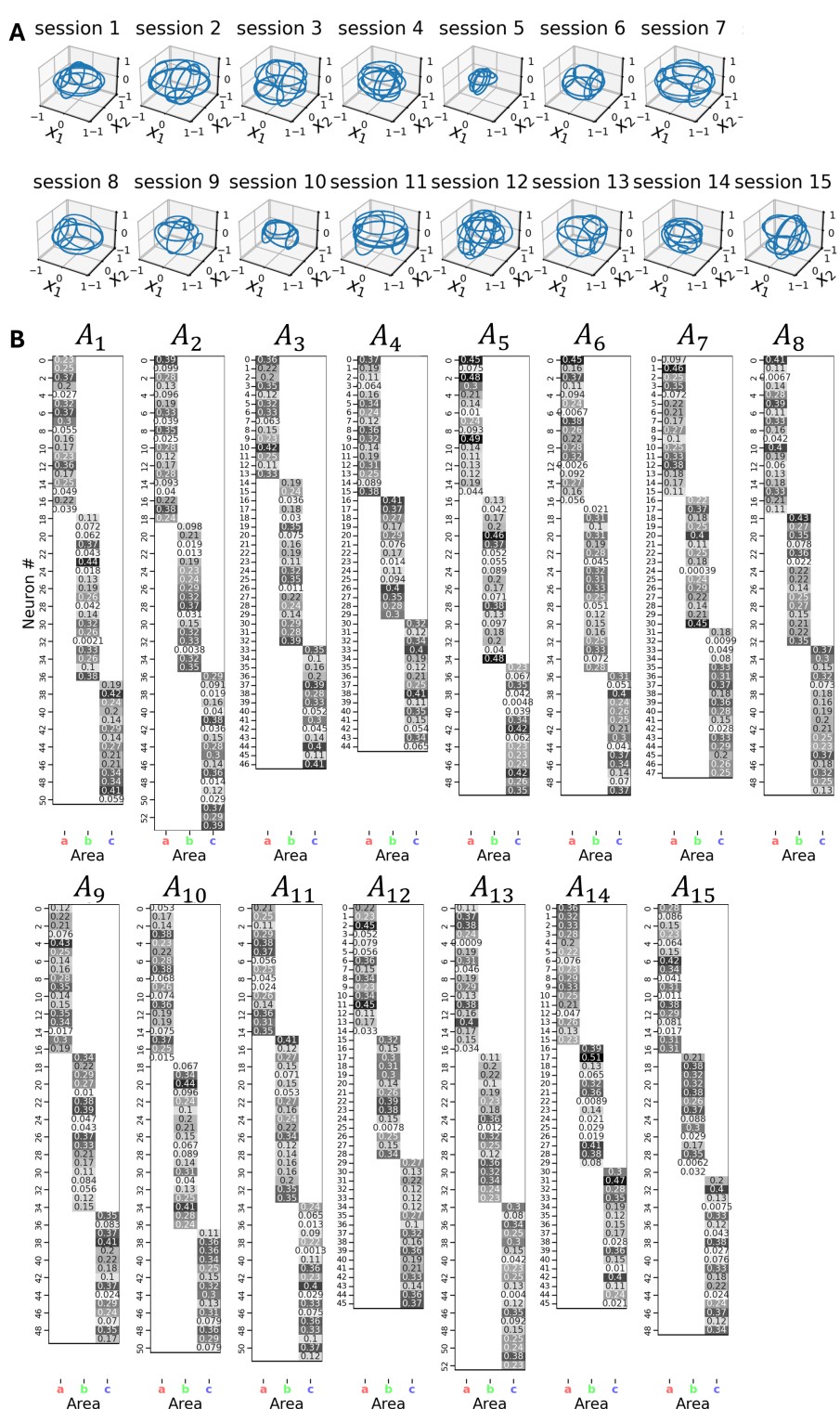

Figure 32: Cont. of Figure 11. **A:** Ground truth latent dynamics ($\{\boldsymbol{X}^d\}_{d=1}^{D=15}$). **B:** Ground truth ensemble compositions ($\{\boldsymbol{A}^d\}_{d=1}^{D=15}$).

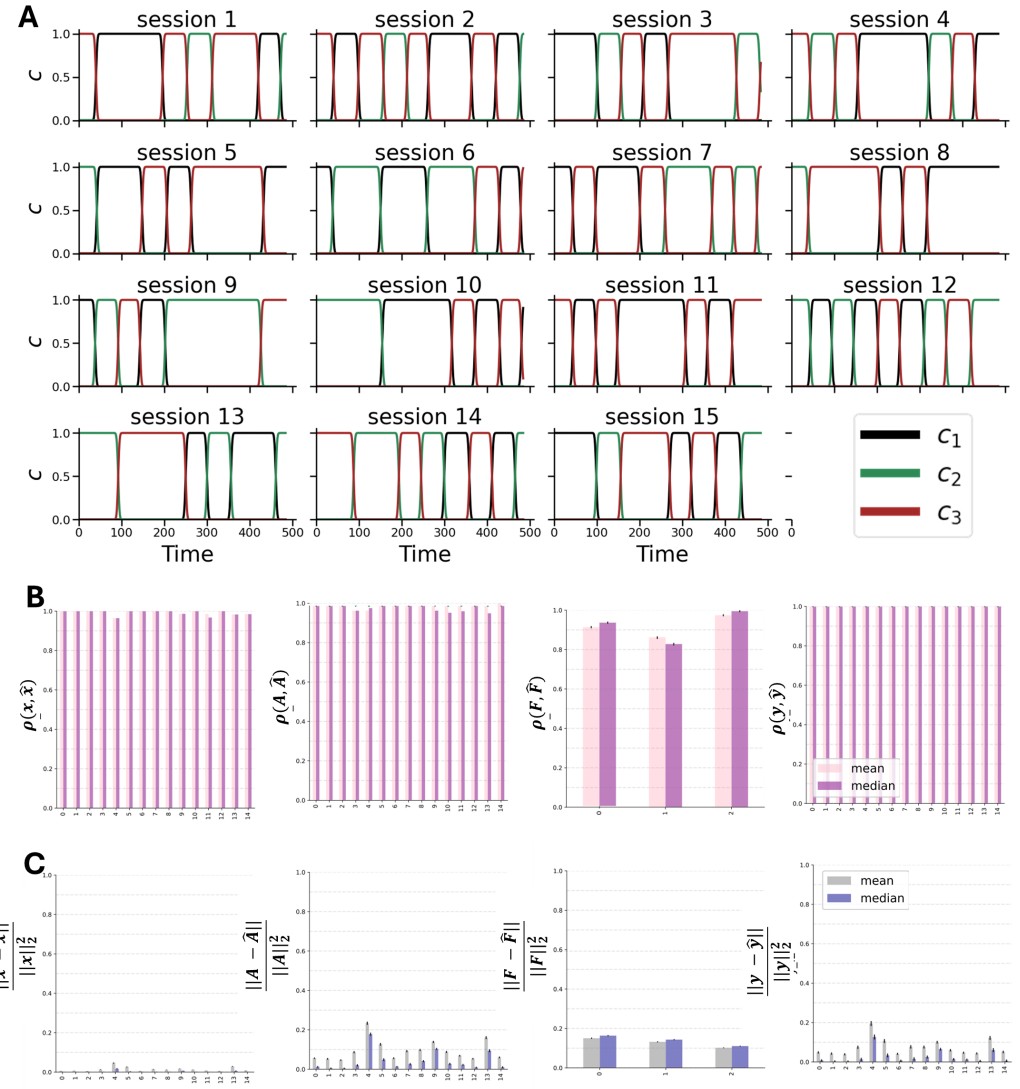

Figure 33: Cont. of Figure 11. **A:** Ground truth sub-circuits coefficients ($c$) vary between sessions. **B:** Correlation between ground truth and identified components for **A** (left), **x** (middle left), **F** (middle right), **Y** (right). **C:** $\ell_2$ distance between the components learned by CREIMBO and the ground-truth components for **A** (left), **x** (middle left), **F** (middle right), **Y** (right).

