# OpenReview forum: "CREIMBO: Cross-Regional Ensemble Interactions in Multi-view Brain Observations"
_ICLR.cc/2025/Conference — ICLR 2025 Spotlight_

### Official Review · Reviewer_eTfH · 2024-10-20

**Soundness:** 3
**Presentation:** 3
**Contribution:** 3
**Rating:** 8
**Confidence:** 4

**Summary:**

The authors propose CREIMBO that can extract interpretable neural subcircuits underlying multiregional neural signals by utilizing multi-session recordings that can have a variable number of recording units, trials, and durations. Through simulations, they show that their model successfully uncovers the ground truth dynamics when multi-session recordings are modeled together. In real data analysis, authors show that identified subcircuits can be sparse indicating specialized functionality, and reveal across-region interactions.

**Strengths:**

- Authors provide a new perspective over existing models of multi-regional neural models by allowing their model to leverage multi-session recordings that can help extract global and robust interregional interactions. While doing that, they keep the interpretability in-tact unlike deep-learning approaches.
- Authors performed exhaustive simulation experiments to show the importance of their model formulation.
- The paper is well-written and easy to follow. The proposed model architecture and training framework are intuitive and effective.

**Weaknesses:**

- Even though the paper is well written, the subfigures are very small and too crowded, which makes it hard to understand. Also, I think captions/labels for some appendix figures such as Fig. 11 should be improved.
- It seems like the number of ensembles per region ($p_j$) is an important hyperparameter for CREIMBO. I wonder if the model would reveal some consistent subcircuits with small $p_j$, such that these subcircuits explain most of the variance and are consistent across sessions and subjects. Also, how would training times vary with large $p_j$ values such that $p \approx N$? Overall, I think the scalability of such a model is an important aspect since modern neural recordings can include hundreds of neurons from one region, in such case, max($p_j$) = 7, can limit the interpretability of the identified subcircuits.
- In simulations (Fig. 2F), the authors show that the single-session model underperforms CREIMBO by a large performance gap. This can be caused by a small number of trials in each simulated session and short trials, but I could not find these details in the text (if it is in Fig. 11, I think it requires explanations in the caption). If this is indeed the case, I wonder how single-session models' performance would increase with longer sessions.
- I think the biggest contribution of CREIMBO is its multisession modeling over other approaches like mDLAG. However, the modern recording session can have hundreds of trials of data that can be sufficient to train models to understand multiregional dynamics. Therefore, I think it would be nice to see how their model compares to mDLAG even if it operates on a single-session basis. Based on Fig. 22, for the real dataset considered in this study, using multiple sessions in modeling seems important, and a comparison to mDLAG would highlight the importance of multisession modeling. Even though mDLAG does not learn dynamic matrices for temporal evolution, its learned readout matrices and lag parameters would still indicate interregional interactions, and I wonder if interregional interactions learned by mDLAG would be as poor as 'Session # (SPARSE)' in Fig. 22. Also, did authors compare their model to SLDS variants in real data as done for simulations? Overall, I think this work would benefit from more baseline comparisons to existing approaches in real data.

**Questions:**

- In line with my previous comment in weaknesses item 2, the authors show subject 10 in Fig. 3 which has a small number of available neurons compared to some other subjects (for Screening task). Are the example subcircuits and latent states extracted for the Screening task? Do the identified subcircuits and A matrices look denser for a higher number of neurons?
- In the ablation study in Fig. 2 ‘All regions (sparse)’ A matrix, authors show that the CREIMBO cannot infer the true underlying connectivity matrix, which makes sense since the inductive bias on block-diagonal A matrix is removed. Do authors use the same A matrix initializations in this ablation study? If not, how would the results change if the same block-diagonal initialization is applied for ‘All regions (sparse)’ case? Can authors provide an intuition on why inferred latent factors deviate significantly from the true latents? Would the same hold in the K=1 case, in which, a similarity transform would exist between not block diagonal sparse, not block diagonal not sparse, and block-diagonal sparse A matrices? Also, are the authors showing trial and session averaged latent states in these figures? If so, how does single-session latents look like for CREIMBO?
- The block structure imposed on A matrix seems like no regions have shared latent states and interregional interactions are captured by temporal dynamics. Did the authors try having some latent factors shared across all regions? How would it change the performance?

---

> ### Author Response · Authors · 2024-11-28
> **Response to reviewer eTfH - part 0**
>
> ### We truly thank the reviewer for their comments and for recognizing that CREIMBO provides a “new perspective over existing models of multi-regional neural models”, along with the importance of CREIMBO’s interpretability, the exhaustive simulation experiments we performed, and the well-written nature of the paper.
>
> **We would like to address their questions below:**
>
> ## 0) The reviewer suggested increasing subplots sizes.
>
> Thank you for pointing out that some of the figures were difficult to read. In the updated PDF, we increased subplot sizes and split Figure 11 into three figures (now Figures 11, 32, and 33 in the updated PDF, which will be consecutively numbered in the final version. The reason we chose to place the latter two at the end for now is to maintain consistency with the figure references during the authors-reviewers discussion).
>
> ## 1) The reviewer asked about the model scalability and the effect of the number of ensembles.
>
> We agree that scalability is important for modeling.  We thus included a computational complexity analysis in Appendix E.
>
> Importantly, we emphasize that CREIMBO also provides computational advantages by using a fixed set of ensemble-interactions sub-circuits across sessions, which makes it less costly than training the same procedure on individual sessions separately.
> The number of ensembles (e.g., $p_j = 7$) is indeed a model parameter, but a rough estimation of how many ensembles can explain $x%$ of the variance can be obtained via simple dimensionality reduction (e.g., PCA) on each session, with the option to run CREIMBO with slightly more ensembles to account for the sparsity applied.
> As with PCA (and most dimensionality reduction methods), at very low $p_j$ values (close to 1), the ensembles primarily capture the average activity within an area. Conversely, when $p_j \to N$ then 1) The model will capture significantly more noise, 2) Interpretability will decrease, as limiting dimensions is key for interpretability, 3) Each ensemble may become much sparser, missing the point of identifying meaningful groups of neurons, and 4) Cross-session alignment of ensembles functionality will be impaired.
> Hence our goal is to identify and operate in an optimal range for $p_j$ that balances sufficient data representation, interpretability, and preservation of neural group properties.
>
> With respect to training times, several factors influence them, including: 1) data-dimension aspects (e.g., number of neurons, sessions, and time points in each session), 2) model parameters related to representation dimension (e.g., number of ensembles, number of sub-circuits), 3) model parameters related to optimization (e.g., choice of solver for LASSO, batch size in iterations), and 4) hardware considerations (e.g., number of cores and nodes, system load). In our experience, synthetic data and sample checks on small subsets of real data can take a few minutes, the real-world human data takes around an hour, and the richer mouse brain data may take a few hours, depending on the number of sessions and number of trials per session we use. In our experience, the number of ensembles (as long as it is within a reasonable range considering each session dimensions) did not significantly impact the aforementioned training times, as the sparsity applied to each row in the ensemble matrices promotes sparse neurons-to-ensemble membership patterns, which helps limit the complexity.

---

> > ### Author Response · Authors · 2024-11-28
> > **Response to reviewer eTfH - part 1**
> >
> > ## 2) The reviewer asked about the importance of multiple sessions versus longer or more trials within a single session
> >
> > Thank you for this insightful question. This touches on how much data the model needs to learn robust and rich representations.
> >
> > The main data dimensions that affect model performance are: 1) the number of time points (within each session), 2) the number and distribution of neurons recorded in a session across populations and areas, and 3) the number of sessions. Each contributes important axes of information and cannot be fully replaced by the others.
> >
> > When focusing on the advantage of multiple sessions over a single session with longer or more trials, it is important to remember that real-world data (e.g., using Neuropixels probes) is often limited by the number of neurons it can record simultaneously within a session. While these numbers are increasing with technological advances, it is still not possible to capture the activity of all neurons in the brain or even all neurons within an area of interest, as practical limitations (e.g., electrode crowding, interference, and cross-talk) restrict the amount and diversity of neurons that can be recorded together.
> >
> > **Therefore, each session typically captures only a small subset of neurons or brain regions, and different sessions record different subsets of neurons that cannot match in identity** We reframe this cross-session variability, typically seen as a challenge, as an advantage that offers complementary perspectives into the brain and enables the leveraging of cross-session information that cannot be obtained in a single session, regardless of its duration. For example, in the new mesoscale data we have added to the paper [1] and in other datasets, such as the IBL dataset [2], different sessions include different subsets of areas, with none encompassing all the areas featured in these datasets. Therefore, learning dynamics from a single session, even with an unlimited number of trials, would prevent us from capturing the dynamics of certain populations and regions.
> >
> > Regarding the performance of single-session models with longer sessions (i.e., more trials or longer trials), we believe there may be some improvement in robustness with very long sessions due to learning from more samples. However, this performance improvement will still not allow the model to access areas or populations unobserved within that session. Thus, while longer trials or more trials within a single session may help for more robust representations, they cannot fully replace the advantage of leveraging information from multiple sessions.
> >
> > We appreciate raising this question and have also clarified the caption in the text.
> >
> > [1] Susu Chen, Yi Liu, Ziyue Aiden Wang, Jennifer Colonell, Liu D Liu, Han Hou, Nai-Wen Tien, Tim Wang, Timothy Harris, Shaul Druckmann, et al. Brain-wide neural activity underlying, memory-guided movement. Cell, 187(3):676–691, 2024
> >
> > [2] International Brain Laboratory, Benson, B., Benson, J., Birman, D., Bonacchi, N., Carandini, M., ... & Witten, I. B. (2023). A brain-wide map of neural activity during complex behaviour. Biorxiv, 2023-07.

---

> > > ### Author Response · Authors · 2024-11-28
> > > **Response to reviewer eTfH - part 2**
> > >
> > > ## 3) The reviewer suggested comparing CREIMBO to mDLAG.
> > > We appreciate the suggestion. While we agree that comparing the interactions found by CREIMBO to those found by mDLAG [1] can be exciting, we believe these models aim to uncover different properties in the dynamics. Particularly, as also mentioned in the mDLAG paper’s discussion [1] with respect to mDLAG vs. LDSs, “these approaches can be complementary”. Particularly, the GP-based description of mDLAG can be useful for exploratory data analyses to identify the parametric dynamical model  and optimal delay, while LDSs-based models, including CREIMBO, can then leverage these findings to discover the underlying set of dynamical components. We thus believe that integrating mDLAG and CREIMBO—where mDLAG focuses on finding the optimal delay and CREIMBO exploit this delay estimation to further identify a multi-time-scale set of co-active underlying sub circuits—can be an exciting direction for future research. Thank you for this suggestion, we have added this idea to our updated PDF (last sentence in the main text).
> > >
> > > [1] Gokcen, E., Jasper, A., Xu, A., Kohn, A., Machens, C. K., & Yu, B. M. (2024). Uncovering motifs of concurrent signaling across multiple neuronal populations. Advances in Neural Information Processing Systems, 36.
> > >
> > > ## 4) The reviewer asked if we compared CREIMBO to baselines in real data.
> > > We now added comparisons of our new real-world CREIMBO experiment to baselines including SLDS and rSLDS variants (Fig. 30, 31 vs. Fig. 29, 4). Notably, in the real data the ground truth components are unknown, and hence the comparison cannot rely on quantitative comparisons to ground truth circuits / ensembles / activations.
> > > However, when exploring the components identified by CREIMBO compared to those identified by the baselines, we can observe apparent differences in the representations these models provide compared to CREIMBO. For details, please refer to our response to reviewer 8qa1 bullet #2 (`2)
> > > With respect to the real-data demonstration, the reviewer asked whether there are more interesting differences or unique patterns that can be obtained from CREIMBO vs. other baseline models.`).
> > >
> > >
> > > ## 5) The reviewer asked about how the results of the real-world data look for more subjects.
> > > We direct the reviewer’s attention to Figures 17,18, 20, 21, which plots the data, ensembles, and dynamic coefficients for more subjects. We emphasize that the subcircuits are shared across subjects and hence are only presented in Figure 3.
> > >
> > > ## 6) The reviewer asked about the ``all regions (sparse)’’ from the ablation experiment
> > > The goal of the ablation experiments is to demonstrate how approaches similar to CREIMBO, but lacking one of its key components, perform, thereby underscoring the necessity of each component. If the reviewer is referring to initializing the ensembles under the mask but training them mask-free, we believe that, due to random initialization, off-block diagonal values could become occupied over iterations, resulting in a structure similar to the one currently presented in this ablation. Notably, all variables in the ablation experiments were initialized using the same statistics as CREIMBO. However, while CREIMBO was trained strictly within the support of the block-diagonal mask, the `all regions (sparse)` condition was trained without a mask.
> > > If we have misunderstood your question, we would be grateful for your kind clarification.
> > >
> > >
> > > ## 7) The reviewer asked about the possibility of latent states being shared by different regions.
> > > Thank you for raising this insightful question about latent states being shared across regions. However, our modeling choice here of designing ensembles to include only neurons from a single area, was intentionally made to support:
> > > 1) distinguishing between within- and between-area interactions,
> > > 2) enhancing interpretability by clearly defining single-area ensembles and each sub-circuit node corresponding to a single area, and
> > > 3) facilitating consistency in ensemble functionality across sessions.
> > >
> > > Importantly, given sufficient ensembles, CREIMBO should be able to capture cases where ensembles cover more than one area. However, it will represent them as two separate ensembles—one for each area—and their belonging to a single large ensemble will be reflected in their exhibiting similar temporal traces. Namely, if two ensembles from different areas actually come from the same larger ensemble, this will be evident in their traces ($x$) similarity and in the values of the corresponding rows and columns of the $f$s.
> > >
> > > Thus,  CREIMBO promotes interpretability of inter- vs intra- regional interactions by capturing single-area ensembles, yet remains capable of identifying more complex cross-areas patterns.

---

> > > > ### Comment · Reviewer_eTfH · 2024-12-03
> > > >
> > > > I thank the authors for their response. I increased my score to 8 and recommend accept for this work.

---

### Official Review · Reviewer_jy8o · 2024-10-29

**Soundness:** 3
**Presentation:** 4
**Contribution:** 2
**Rating:** 6
**Confidence:** 4

**Summary:**

The recent developments in neural recording technologies allow recording from large populations of neurons from multiple brain regions. Latent space models are often used to analyze these datasets, but they are generally limited to the study of single or few populations of neurons recorded simultaneously. To overcome these limitations, the presented work introduces a new algorithm that can capture variability across recording sessions and across and within brain regions. The method assumes a shared latent representation across areas and structured priors given each session. The authors validated the method on simulated data and neural data, showing that the model can capture variability across and within brain regions.

**Strengths:**

The manuscript is clearly presented, referenced in the context of the relevant work and technically sound. The method was tested in simulations coming from the generative model, where the model was able to recover the parameter settings, and was able to better capture the variability compared to existing models. They also tested the model in neural recordings identifying functional connectivity between and within brain regions. Moreover, they tested the robustness to increasing levels of noise in the data.

**Weaknesses:**

Further work in needed to fully understand the impact of this work. The authors motivate their work from their ability to better understand behavioral tasks from asynchronous recordings of spiking neural activity. However, the authors only tested the model in a single dataset where they limit the analysis to functional connectivity. It would be relevant to assess behavioral variables, such as ability to decode the present image stimuli from the learnt representations. Since the emphasis of the model is brain functional connectivity, the method should be compared to alternative recording methods such as fMRI or the LFP present in the dataset. Moreover, the comparison to alternative models in the neural data is limited to one qualitative test. A comprehensive quantitative comparison, such as decoding or reconstruction performance, is needed to understand the capabilities of the proposed method. Along the same lines, it is not surprising that the model outperforms alternative methods when the simulated data is tailored to the given model. A more relevant comparison would be simulating neural data from a neural process with temporal, task and/or behavioral variability and fit the different models there. It would also be relevant to highlight the model strengths and weaknesses in simulated data. One of the motivations for this work is its application to spiking data, but the Gaussian assumption limits its application to this kind of observation model and must be handled in preprocessing. While the authors verbally list this and other limitations in the discussion, it would be informative to show the limitations and, more importantly, capabilities of the model as results.

**Questions:**

Can this model be applied to other organisms, like mice or rats? Extending the application to neuroscience animal models would greatly increase the impact of the presented work.

How difficult would it be to extend this model to use a Poisson observation model to better capture neural spiking activity?

---

> ### Author Response · Authors · 2024-11-26
> **Response to reviewer  jy8o - part 0**
>
> We sincerely thank the reviewer for the thorough feedback, and their recognition of the clear presentation and technical soundness of our method and validation. We would like to address their concerns below:

---

> > ### Author Response · Authors · 2024-11-26
> > **Response to reviewer  jy8o - part 1**
> >
> > ## 1) The reviewer’s main concern appears to center on the impact of the work
> > We thank the reviewer for expressing this concern and apologize if the impact was not demonstrated enough in the original submission. Below we outline why we believe that our work is impactful:
> >
> > We will start by (A) reviewing CREIMBO’s capabilities over other approaches, and then (B) show how these capabilities translate to understanding of neural mechanisms and task-variables encoding. **Specifically, we added new results to the paper based on an additional dataset of multiple neuropixel recordings in mice performing a decision making task.**
> >
> > ### *(A) Summarizing CREIMBO’s capabilities over other approaches as a basis for impact*
> > Emerging neural datasets often involve non-simultaneous observations of non-matching subsets of neurons and brain areas. These datasets are typically analyzed individually or via uninterpretable deep-learning models. CREIMBO leverages the shared information inherent in cross-session recordings, overcoming the inability to match cross-session neurons in terms of individual identities. **Currently, no other computational methods address all four of the following key abilities that CREIMBO offers. i.e.:**
> > - disentangling multiple co-occurring neural circuits,
> > - distinguishing within and between area interactions,
> > - providing a unified cross-session model while capturing session and trial variability, and
> > - maintaining representational interpretability with respect to co-occurring neural interactions.
> >
> > **Thus, the concept, modeling, and capabilities of CREIMBO  enable the discovery of interpretable non-stationary co-active neural dynamics that are not accessible with current methods.**
> >
> > ### *(B) Demonstrating CREIMBO’s impact in understanding neural mechanisms and task-variable encoding during decision-making via additional Mice decision-making task experiment.*
> > We agree with the reviewer that providing additional evidence of CREIMBO’s impact can further strengthen our work. To address this, **we added another application to the updated paper PDF**, to mouse multi-regional whole-brain Neuropixels data [1] during a memory-guided decision-making task **(Appendix F and the end of the  experiments section, including Figures 4, 26–29 in the update PDF).**
> >
> > In this additional dataset, we demonstrate CREIMBO’s significance in identifying cross- and within-region interactions, including in key brain areas assumed to be important for the task.  The subcircuit activations exhibit distinct patterns across task outcomes (hit  vs. miss) with pattern consistency across same-outcome trials, which underscores its ability to recover meaningful neural processes related to the task outcome (see Figures 4A,B, and Figure 29).
> > **Notably, we were able to decode task variables based on the subcircuit activity identified by CREIMBO as the only input**. Specifically, we trained an L1-regularized Logistic Regression (LogReg) model to predict, both individually and jointly, trial outcome, lick side, and early lick information based on the dynamic coefficients $c_{kt}$, using time-averaged values of $c_t$ within a time window for each trial (4 windows per trial, 10 subcircuits, 40 features).
> > This classifier, trained on the sub-circuits activations from all sessions together as its only input, was able to predict task variables significantly above chance (p < 1e-10, see Fig. 4 and the last experiment in the main text). This test demonstrates the behavioral significance of the neural circuits and activations learned in CREIMBO.
> >
> > When examining the `feature importance' of that coefficient-driven classifier (i.e.. the features weights from the LogReg classifier), to identify which sub-circuits ({$\{c_{kt}\}$}) are important for each task variable across these four trial time windows—we found that the sub-circuits and time-windows important for predicting different task variables capture cross-region interactions that align with pathways thought to play a role in memory guided decision making (see Appendix F and Figures 4, 26-29 in updated PDF).
> >
> > For example, CREIMBO identifies a subcircuit with flows into the hippocampus that presents increased importance at the beginning of the trials (Figure 4E $c_8$ & $t_0$, Figure 29C, ${f}_8$). This aligns with the hippocampus’ known role in initial processing in memory-related tasks and further provides additional insight into what are the inputs that modulate its activity (see Fig. 4, 29, Appendix F, last experiment in the main text). **We observe similar trends in other areas, with additional examples provided in the updated PDF (Appendix F).** These examples highlight the biologically significant components that can be discovered using CREIMBO.
> >
> > We believe that these results & insights further clarify the impact of CREIMBO for the ML-neuroscience community and address the reviewer's main concern.
> >
> > [1] Susu Chen, et al. Brain-wide neural activity underlying memory-guided movement. Cell, 2024.

---

> > > ### Author Response · Authors · 2024-11-26
> > > **Response to reviewer jy8o - part 2**
> > >
> > > ## 2) The reviewer suggested to further assess the model compared to behavioral variables.
> > >
> > > We thank the reviewer for their insightful suggestion. In the original submission we focused on dynamical connectivity from human data, believing this would be the main focus of interest given ICLR’s emphasis on machine learning.
> > >
> > > We agree with the reviewer, and have now also run and included an additional experiment using a rich mouse mesoscale dataset that has more defined behavioral variables than the human data. Please find this additional experiment in the updated PDF, with explanation about the data, processing, and results in Appendix F and Figures 4, 26-29. To further highlight this aspect, we also added a brief discussion about them within the main text (inside the Experiments section, titled “CREIMBO Discovers Regional Interactions Predictive of Task Variables”, instead of the noise-robustness figure which we moved to the Appendix).
> > >
> > > In short (and following  the former point ), we showed that CREIMBO’s dynamic coefficients over time can be significant predictors of task variables, including outcome (e.g. hit or miss), instructed lick side, performed lick side, and whether there was an early lick. Moreover, the coefficients’ ($\{{c}_{kt}\}$) importance for tak-variable prediction (as extracted via the LogReg feature importance)  identified in this analysis meaningfully align with the biological importance of the corresponding identified $\{f_k \mid k=1, \dots, K\}$
> > >  networks that are covering flows into/from areas thought to be involved in differet priods for solving the task.

---

> > > > ### Author Response · Authors · 2024-11-26
> > > > **Response to reviewer jy8o - part 3**
> > > >
> > > > ## 3) The reviewer suggested to compare CREIMBO to methods that identify functional connectivity in fMRI or LFP datasets
> > > >
> > > > We thank the reviewer for this suggestion. We note that while our model is designed to identify multi-scale neural ensemble interactions, its output differs significantly from the functional connectivity typically referenced in fMRI and LFP methods.
> > > >
> > > >
> > > > Specifically, the ensemble architecture gives neuronal-population-level resolution as to which neurons are part of each ensemble responsible for the dynamical influence from one area to another, along with the internal dynamics within each brain area.
> > > >
> > > >
> > > > Moreover the relationships CREIMBO aims to find are directional (which is different from many of the correlational based definitions of functional connectivity) and dynamic in that the strength of these connections can change moment-to-moment in time with fast temporal resolution, which is different from what some fMRI methodos look for.
> > > >
> > > > Additionally, CREIMBO is specifically designed for firing rate data (e.g., from Neuropixels or similar high-density electrode arrays), which captures neural dynamics at high temporal and spatial resolution beyond what is available in fMRI and LFP functional connectivity maps.
> > > >
> > > >  fMRI, which measures large-scale, slow hemodynamic responses across the whole brain, lacks the temporal resolution required to observe individual neuron activity or rapid neural interactions. Similarly, LFP data reflects local field potentials and high-frequency signals but does not capture the same level of individual neuron dynamics.
> > > >
> > > > Beyond that, no methods for LFP or fMRI can address the specific problem CREIMBO was designed for: disentangling co-active neural networks from firing rate data across sessions. Thus we look for significant additional information over LFP and fMRI based functional connectivity, such as, what are the internal neuronal populations within a brain area that are responsible for the dynamic connectivity to other populations within the same area and in different brain areas.

---

> > > > > ### Author Response · Authors · 2024-11-26
> > > > > **Response to reviewer  jy8o - part  4**
> > > > >
> > > > > ## 4) The reviewer expressed concern about the scope of comparison in the real-world neural data.
> > > > >
> > > > > In our neural data, we do not have ground truth for the components. While reconstruction performance is important, a lot of methods (e.g., simple PCA with sufficient rank) can yield a good reconstruction, but does not provide insights into brain interactions.
> > > > >
> > > > > Hence, while important,  reconstruction score should not be the main criteria here as for which method to use. Particularly, a major aspect in CREIMBO is its ability to provide **interpretable** representation (due to shared cross-session dynamical components) into the brain internal dynamics across sessions, while maintaining flexibility/expressivity via the circuits time changing coefficients. **We would like to find the balance between these two factors, i.e. to ensure good reconstruction, however to also promote this kind of interpretability.**
> > > > >
> > > > > Notably, the ideal test of reconstruction performance would involve comparing the identified components to their ground truth, rather than focusing solely on the reconstruction of the observations. However, as mentioned, the true components are unknown in the real-world data. Hence, as a proxy, we further tested the robustness and consistency of the real-world human data results under added noise injected into the observations. Finding that the model is robust to added noise, can be an interim proxy to understand the quality of the model recovered components. Namely, as we observed that the model results tend to remain consistent within a certain noise threshold, it is possible that, if the observation noise approaches Gaussian statistics, the results may also remain consistent within the unobserved range from the clean (completely ideal non-noisy) data to the natural-noisy observations.

---

> > > > > > ### Author Response · Authors · 2024-11-26
> > > > > > **Response to reviewer  jy8o - part  5**
> > > > > >
> > > > > > ## 5) The reviewer suggested another synthetic experiment that simulates a neural process.
> > > > > >
> > > > > > We truly appreciate the suggestion, but believe it is beyond the scope of the paper. While adding more synthetic experiments would naturally add results, this is endless, since more synthetic data can always be created. We have presented three synthetic datasets with varying properties (e.g., number of ensembles, neurons), across hundreds of random initializations and seeds, to demonstrate that CREIMBO can recover ground truth components, including subcircuits, activations, and ensemble structure across varying characteristics.
> > > > > >
> > > > > > **The motivation behind these experiments we included was to generate observations that emerge from ``ground-truth'' latent subcircuits, their activations, and ensemble structures, and test CREIMBO’s ability to recover these hidden components based on the observations only, while holding a ground truth version of them.**  While the reviewer's idea of modeling a neural process is  insightful, it brings us back to the challenge we face in real data, where the ground truth for circuits/activations/ensembles is unavailable, making it less applicable for the validation intended in this synthetic demonstrations.
> > > > > >
> > > > > >
> > > > > >
> > > > > > ## 6) The reviewer asked how difficult it would be extending the model to a Poisson distribution instead of a Normal distribution.
> > > > > >
> > > > > > This is a good point, and we thank the reviewer for their suggestion.
> > > > > >
> > > > > > We started with a Gaussian assumption since 1) it promotes comparability with many existing computational-neuroscience methods (e.g., SLDS, rSLDS, mTDR), which commonly use Gaussian distributions in their standard implementations, and 2) the Gaussian approximation is valid for common spiking rates across various species with sufficient frequency.
> > > > > >
> > > > > > As we indeed mentioned in the original submission, extension to other statistics (e.g., Poisson) can be an exciting step, particularly to address cases of very low firing rates. Hence, we agree with the reviewer that it is a good point and that a further explanation into what it requires may be useful to further improve our work.
> > > > > >
> > > > > > **Hence, we now included the development for a Poisson observation model in our updated PDF version. Please refer to section G in the appendix**
> > > > > >
> > > > > > Briefly, this involves replacing the Frobenius and L2 errors (derived from the Gaussian log-likelihood) with terms from the Poisson exponent. While some of the derivatives differ, the extension is conceptually consistent with the approach presented in the main text.

---

> > > > > > > ### Author Response · Authors · 2024-11-26
> > > > > > > **Response to reviewer  jy8o - part  6**
> > > > > > >
> > > > > > > ## 7) The Reviewer Asked About the Applicability of the Model to Other Organisms (e.g., Mice or Rats)
> > > > > > > That is a great question, definitely--CREIMBO is generalizable to multi-area neural recordings across species, which we now further emphasized in the manuscript via the additional experiment on mouse multi-regional mesoscale data (data from [1], please all see previous comments as well as Appendix F and the experiments section in the updated PDF, including Figures 4, 26-29).
> > > > > > >
> > > > > > > [1]  Chen, S., Liu, Y., Wang, Z. A., Colonell, J., Liu, L. D., Hou, H., ... & Svoboda, K. (2024). Brain-wide neural activity underlying memory-guided movement. Cell, 187(3), 676-691.
> > > > > > >
> > > > > > > # Summary of changes and additions with respect to reviewer jy8o comments:
> > > > > > > 1) We added an additional demonstration of CREIMBO on new data to further demonstrate its applicability and impact. Specifically, the data include mouse whole-brain Neuropixels recordings during decision making, and we believe this addresses:
> > > > > > > - The main concern of the reviewer regarding impact.
> > > > > > > - The reviewer’s question about additional organisms.
> > > > > > >
> > > > > > > Please refer to the updated PDF (last experiment, Supplementary F, and Figures 4, 26–29).
> > > > > > >
> > > > > > > 2) Using the above mesoscale mouse data, we further performed an analysis exploring the predictive power of CREIMBO's dynamic coefficients in predicting task outcomes and decisions, which we assume addresses:
> > > > > > >
> > > > > > > - The concern about the scope of real-world evaluations.
> > > > > > > - The suggestion about the ability to link the model outputs to task variables.
> > > > > > >
> > > > > > > 3)  We showed how the predictive contributions of different dynamic coefficients across various time windows within the task capture biologically meaningful brain interactions thought to play a role in the task. We hope this further demonstrate CREIMBO's impact.
> > > > > > >
> > > > > > > 4) We added a full Poisson-statistics development of the model in the updated PDF (Section G), which we believe addresses the second question of the reviewer.
> > > > > > >
> > > > > > > We believe these updates address the reviewer’s questions and concerns, and we sincerely thank them again for their suggestions, which have improved our work.

---

> > > > > > > > ### Comment · Reviewer_jy8o · 2024-11-27
> > > > > > > >
> > > > > > > > Firstly, I updated my score to reflect that the work with the added results should be accepted. I really appreciate the authors additional efforts to highlight the relevance of the proposed method and validate its applicability to a wider range of datasets.
> > > > > > > >
> > > > > > > > In response to the specific comments: 1) the addition of the new results applied to the other dataset clearly demonstrates its applicability to broader neuroscience applications. 2) Decoding methods can be used to understand how/which information is encoded in the latent representation. As such, it would be another way to validate the method's ability to extract the relevant information. 3) I agree, adding the results on Neuropixel recordings helps drive this point home. 4) The are other alternative methods to model inter-area dynamics (Glasser et al., NeurIPS 2020; or Balzani et al., 2023 ICLR; to name a couple). While I still believe that those comparisons could help understand the strengths and weaknesses of the model assumptions, they are not necessary. 5) The additional results are even better than further simulation, unless this is targeted to test the range of viable modeling regimes. 6) Nice work, I look forward to seeing the extensions in future work. 7) Addressed by new results.

---

> > > > > > > > > ### Author Response · Authors · 2024-11-28
> > > > > > > > >
> > > > > > > > > Thank you for your thoughtful feedback. We are glad that our additional experiments and results were able to better demonstrate the impact of our work and the significance of CREIMBO.

---

### Official Review · Reviewer_8qa1 · 2024-10-31

**Soundness:** 4
**Presentation:** 3
**Contribution:** 3
**Rating:** 8
**Confidence:** 4

**Summary:**

Through the study, the authors have proposed a novel analytical approach (named as “CREIMBO”) for learning dynamics of latent representations from high-dimensional electrophysiology. The major advances of CREIMBO comes from that compression of high-dimension and extraction of dynamics were conducted simultaneously in CREIMBO, while keeping the interpretability. Through experiments with synthesized- and real data, the authors have properly demonstrated the validity of CREIMBO in the study. As CREIMBO contains conceptual novelty and its validated effectiveness, I believe this model can be one of useful candidate models for studying high-dimensional, partially overlapped, data, such as intracranial EEG or multi-array spike data. Thus, CREIMBO will be useful to neuroscientists.

**Strengths:**

The major strength of this study comes from its conceptual advances embedded in the proposed CRIEMBO. While there have been efforts to model the high-dimensional brain dynamics via deep learning, many of models failed to yield interpretable features, which is the most important aspect in the neuroscience field. Thus, the neuroscience field still tends to rely on relatively simpler yet interpretable models. In this regard, I believe CRIEMBO proposed in this study can be a good solution for this gap, clearly upholding the originality of this study. The authors thoroughly examined the validity of the proposed model using simulated- and experimental data, leading to the high quality of this work. Plus, the clarity of this paper is relatively high as the model was well described in the manuscript, whereas the reviewer believes there are some rooms requiring the attention of the authors. Altogether, the scientific significance of this paper is clear and can be interesting to the electrophysiology field.

**Weaknesses:**

While the overall strength of this study is obvious, there is a major weakness: Insufficient details in simulation study with synthetic data. It is unclear how the synthetic data was generated, e.g., what was the noise level, what kinds of parameters used for. Due to this uncertainty, it is nearly impossible to understand some of the intriguing findings in this study, especially with synthetic data. For example, there is very high inconsistency in performance across sessions in other methods, but not for CRIEMBO (Fig. 2M). While it can be interpreted as robustness of CRIEMBO, it is also possible the choice of benchmark models was not optimal for this type of synthetic data. Related to it, there was no comparison work with real data. Thus, the superiority of CRIEMBO needs further validation.

**Questions:**

1. Please provide details of how synthetic data was generated.

2. In Fig. 2M and O, I observed the high inconsistency in reconstruction accuracy in other models, e.g.,  mp_rSLDS_Gauss (per tial). This is synthetic data and, I assume, each session contains similar level of noise. It is very unclear how other baseline models nearly completely fails but performed near perfectly in some sessions. My question is, does this inconsistency supposed to support the robustness of CRIEMBO?
3. With real data, the authors demonstrated the validity of CRIEMBO. While I agree with the author’s claim, it does not necessarily lead to the strength of CRIEMBO. Are there any interesting differences or unique patterns that can be obtained from CRIEMBO vs. other baseline models?
4. The authors checked the robustness of CRIEMBO over different noise levels, with real data. More common practice is evaluating the robustness of the models with synthetic data, one with grountruth. If there were any specific reasons why it was done with real data, please specify it.

---

> ### Author Response · Authors · 2024-11-28
> **Response to reviewer 8qa1 - part 0**
>
> ### We deeply thank the reviewer for their detailed feedback and acknowledge their recognition of CREIMBO’s "conceptual novelty and its validated effectiveness" and the importance of its interpretability in contrast to other computational neuroscience models.
>
> **We address the reviewer's questions below:**
>
> ## 1) The reviewer was concerned there are insufficient details about the synthetic data and asked about the consistency of CREIMBO’s results across sessions in contrast to other methods.
>
> We agree with the reviewer that the explanation of the synthetic data generation could be clearer. **We have thus added an additional description in the updated PDF (Appendix B) detailing the data creation and the considerations behind its generation.**
>
> Regarding the inconsistency in reconstruction accuracy across models (in contrast to CREIMBO’s robustness), we assume that the superior performance of the non-CREIMBO methods in sessions 1, 2, and 5 compared to their performance in sessions 3 and 4 may be due to the similarity in observations within the session group (1, 2, 5) vs. the session group of sessions (3, 4). Specifically, as shown in the cross-session data correlation values we added in Fig. 6 of the updated paper PDF, the observations in sessions 1, 2, and 5 are more similar to each other, while sessions 3 and 4 are more similar to each other, with decreased similarity between sessions across these groups. The greater number of sessions in the (1, 2, 5) group may cause these models to prioritize learning operators that are biased towards the linear approximations that better fit the similarity group with more sessions (i.e., the 1, 2, 5 group) over the smaller (3, 4) group.
>
> We assume that these switching models may be sensitive to and affected by this bias when learning from sessions together, since they rely on binary activations of a single operator at a time. This makes them less expressive when constrained to the same number of dynamic operators, as compared to CREIMBO. In contrast, CREIMBO mitigates this bias by allowing session-specific, time-varying (non-binary) weights for the shared operators, making it more flexible and robust to variations across session groups. We hope this clarifies the superiority of CREIMBO in this regard.
>
>
>
>
>
>
> Regarding the choice of benchmark models, we selected SLDS, rSLDS, mp-rSLDS, etc. because they are the closest existing methods to CREIMBO in terms of the meaning of the dynamical components they produce, making them the ideal and natural candidates for comparison. CREIMBO, however, addresses what we see as their major limitation concerning neural activity modeling—their assumption that brain dynamics follow a strict switching pattern.
>
> In contrast, brain activity is believed [1,2,3]  to involve inherently distributed or parallel processes, potentially encompassing multiple co-active mechanisms (e.g., processing task variables, feedback inputs, etc.). This biological assumption motivated our approach to synthetic data generation, which reflects activity emerging from co-occurring  processes.
>
>
> [1] Sigman, M., & Dehaene, S. (2008). Brain mechanisms of serial and parallel processing during dual-task performance. Journal of Neuroscience, 28(30), 7585-7598.
>
> [2]Mizumori, S. J., Yeshenko, O., Gill, K. M., & Davis, D. M. (2004). Parallel processing across neural systems: implications for a multiple memory system hypothesis. Neurobiology of Learning and Memory, 82(3), 278-298.
>
> [3]  Nelson, M. E., & Bower, J. M. (1990). Brain maps and parallel computers. Trends in neurosciences, 13(10), 403-408.

---

> > ### Author Response · Authors · 2024-11-28
> > **Response to reviewer 8qa1 - part 1**
> >
> > ## 2) With respect to the real-data demonstration, the reviewer asked whether there are more interesting differences or unique patterns that can be obtained from CREIMBO vs. other baseline models.
> >
> > Thank you for raising this insightful question.
> > We agree that emphasizing the scientifically interesting differences and unique patterns CREIMBO identifies in real-world data, compared to other methods, can further enhance our work.
> >
> > Hence, we conducted an additional real-world experiment using mouse mesoscale data ([1], see updated PDF Figs 4, 26-29), which features a more structured task (stimulus → lick → reward). This structured task better supports scientific interpretation of the components in relation to trial evolution compared to the human recordings from the original submission. On this dataset, we further compared CREIMBO’s results to several baselines, specifically SLDS and rSLDS variants, as they are the closest existing methods to CREIMBO in terms of the components they produce. Importantly, since the `true’ components are unknown in real-world data, comparison here cannot rely on quantitative comparisons to ground truth (as we did with the synthetic data), and so instead we focus on the interpretability of the sub-circuits,  trial consistency, and the dynamic evolution of the circuits coefficients.
> >
> > As shown in the updated PDF (Figs 30 and 31), the components identified by the baselines on this real data—resulted in more distributed and dense subcircuits, which we found less interpretable. Importantly, when analyzing the dynamic coefficients (the HMM states corresponding to $c$ in CREIMBO) produced by these methods in relation to task variables (the $c$ subplots in Figs. 30, 31), SLDS (Fig. 30) exhibits very frequent and fast switching patterns between subcircuits. This rapid switching behavior  potentially comes to compensate for its inability to model multiple co-occurring processes, requiring significantly more subcircuits overall which alternate very frequently within each trial to achieve enough expressiveness. Alternatively, for rSLDS trained on sessions individually (Fig. 31 B), the coefficients remain almost static throughout the task, with mainly one sub-circuit being active throughout the entire trial. This obscures the temporal evolution and phases (stimulus, choice, outcome, etc.) within the trials, which should require the involvement of multiple brain systems we aim to distinguish between, thereby limiting the interpretability of the representation with respect to decision-making evolution.
> >
> >
> > [1] Susu Chen, Yi Liu, Ziyue Aiden Wang, Jennifer Colonell, Liu D Liu, Han Hou, Nai-Wen Tien, Tim Wang, Timothy Harris, Shaul Druckmann, et al. Brain-wide neural activity underlying, memory-guided movement. Cell, 187(3):676–691, 2024

---

> > > ### Author Response · Authors · 2024-11-28
> > > **Response to reviewer 8qa1 - part 2**
> > >
> > > ### [continuing from bullet 2) ]
> > >
> > > Moreover, when examining the coefficients identified by these baselines (Figs 30,31), we observe they are not highly consistent across trials, which may arise from these models' attempt to capture more complex mechanisms than they can handle with a switched structure. For instance, for rSLDS all-sessions (Fig. 31A, top), while we generally see increased activity in certain sub-circuits (e.g., $f_{10}$ in cyan) across trials, the frequent switching patterns vary inconsistently across trials, making interpretation challenging. Similarly, for SLDS, we also observe increased activity in certain sub-circuits (Fig. 30, top; pink and yellow sub-circuits, $f_7$ and $f_9$), but again, the rapid switching between multiple circuits prevents clear consistency across trials, obfuscating the evolving interactions during different phases of the trial and how they evolve within trial types.
> > >
> > > It is important to note that SLDS and rSLDS are not designed for multi-regional communication, and as such, the sub-circuits identified by them (Figures 30, 31) mix neurons from different areas within ensembles, meaning they do not distinguish between intra- and inter-area dynamics.
> > > Importantly, we wish to highlight that while a further extension of rSLDS exists for multi-area communication [1], unlike CREIMBO, the implementation in [1] does not support missing (i.e., unobserved) areas in portions of the sessions, despite this scenario being common in real-world datasets, as seen in the mouse dataset we added in the revision [2] and other similar datasets [3].  Ideally, such unobserved areas could be considered as missing data and imputed by e.g., leveraging cross-session information, as CREIMBO does via the session-shared dynamics dictionary which define the universal patterns that span the possible time-varying ensemble interactions.
> > >
> > > Due to the above differences in what the multi-area rSLDS model [1] can handle, and despite the numerous adjustments we made to their implementation to accommodate these differences, running [1] on this real-world mice data [2] was not feasible with their current implementation and modeling assumptions. Specifically, as seen in [4,5] (lines 221 and 149, respectively), there is an assertion that the number of neurons per area must exceed the number of ensembles per area. Removing this assertion caused the model to fail to run. Further attempts from our hand to process the data and adjust it to [1]’s implementation requirements—such as extending the dataset by adding zero-activity 'synthetic' neurons from missing areas (with "number of ensembles per area +1" synthetic neurons per missing area)—led to optimization errors due to non-invertible matrices in [1]'s solvers.
> > >
> > > Altogether, we believe the additional analyses we performed on new real-world data (Figs. 4, 26-29), coupled with the extra comparisons to the SLDS and rSLDS baselines on this real mouse data (Figs. 30,31) demonstrate that the patterns and components identified by CREIMBO provide clearer, more nuanced and interpretable representation of the brain-wide behavior, than what alternative methods can provide.
> > >
> > >
> > > [1] Glaser, J., Whiteway, M., Cunningham, J. P., Paninski, L., & Linderman, S. (2020). Recurrent switching dynamical systems models for multiple interacting neural populations. Advances in neural information processing systems, 33, 14867-14878.
> > >
> > > [2] Susu Chen, Yi Liu, Ziyue Aiden Wang, Jennifer Colonell, Liu D Liu, Han Hou, Nai-Wen Tien, Tim Wang, Timothy Harris, Shaul Druckmann, et al. Brain-wide neural activity underlying, memory-guided movement. Cell, 187(3):676–691, 2024
> > >
> > > [3] International Brain Laboratory, Benson, B., Benson, J., Birman, D., Bonacchi, N., Carandini, M., ... & Witten, I. B. (2023). A brain-wide map of neural activity during complex behaviour. Biorxiv, 2023-07.
> > >
> > > [4] https://github.com/lindermanlab/ssm/blob/master/ssm/emissions.py
> > >
> > > [5] https://github.com/lindermanlab/ssm/blob/master/ssm/extensions/mp_srslds/emissions_ext.py

---

> > > > ### Author Response · Authors · 2024-11-28
> > > > **Response to reviewer 8qa1 - part 3**
> > > >
> > > > ## 3) The reviewer asked why we chose to check noise robustness on the real data rather than the synthetic data.
> > > > This is a great question, and we agree with the reviewer that it is typically common to test noise robustness on synthetic data. We were motivated to this analysis by the lack of ground-truth knowledge of components in real data, which prevents a direct assessment of the correctness of the components found with CREIMBO. Hence, the goal of this experiment was to find an alternative evaluation method for CREIMBO’s performance on the human recordings, through an analysis of CREIMBO’s consistency over varying SNR levels.
> > > >
> > > > Specifically, while traditional robustness experiments on synthetic data test the limits of how much noise is tolerable before the identified components no longer represent ground truth,  in our analysis we wanted to see if the components resulting from CREIMBO change dramatically as additional noise is added. A consistent set of estimated components would thus indicate that the results of CREIMBO are the product of the signal statistics rather than artifacts of the noise in the data.
> > > > Thus, the robustness to added noise injected to the observation serves as an interim proxy to evaluate the quality of the recovered components when no ground truth is available.
> > > > **We recognize that the motivation for this experiment could have been more clearly stated, and hence we have added this clarification in the updated PDF (see lines 453-456).**

---

### Author Response · Authors · 2024-12-04

We thank all reviewers for their thoughtful and positive feedback and recommendations that have helped improve our manuscript.

---

### Meta-Review · Area_Chair_fLyG · 2024-12-17

**Metareview:**

The authors propose CREIMBO, a novel model that extracts interpretable neural subcircuits from multi-region, multi-session electrophysiological data, maintaining both compression and interpretability. CREIMBO demonstrates its effectiveness in both simulated and real data, revealing specialized subcircuits and interregional interactions, making it a promising tool for neuroscience applications, particularly for analyzing complex, high-dimensional neural recordings.

The major weakness lies in insufficient details regarding the simulation study, particularly the generation of synthetic data, which raised concerns about the validity of some findings. There were also queries about the robustness of CREIMBO and its comparison to real data and other baseline models. However, these concerns were largely addressed in the rebuttal, with the authors clarifying the synthetic data generation process and providing more context around model comparisons. Another suggestion is that the font size in the figures is excessively small, making them difficult to read. Additionally, the paragraphs are overly compact. The introduction to the approach feels too vague and lacks mathematical detail. The models should be presented with clear mathematical formulations rather than relying solely on textual descriptions. Please address these points as thoroughly as possible in the final revision.

The model's conceptual novelty and its ability to simultaneously compress data while preserving interpretability are major strengths, especially compared to deep learning approaches often used in neuroscience. CREIMBO was thoroughly validated through simulations and real data analysis, showcasing its potential in understanding multiregional neural dynamics. The paper is clear, well-written, and makes significant contributions to the field, demonstrating both technical rigor and scientific relevance.

Despite some weaknesses in the simulation study and comparisons with other models, the novelty, clarity, and demonstrated effectiveness of CREIMBO make it a valuable contribution to the neuroscience field, and I recommend it for acceptance.

**Additional Comments On Reviewer Discussion:**

A reviewer requested comparisons of CREIMBO to baseline models in real-world datasets. In response, the authors added comparisons to SLDS and rSLDS variants, emphasizing the differences in the representations provided by these models versus CREIMBO. These comparisons were incorporated into the manuscript and are now presented in Figures 29–31.

Another reviewer raised the need for analyzing results across a larger number of subjects in real-world datasets. To address this, the authors provided additional analyses, including data and figures from more subjects. These updates are detailed in Figures 17, 18, 20, and 21, ensuring that the manuscript now includes results that better reflect subject-level variability.

Finally, a reviewer sought clarification on the ablation experiments and the effect of excluding specific model components. The authors explained the rationale behind these experiments and provided detailed comparisons between CREIMBO and its ablated versions. The manuscript was updated accordingly to include these clarifications and results, further illustrating the importance of each model component.

I consider these comments to be the most important and they were all addressed by the authors.

---

### Decision · Program_Chairs · 2025-01-22

Accept (Spotlight)